# Proteomic characterization of gastric cancer response to chemotherapy and targeted therapy reveals potential therapeutic strategies

Yan Li [1,12], Chen Xu [2,12], Bing Wang [3,12], Fujiang Xu [1,4,12], Fahan Ma [1,12], Yuanyuan Qu [5,6,7], Dongxian Jiang [2], Kai Li [1], Jinwen Feng [1], Sha Tian [1], Xiaohui Wu [1], Yunzhi Wang [1], Yang Liu [1], Zhaoyu Qin [1], Yalan Liu [2], Jing Qin [8], Qi Song [1], Xiaolei Zhang [2], Akesu Sujie [2], Jie Huang [2], Tianshu Liu [9] ✉, Kuntang Shen [8] ✉, Jian-Yuan Zhao [10,11] ✉, Yingyong Hou [2] ✉ & Chen Ding [1] ✉

Chemotherapy and targeted therapy are the major treatments for gastric cancer (GC), but drug resistance limits its effectiveness. Here, we profile the proteome of 206 tumor tissues from patients with GC undergoing either chemotherapy or anti-HER2-based therapy. Proteome-based classification reveals four subtypes (G-I–G-IV) related to different clinical and molecular features. MSI-sig high GC patients benefit from docetaxel combination treatment, accompanied by anticancer immune response. Further study reveals patients with high T cell receptor signaling respond to anti-HER2-based therapy; while activation of extracellular matrix/PI3K-AKT pathway impair antitumor effect of trastuzumab. We observe CTSE functions as a cell intrinsic enhancer of chemosensitivity of docetaxel, whereas TKTL1 functions as an attenuator. Finally, we develop prognostic models with high accuracy to predict therapeutic response, further validated in an independent validation cohort. This study provides a rich resource for investigating the mechanisms and indicators of chemotherapy and targeted therapy in GC.

Gastric cancer (GC) is one of the most common malignant tumors of the digestive system, and the second leading cause of cancer-related deaths worldwide[1]. The risk factors for GC include smoking, a high-salt diet, a high intake of meats, bile reflux, and infection with *Helicobacter pylor*[2]. Currently, surgery, chemotherapy, and radiotherapy are the major treatment strategies for GC. However, 70–90% of the patients with GC are diagnosed at advanced stages, with poor prognosis. The MAGIC trial revealed that the 5-year survival rate of patients receiving perioperative chemotherapy is significantly higher than those undergoing only surgical resection (36% vs. 23%)[3]. Preoperative chemotherapy, which is

regarded as a standard treatment, is also a promising approach to improve survival in patients with locally advanced GC[3,4]. The triplet combination chemotherapy DOS (docetaxel, oxaliplatin, and S-1)[5] and the doublet chemotherapy XELOX (capecitabine and oxaliplatin)[6] have been established as the first-line therapies in the treatment of both local and metastatic GC. The trastuzumab-based chemotherapy exhibited a survival benefit for human epidermal growth factor receptor-2 (HER2)-positive GC patients in the ToGA trial[7], where an anti-HER2 targeted strategy was proposed as a standard approach for HER2-positive GC patients. Several recent phase II studies and a recent EVIDENCE of the

---

A full list of affiliations appears at the end of the paper. ✉e-mail: liu.tianshu@zs-hospital.sh.cn; shen.kuntang@zs-hospital.sh.cn; zhaojy@fudan.edu.cn; hou.yingyong@zs-hospital.sh.cn; chend@fudan.edu.cn

combination of trastuzumab with a range of treatment patterns have supported the efficacy and safety of trastuzumab + XELOX in patients with HER2-positive advanced GC[8–11]. However, the trastuzumab + DOS combined therapy has been proved less effective with a shorter median OS (27.4 months) compared with trastuzumab + XELOX (34.6 months), which has not been widely used in clinic[12]. The resistance mechanism of the combination of trastuzumab and chemotherapy remained unclear.

Extensive chemotherapeutic drug resistance occurs in DOS, XELOX, and anti-HER2 therapies for gastric cancer. It is reported that 80% of the patients with GC show drug resistance. DOS therapy showed a relatively low clinical efficacy, with an objective response rate (ORR) of 54.5%[13]. Similarly, in our previous study[14], we retrospectively reviewed 248 patients with locally advanced GC treated with a XELOX or DOS regimen and concluded that the response rates in the XELOX and DOS groups were as low as 34.5% and 38.1%, respectively. Even for the 20% HER2-positive GC patients, the ORR was variable (~32–68%). With the increasing emergence of chemotherapy and targeted therapy resistance nowadays, exploration of resistance mechanism and identification of predictive biomarkers to these therapies would be important for improving therapeutic effects for GC patients. Many studies of chemotherapeutic response have focused on a single drug in cell lines but not combined therapy over the past few years[15–17]. Chemotherapeutic drugs of these first-line therapies included fluoropyrimidines (5-fluorouracil, capecitabine, and S-1), platinums (cisplatin and oxaliplatin), and taxanes (paclitaxel and docetaxel). The clinical trials revealed a questionable benefit of triplet chemotherapy compared with doublet treatment, the major difference of which was docetaxel component[18,19]. Docetaxel, as microtubule-targeting antitumor agents, involved in a complex manner and altered multiple cellular oncogenic processes, including mitosis, angiogenesis, apoptosis, inflammatory response, and ROS production[20]. In addition, trastuzumab, a monoclonal antibody, which was used to treat patients with HER2-overexpressing gastric cancer[21]. In the randomised controlled TOGA trial, patients treated with trastuzumab plus cisplatin and fluoropyrimidine chemotherapy had improved median overall survival compared with patients treated with chemotherapy alone[7]. However, the priority of patient selection and prevention of trastuzumab resistance for the trastuzumab combination therapy still await to be convinced. Overall, a comprehensive molecular landscape for predicting the therapeutic response in GC is still lacking.

Several molecular classifications have been proposed to connect molecular patterns to clinical features. The Cancer Genome Atlas (TCGA) project mapped the genomic landscape of GC and classified GC into four subtypes [Epstein-Barr virus (EBV) positive, microsatellite instable (MSI), genome stable (GS), and chromosomal instability (CIN)], which showed distinct salient genomic features, and implicated candidate therapeutic targets[22]. The Asian Cancer Research Group (ACRG) project classified GC based on gene expression data into four subtypes: microsatellite instable (MSI), microsatellite stable (MSS) or epithelial mesenchymal transition (MSS/EMT), MSS with TP53 intact (MSS/TP53+), and MSS with TP53 loss (MSS/TP53-), which were associated with distinct clinical outcomes[23]. Retrospective analyses and large clinical trials suggested that MSI GC patients had a favorable prognosis compared with MSS GC patients, but the benefit from perioperative or adjuvant chemotherapy have been topic of debate[24,25]. According to the MAGIC trial, high MSI was associated with a positive prognostic effect in patients treated with surgery alone and a differentially negative prognostic effect in patients treated with perioperative epirubicin, cisplatin, and fluorouracil chemotherapy[26]. However, no systematic data regarding the outcome of MSI GC patients with first-line chemotherapy and targeted therapy (such XELOX, DOS, and anti-HER2 therapies) has been reported.

Proteins, regarded as the "executors of life," provide insight into the disease at the protein level, and may bridge the gap between research and clinical practice. Previously, we collected surgical samples from 84 patients with diffuse-type gastric cancer (DGC), and presented a proteomic profiling of the Beijing Proteome Research Center (BPRC) DGC cohort. The BPRC DGC cohort was classified into three groups (PX1–3) based on the proteomic profile, in which group 3 (PX3) had the worst prognosis and was resistant to chemotherapy[27]. This suggested an association between the chemotherapy response and proteome signatures in patients. Based on this finding, we sought to establish a comprehensive connection between the proteomic panel and clinical outcomes.

Herein, we set out to investigate the responses to first-line therapies (DOS, XELOX, and anti-HER2-based therapies) for GC through a comprehensive proteomic analysis. We collect the biopsy tumor FFPE samples derived from 206 therapy-naïve GC patients, and construct a GC cohort that covers three clinical therapy subcohorts, including DOS subcohort (44 patients treated with DOS therapy), XELOX subcohort (70 patients treated with XELOX therapy) and HER2 subcohort (71 patients who received anti-HER2-based therapy). Proteomic clustering results in four GC molecular subtypes with distinct functions that show associations with drug response and clinical outcomes. We validate the GC subtyping and their prognosis difference in other independent GC cohorts. We find GC patients with MSI-sig high status show sensitive response to DOS but not XELOX therapy. Bioinformatic analysis reveals higher T cell receptor signaling and aggerated CD8 + Tcm in DOS sensitive group compared with non-sensitive group, while it is opposite in response to XELOX therapy. Further comparative analysis reveals that patients with high TCR signaling are unlikely to benefit from XELOX, but instead the combination of anti-HER2-based therapy; while the activation of ECM and the downstream PI3K-AKT pathway impairs the antitumor effect of trastuzumab. We perform further validation experiment to confirm the synergistic effects of the combination of trastuzumab with XELOX, or the PI3K-AKT inhibitor in vitro. Furthermore, we develop prognostic models with high accuracy to predict the chemotherapeutic response, which are validated by PRM assay in an independent validation cohort composed of 60 GC patients (50% sensitive and 50% non-sensitive patients) receiving either DOS ($N = 20$), XELOX ($N = 20$), or anti-HER2 ($N = 20$) therapies. Finally, we validate CTSE functions as a cell-intrinsic enhancer of chemosensitivity of docetaxel, whereas TKTL1 functions as an attenuator of docetaxel. This study presents a comprehensive proteomic approach for the prediction of the response to chemotherapy and targeted therapy, and implicates its prognostic and therapeutic significance as well as the underlying regulatory mechanisms that may benefit clinical practice.

## Results

### The gastric cancer cohort: The response to chemotherapy and targeted therapy in patients with gastric cancer

To investigate the proteomic patterns associated with the response to chemotherapy and targeted therapy, we collected 206 GC patients which received either DOS therapy (DOS subcohort, $N = 44$, 21.4%), XELOX therapy (XELOX subcohort, $N = 70$, 34.0%), or anti-HER2-based therapy (HER2 subcohort, $N = 71$, 34.5%). Another 21 patients (10.1%) were assigned as "Others," of which 3 patients received apatinib or docetaxel therapies, and 18 cases had no chemotherapy information. All the chemotherapy regimens were given at standard dosing as described in previous studies[8,11,14,28] (Methods). All samples were histologically scored by two expert gastrointestinal pathologists (C.X. and Y.H) according to the widely accepted Response Evaluation Criteria in Solid Tumors (RECIST) (version1.1) by CT/MRI scanning and grouped into complete response (CR), partial response (PR), stable disease (SD), or progressive disease (PD). Here, the ORR, defined as PR plus CR, was selected for the efficacy evaluation; patients with CR and PR were defined as sensitive (S) and those with SD and PD were defined as non-sensitive (NS). DOS subcohort was grouped into DOS-sensitive group (DSG, $N = 22$) and DOS-non-sensitive group (DNSG $N = 22$); XELOX subcohort was grouped into XELOX-sensitive group (XSG, $N = 27$), XELOX-non-sensitive group (XNSG, $N = 42$), and NA (not available of

the information of therapy response), $N = 1$]; HER2 subcohort was grouped into anti-HER2-sensitive group (HSG, $N = 32$), anti-HER2-non-sensitive group (HNSG, $N = 37$), and NA ($N = 2$). Archival formalin-fixed paraffin-embedded (FFPE) tissues with at least 80% tumor purity were collected from 206 chemotherapy-naive patients with GC, and then subsequently analyzed by the mass spectrometry (MS)-based label-

free quantification strategy[27,29] (Fig. 1a and Supplementary Fig. 1a). The detailed clinical characteristics were shown in Supplementary Table 1, Fig. 1b, and Supplementary Data 1. We made a formal assessment of baseline clinical characteristics of all patients enrolled in the study, especially in DOS, XELOX, and HER2 subcohorts. There was no baseline difference in gender, age, grade, and Lauren type in the DOS, XELOX,

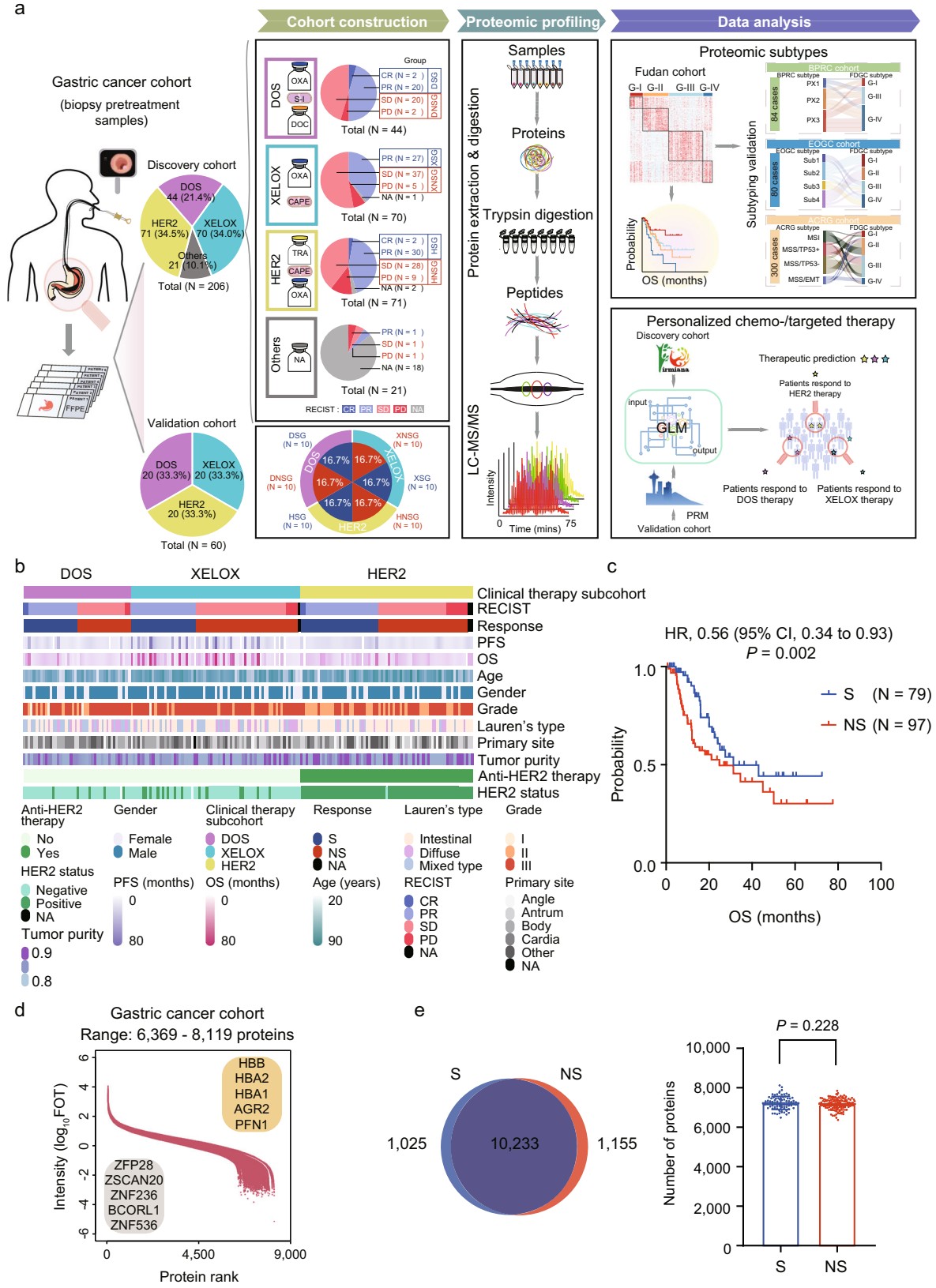

**Fig. 1 | Summary of the proteomic analysis of gastric cancer. a** The proteomics workflow involved three modules: cohort construction (including discovery cohort and validation cohort), proteomic profiling, and data analysis. The proteins were trypsin-digested, and then analyzed in a single-run (75 min) high-performance liquid chromatography mass spectrometry (HPLC-MS) using a Q Exactive HF-X Hybrid Quadrupole-Orbitrap Mass Spectrometer. MS proteomics data were quantified with the Firmiana proteomics workstation, and PRM-MS proteomics data were quantified with Skyline-daily. **b** The gastric cancer (GC) cohort included three clinical therapy subcohorts: DOS, XELOX, and HER2. Clinical parameters are indicated in the heatmap. **c** Kaplan–Meier plots show significant differences between the sensitive group (S) and non-sensitive group (NS) in overall survival (OS). *P*-value

is calculated by two-sided log rank test. **d** Overview of the proteomic profile of patients with GC. Shown are the dynamic range of the protein identification of each sample according to the descending sort of protein abundance in this sample. Range: 6369–8119 proteins. Proteins were quantified as a normalized intensity-based fraction of total (FOT) value and $\log_{10}$ transformed. The highest- and lowest-abundance proteins are shown in the box. **e** Venn diagram showing the protein overlap of S and NS. Number of proteins were quantified in S and NS (two-sided Student's t test, *P* = 0.228). *n* (S) = 82, and *n* (NS) = 103 biologically independent samples examined. Data are shown as mean values ± SD. *P* < 0.05 considered statistically significant. Source data are provided as a Source Data file.

and HER2 subcohorts (Supplementary Data 1). Survival analysis showed that the sensitive group (S, *N* = 79) had a higher survival rate than the non-sensitive group (NS, *N* = 97) (log rank test, *P* = 0.002; hazard ratio (HR) = 0.56; 95% CI, 0.34–0.93) (Fig. 1c).

## Proteomic analysis of the gastric cancer cohort

For the quality control of the performance of mass spectrometry, the HEK293T cell lysate was measured every two days as the quality-control standard, which was adopted in proteomic studies[29,30]. A Pearson's correlation coefficient was calculated for all the quality-control runs, and the results were shown in Supplementary Fig. 1b. The average correlation coefficient among the control samples was 0.964, demonstrating the consistent stability of the MS platform. Proteomics measurement resulted in 6369–8119 gene products (GPs) in each sample (Fig. 1d and Supplementary Fig. 1c, d). A total of 12,519 gene products (GPs) were identified in all patient samples (*N* = 206) (Supplementary Data 2), 10,496 GPs, 10,766 GPs, 10,920 GPs were identified in the DOS, XELOX, and HER2 subcohorts, respectively (Supplementary Fig. 1e). Principal components analysis (PCA) revealed no batch effects among DOS, XELOX, and HER2 subcohorts (Supplementary Fig. 1f). No major differences in the proteomic coverage between S (11,258 GPs) and NS (11,388 GPs) was observed (Student's t test, *P* > 0.05) (Fig. 1e). Our study has so far presented a comprehensive view of the proteomic landscape of this GC cohort treated with first-line therapies.

## Proteomic subtyping of the GC cohort and their association with a therapeutic response

Consensus clustering analysis was performed on the 206 samples based on 1,000 most variable proteins (proteins with the top 10% median absolute deviations) (Methods), which resulted in four subtypes: G-I (*N* = 29), G-II (*N* = 60), G-III (*N* = 97), and G-IV (*N* = 20) (Supplementary Fig. 2a and Supplementary Data 3). Further analysis of 179 patients with complete prognosis among the four proteomic subtypes revealed significant association of proteomic subtypes with survival, among which the G-IV subtype had the worst overall survival (log rank test, *P* = 0.034) (Supplementary Fig. 2b). After excluding the patients who were lost to follow-up, we then performed the same consensus clustering analysis only on the 179 patients. Consensus clustering analysis of the proteomic profiles among 179 samples identified four proteomic subtypes of GCs, in which 28, 56, 75, and 20 patients were grouped into subtypes G-I, G-II, G-III, and G-IV, respectively (Fig. 2a and Supplementary Fig. 2c, and Supplementary Data 3). Sankey plot showed high concordance between the two proteomic subtyping systems of 206 samples and 179 samples (Supplementary Fig. 2d and Supplementary Data 3). We then mainly focused on the proteomic subtyping system of 179 samples with the complete survival information during the further analysis. The proteomic subtypes of 179 samples displayed distinct clinical outcomes (therapy response and overall survival). The chemotherapy and targeted therapy response exhibited a gradual resistance phenomenon from G-I to G-IV, as the percentage of sensitive patients (CR and PR) dramatically decreased from 60% in G-I to 20% in G-IV. Conversely, the percentage of non-sensitive patients (SD and PD) increased from 40% in G-I to 80% in G-IV (Fisher's exact test, *P* = 0.04) (Fig. 2b and

Supplementary Fig. 2e). Among four subtypes, the G-IV had the worst prognosis (log rank test, *P* < 0.05) (Fig. 2c). In addition, an obvious association between proteomic subtyping and therapy subcohort or TNM stage was determined (Fisher's exact test, *P* < 0.01), but this association was not observed with either grade, Lauren's type, the primary site, or tumor purity (*P* > 0.05) (Fig. 2b and Supplementary Fig. 2e). Further statistical analysis revealed the significant distribution difference of therapy subcohorts among proteomic subtypes, which was mainly derived from HER2 subcohort (mainly enriched in G-II subtype (Fisher's exact test, *P* < 0.05)); while no significant difference was observed in DOS and XELOX subcohorts (Supplementary Fig. 2e). The distribution difference among proteomic subtypes could be caused by the specific proteomic feature related to HER2 expression, due to the selection of HER2-positive patients for anti-HER2 targeted therapy, which was further validated by HER2 evaluation by IHC and FISH analysis (Supplementary Fig. 2e). This result showed the specific molecular pattern in HER2-positive GC patients, which could be identified at proteome level. Formal statistical tests for interaction analysis revealed there was no significant interaction between therapy responses and other baseline clinical characteristics, such as therapy subcohort and TNM stage (Supplementary Table 2). Furthermore, univariate cox analysis of overall survival showed proteomic subtyping was associated with clinical outcome irrespective of gender, age, and therapy subcohort. Remarkably, proteomic subtyping served as an independent predictive factor (Cox *P* trend = 0.009; HR = 1.49; 95% CI, 1.10 to 2.02) in the multivariable analysis after adjusting for clinical TNM stage and other covariates (Supplementary Table 3). Overall, these results demonstrated the strong association of proteomic subtyping with therapy response and prognosis, and supported its reliability at proteomic level.

Comparative analysis of proteomic profiling resulted in 301 (G-I), 611 (G-II), 925 (G-III), and 467 (G-IV) GPs (*P* < 0.05; fold change >2; identification frequency ≥10%), showing distinct molecular features among the four proteomic subtypes (Supplementary Fig. 3a). We performed a functional enrichment analysis according to the Kyoto Encyclopedia of Genes and Genomes (KEGG) pathway annotations, and determined the dominant bioprocesses of each subtype. As shown in Supplementary Fig. 3b, c, the same criteria were applied in proteomic subtypes of 206 samples, and KEGG pathway enrichment analysis revealed the consistency of the overrepresented pathways dominant in four proteomic subtypes both in 179 samples and 206 samples. The G-I subtype was dominant for endocytosis (*P* = 4.73E − 4), and ssGSEA analysis revealed that the activation of endocytosis pathway indicted a better prognosis (log rank test, *P* < 0.05) (Supplementary Fig. 3d). As shown in Supplementary Fig. 3e, a group of endocytosis related proteins, such as DNM2, EPS15, WIPF1, ACAP2, and CHMP6, showed positive association with overall survival (hazard ratios range: 0.10–0.54, *P* < 0.05). In the G-II subtype, we observed a significant enrichment of glycolysis/gluconeogenesis (*P* = 3.53E − 2) and pantothenate/CoA biosynthesis (*P* = 3.59E − 3) (Fig. 2d and Supplementary Data 3). In the G-III subtype, lysosomal acid hydrolases proteases (such as CTSA, CTSB, CTSC, and CTSE) and synthesized lysosomal enzymes (such as CLTA, CLTB, GGA1, and

AP1S2) were exclusively upregulated. In contrast with other three subtypes, the G-IV subtype was characterized by ECM-receptor interaction, focal adhesion, complement/coagulation cascades, and PI3K-AKT signaling pathway, of which the activation of ECM-receptor interaction pathway indicted a worse prognosis (log rank test, $P < 0.05$) (Supplementary Fig. 3f). The extracellular matrix proteins, such as FGB, TGFB1, THSD4, LAMB2, and LAMB4, exhibited a significant upregulation in G-IV subtype compared with other subtypes (Student's $t$ test, $P < 0.05$; fold change >2), among which the expression of THSD4 was significantly associated with poor prognosis ($P = 0.007$; HR = 1.96; 95% CI, 1.12–3.44) (Supplementary Fig. 3g). These results suggested that activation of endocytosis in G-I subtype was associated with drug sensitivity; while high expression of ECM in G-IV subtype was associated with drug resistance. GSEA analysis further revealed the association of endocytosis/ECM pathways and drug sensitivity/resistance was also applied for other proteomic subtypes (Supplementary Fig. 3h), indicating the consistency of therapy mechanism in different proteomic subtypes.

To investigate the association of proteins expression with therapy resistance, we analyzed the correlation of the extracellular matrix proteins significantly upregulated in the G-IV subtype with drug sensitivity (half maximal inhibitory concentration [IC50]) using gastric cancer cell lines data from the Cancer Dependency Map Project (DepMap). We identified five proteins, including THSD4, SRPX2, TGFBI, THBS1, and LAMB2, which showed high correlation with drugs response (5-FU, oxaliplatin, or docetaxel) (Pearson $r > 0.4$, $P < 0.05$) (Fig. 2e). Among these proteins, only THSD4[31] was resistant to all three drugs (5-FU, oxaliplatin, or docetaxel), and the high expression of THSD4 was significantly associated with poor prognosis (log rank test, $P < 0.05$) (Fig. 2e and Supplementary Fig. 3i). To further verify that high expression of THSD4 was associated with drug resistance, we firstly overexpressed THSD4 in two gastric cancer cell lines MKN45 and MGC803 (Supplementary Fig. 3j, k). The CCK8 assay revealed that THSD4 overexpression significantly promoted the proliferation of MKN45 and MGC803 cells (Student's $t$ test, $P < 0.05$), compared with the cells transfected with an empty vector (Supplementary Fig. 3l, m). Then, we treated THSD4-overexpressing and empty vector-overexpressing MKN45 and MGC803 cells with docetaxel, oxaliplatin, and 5-FU, respectively. The drug sensitivities in the MKN45 and MGC803 cells were estimated by their half-maximal inhibitory concentration (IC50) values. We observed IC50 values of docetaxel, oxaliplatin, and 5-FU were significantly higher (4.26-fold, 1.44-fold, and 1.38-fold increase in MKN45, respectively; Student's $t$ test, $P < 0.05$) in the THSD4-overexpressing MKN45 cells, compared with the empty vector-overexpressing cells (Fig. 2f). Consistently, we observed the similar change of IC50 values of docetaxel, oxaliplatin, and 5-FU in THSD4-overexpressing MGC803 cells (3.60-fold, 1.47-fold, and 1.20-fold increase in MGC803, respectively) (Fig. 2f). In conclusion, the in vitro experiments further validated THSD4 overexpression reduced the anti-tumor effect of chemotherapeutic drugs including docetaxel, oxaliplatin, and 5-FU.

The parallel reaction monitoring (PRM) assays are powerful targeted approaches to detect and quantify pre-specified proteins with a high throughput using high-resolution mass spectrometers. We firstly constructed an independent cohort composed of 60 GC patients receiving either DOS ($N = 20$: DSG, $N = 10$; DNSG, $N = 10$), XELOX ($N = 20$: XSG, $N = 10$; XNSG, $N = 10$), or anti-HER2 ($N = 20$: HSG, $N = 10$; HNSG, $N = 10$) therapies (Supplementary Data 3). To further validate the association between extracellular matrix proteins and resistance of drugs (5-FU, oxaliplatin, and docetaxel), we employed the targeted MS approach, PRM assays, which has been adopted in classifier's validation in recent proteomic research[32,33], to quantify these proteins in FFPE tumor tissues from patients receiving DOS therapy (triplet combination chemotherapy of 5-FU, oxaliplatin, and docetaxel). We then selected a set of target peptides that unique to these ECM proteins (including THSD4, SRPX2, TGFBI, THBS1, and LAMB2) using the

library search results (Supplementary Data 3). The fragment total areas of targeted peptides reported by Skyline-daily (4.2.1.19004, University of Washington, USA) were used to quantify these proteins. As a result, we observed these ECM proteins were higher expressed in DNSG compared with DSG in PRM-MS experiments (Fold change (DNSG/DSG) > 2, $P < 0.05$, Wilcoxon rank-sum test): THSD4 (Fold change (DNSG/DSG) = 2.49, $P = 5.2E – 3$), SRPX2 (Fold change (DNSG/DSG) = 7.10, $P = 0.035$), TGFBI (Fold change (DNSG/DSG) = 1.97, $P = 0.035$), THBS1 (Fold change (DNSG/DSG) = 7.63, $P = 1.5E – 3$), and LAMB2 (Fold change (DNSG/DSG) = 2.94, $P = 0.043$) (Fig. 2g). In conclusion, our data demonstrated the association between these ECM proteins and drug resistance was validated by PRM approach in the independent cohort. Overall, the results illustrated the high expression of extracellular matrix proteins is associated with drug resistance, and these extracellular matrix proteins could serve as indicators to predict chemotherapy response.

**MSS/MSI status of gastric cancer are associated with tumor immune microenvironment and therapy resistance**

To further test if the proteomic subtyping algorithm of the Fudan GC cohort (FDGC) was feasible in other GC cohorts, we constructed the Fast Large Margin classifier model based on the overrepresented proteins in the subtypes of the FDGC cohort using RapidMiner 9.6.0 (Methods). Then, we applied the same model with FDGC subtyping algorithm in three independent cohorts of GC patients (BPRC cohort[27], $N = 84$; EOGC cohort[34], $N = 80$; and ACRG cohort[23], $N = 300$). Consistent with the FDGC cohort, the application of the model in BPRC cohort, EOGC cohort, and ACRG cohort resulted in subtype reallocations: G-I ($N = 15$), G-II ($N = 0$), G-III ($N = 31$), and G-IV ($N = 38$) in BPRC cohort; G-I ($N = 20$), G-II ($N = 20$), G-III ($N = 19$), and G-IV ($N = 21$) in EOGC cohort; G-I ($N = 35$), G-II ($N = 103$), G-III ($N = 127$), and G-IV ($N = 35$) in ACRG cohort (Supplementary Data 4). We observed a significant concordance between FDGC subtype and EOGC subtype or ACRG subtype ($P = 8.5E – 9$, $P = 5.0E – 4$, respectively), but not between FDGC subtype and BPRC subtype ($P = 0.422$) (Fig. 3a, b, and Supplementary Fig. 4a, b), which might due to the similar composition of Lauren's type between FDGC subtype and EOGC subtype or ACRG subtype, while BPRC cohort only included diffuse-type cancer. Although no concordance in the composition of Lauren's type between FDGC subtype and BPRC cohort, the proteomic features associated with therapy response and prognosis revealed in FDGC subtypes were applicable in the BPRC cohort (log rank test, $P = 0.021$) (Fig. 3a and Supplementary Fig. 4a). In the comparison of FDGC and the BPRC subtypes, we found the PX3 subtype accounted for the highest proportion in G-IV subtype (50%) compared with other subtypes (20% in G-I, 26% in G-III, respectively), indicating PX3 subtype was mostly clustered into G-IV subtype. As reported in BPRC subtype, PX3 subtype was characterized with the enrichment of pathways including ECM organization, EMT, and complement, had the worst prognosis and was resistant to chemotherapy. Consistently, in our study, G-IV subtype was characterized by ECM-receptor interaction, focal adhesion, and complement/coagulation cascades, had the worst prognosis and highest proportion (80%) of non-sensitive patients. As for EOGC cohort, the comparison of FDGC subtype and EOGC subtype suggested the Sub2 accounted for most (75%) of G-I subtype. As reported in the research related to EOGC subtype, the Sub2 had the best survival; consistently, in our study, the G-I had the best survival. Importantly, besides the significant concordance with ACRG subtype, FDGC subtyping stratified patients of ACRG cohort into four groups with distinct prognosis (log rank test, $P = 0.022$) (Fig. 3b). Taken together, these results indicated the similar survival patterns in the comparison of FDGC subtype with other subtypes, demonstrating the robustness of our proteomic subtyping in other GC cohorts.

The comparison of FDGC subtype and ACRG subtype showed MSS/EMT subtype originated from ACRG subtyping accounted for 56%

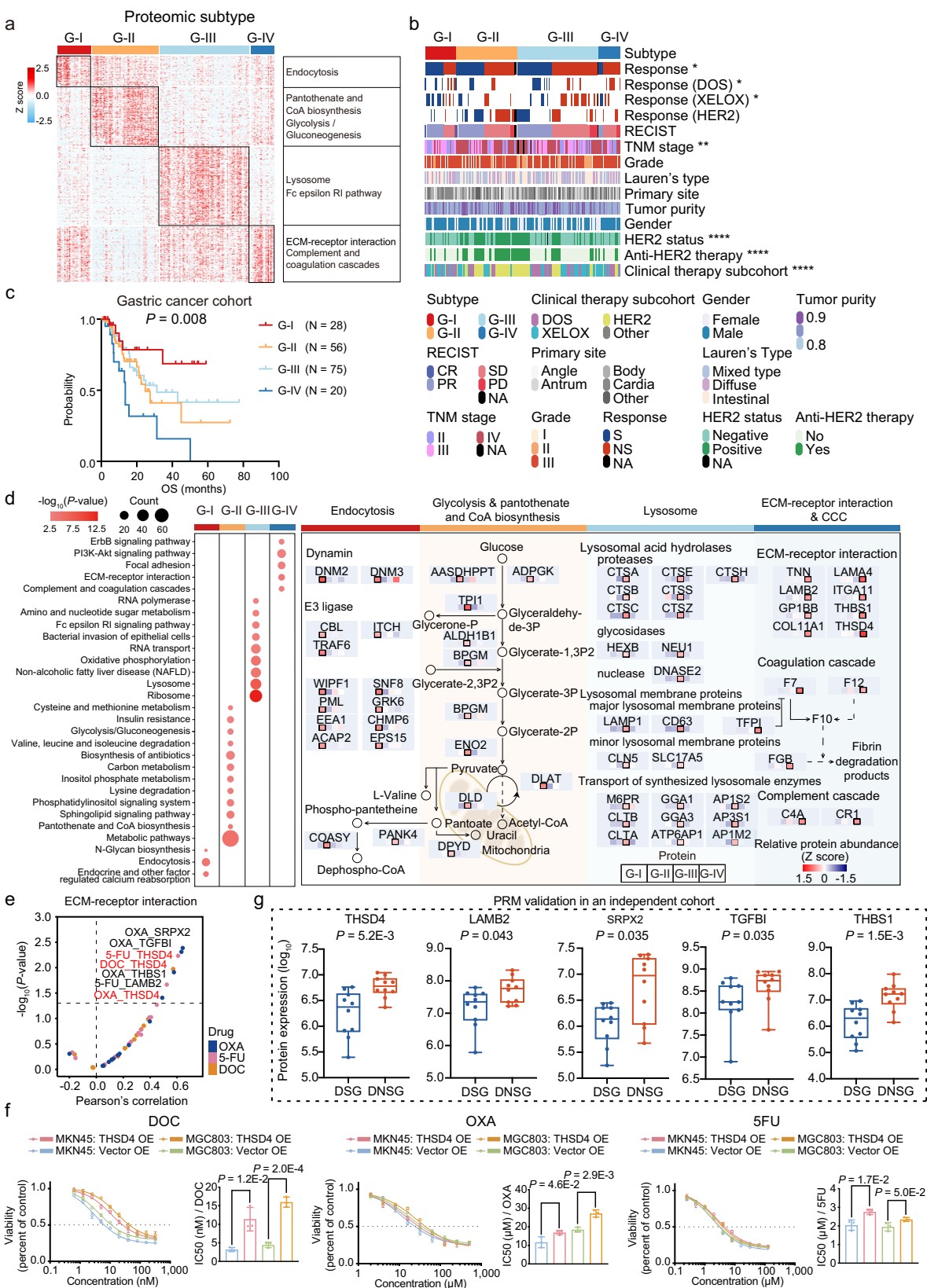

of the G-IV subtype. We speculated that the characteristics of MSS or EMT might be associated with poor survival. Therefore, we further predicted the microsatellite instability (MSI)[35] of the ACRG cohort classified by FDGC subtyping based on a small pre-defined set of gene expression signatures. We found the G-IV subtype was characterized with lower MSI/MSS-sig score, while higher MSI/MSS-sig score was

enriched in the G-II subtype (Fig. 3c and Supplementary Fig. 4c). We assessed whether the MSI/MSS-sig score was associated with distinct somatic alterations, and observed that MSI/MSS-sig score or MSI high (MSI-H) assay status of the G-II was highly consistent with the presence of hypermutation, with mutations in genes such as *MLL4* (10%), *FAT4* (10%), *PIK3CA* (14%), and *KRAS* (8%). Among them, *PIK3CA* and *KRAS*

**Fig. 2 | Proteomic subtyping of the gastric cancer cohort and its association with clinical characteristics. a** The heatmap depicts the relative abundance (Z score of FOT) of the signature proteins in four subtypes of 179 GC samples. Biological functions related to these signature proteins are denoted on the right. **b** The association of four proteomic subtypes with clinical characteristics (including gender, grade, Lauren's type, primary site, HER2 status, RECIST, and tumor purity, etc.) are annotated with *P-values (two-sided Fisher's exact test for categorical variables, and two-way ANOVA test for continuous variables). **c** The Kaplan–Meier curves of overall survival (OS) of each proteomic subtype (G-I, n = 28; G-II, n = 56; G-III, n = 75; and G-IV, n = 20). P-value is calculated by two-sided log rank test. **d** Left panel: Bubble plot showing the KEGG pathway enrichment (two-sided Fisher's exact test) of each proteomic subtype. Right panel: Diagram illustrating the differentially expressed signatures and signaling cascades involved in G-I to G-IV. The little heatmap under each protein depicted the Z score of average protein abundance in

each proteomic subtype. Red, upregulated proteins; blue, downregulated proteins. **e** The correlation of the extracellular matrix proteins and different drugs, including 5-FU, OXA, and DOC. The drug sensitivity (half maximal inhibitory concentration [IC50]) of gastric cancer cell lines was from the Cancer Dependency Map Project (DepMap). P-values are derived from two-sided Pearson's correlation test. **f** Dose-response curves of MKN45 and MGC803 cell lines overexpressing THSD4 after 72-h treatments with DOC, OXA, and 5FU. Barplots showing the comparison of IC50 values in each group. Bars represent the mean of n = 3 independent experiments with error bars indicating SD. P-values are calculated using two-sided Student's t test. **g** Boxplot showing the differential expression of ECM proteins validated by PRM assay (two-sided Wilcoxon rank-sum test, n = 10 biologically independent samples per group). Boxplots show median (central line), upper and lower quartiles (box limits), 1.5× interquartile range (whiskers). Source data are provided as a Source Data file.

were reported as recurrent neoantigen-associated mutations[36]. In addition, further analysis of copy number profiles in ACRG cohort, we found that genomic instability index (termed CNV GI), which was present in 28% cases, showed significant difference among four subtypes (G-I to G-IV) (Fisher's exact test, P = 0.011) and G-IV had the lowest CNV GI (Fig. 3c and Supplementary Fig. 4d). We applied the same set of gene expression signatures to assess the expression of MSI/MSS-sig, and observed the similar trend in FDGC cohort (ANOVA test, P < 1.0E − 4), which suggested the robustness of FDGC subtyping (Fig. 3c and Supplementary Fig. 4c). The low MSI/MSS-sig significantly correlated with poor clinical outcome both in ACRG cohort and FDGC cohort (log rank test, P < 0.05) (Fig. 3d), which was consistent to the previous studies[25,37]. Importantly, the MSI/MSS-sig level showed significant association with the response to chemotherapy and targeted therapy (Fisher's exact test, P = 0.049) (Supplementary Fig. 4e). Furthermore, we found that gastric cancer with high MSI/MSS-sig was featured by oxidative phosphorylation, while gastric cancer with low MSI/MSS-sig showed high expression of extracellular matrix proteins (Fig. 3e).

As previously reported, antigen-driven immune response could be activated in microsatellite instable (MSI) cancers[38]. We firstly evaluated the relative abundance of the cytokines and proteins involved in antigen processing and presenting process among four proteomic subtypes. We observed the consistent downregulation of a group of cytokines (such as CXCL17, IL16, and IL18), proteins involved in antigen processing (such as CASP1, CTSE, PSMD8, and TRIM32), and MHC class II molecules (such as HLA-DMB, HLA-DPB1, and HLA-DQA1) in G-IV subtype (Fig. 3f). For further exploring the possible molecular mechanisms of low MSI/MSS-sig accounting for drug resistance and poor prognosis, we then evaluated the tumor immune microenvironment among four subtypes by xCell analysis (Supplementary Data 5). As reported, the high immunoactivity defined by the microsatellite instability (MSI) is associated with the high degree of infiltration of M1 macrophages[39]. We found that G-IV had the lowest MSI/MSS-sig score (Fig. 3g). In addition, the proteomic subtypes were featured with different cell types, among which macrophages M1 were enriched in G-II. In contrast with macrophages M1, precursor monocytes were aggregated in the G-IV subtype (Fig. 3g, h). Further analysis suggested the significant correlation between MSI/MSS-sig and monocytes (Pearson r = −0.37, P = 2.9E − 7) or macrophages M1 (Pearson r = 0.25, P = 9.3E − 4) (Fig. 3i). Increasing studies have also shown that tumor-associated macrophages (TAMs) can either enhance or antagonize the antitumor efficacy of cytotoxic chemotherapy. For example, the skewing of TAMs to M1-like phenotype contributed to the anti-tumor and anti-angiogenic effects of pharmacological agents such as 5-fluorouracil and docetaxel, respectively in colorectal cancer and breast cancer[40,41]. Therefore, we investigated the effects of monocytes/macrophages M1 on the clinical outcome of GC patients with chemotherapy and targeted therapy. Survival analysis of significantly differential cell types among four proteomic subtypes revealed that xCell

score of monocytes/macrophages M1 showed significant association with prognosis in the patients of stage IV (log-rank P = 0.041, HR = 1.79, 95% CI, 0.99–3.24; log-rank P = 0.013; HR = 0.55; 95% CI, 0.30–1.01, respectively), but not of stage II and III (Fig. 3j and Supplementary Fig. 4f–h). Consistently, these results were also validated in the independent ACRG cohort (Fig. 3j and Supplementary Fig. 4i). Furthermore, we observed that biomarkers of macrophages M1, including HLA-DRA, HLA-DRB3, and IL18, were significantly increased in MSI/MSS-sig high group (Fig. 3k). Among them, IL18, as a secreted pro-inflammatory factor of macrophage M1, showed significantly positive correlation with MSI/MSS-sig both in FDGC cohort and ACRG cohort (Pearson r = 0.31, P = 1.8E − 5; Pearson r = 0.37, P = 2.8E − 11, respectively). The high expression of IL18 also reflected significant association with prognosis in patients of stage IV (log-rank P = 0.035, HR = 0.49, 95% CI, 0.22−0.83) (Fig. 3l). These results demonstrated the MSI/MSS-sig level correlated with clinical outcome; in particular, the MSI/MSS related immune cell types (monocytes and macrophages M1) have been shown to be associated with clinical outcome in the patients of stage IV, indicating the importance of immune cell types to the prognosis of patients in stage IV. Overall, we proposed microsatellite stable tumor cells featured with aggregated monocytes and few macrophages M1 were prone to drug resistance and had a poor prognosis (Fig. 3m).

## T cell receptor signaling pathway exerts diverse effects in response to DOS and XELOX chemotherapy

A better outcome for patients with MSI-H tumors than with MSS tumors has been reported in gastrointestinal tumors[25,42], whereas the benefit of adjuvant chemotherapy seems to be unclear according to exploratory analyses of recent phase III trials. We next examined the association of MSI/MSS-sig level with clinical outcomes in GC patients receiving different chemotherapy regiments. Surprisingly, we observed DSG had a higher MSI-sig level than DNSG (P = 0.02), while no obvious difference between XSG and XNSG (P > 0.05) (Supplementary Fig. 5a). Consistently, survival analysis also indicated GC patients with MSI-sig high level had a positive association with overall survival in DOS therapy; while we didn't observe the difference in association of MSI/MSS-sig level with clinical outcomes in XELOX therapy (Fig. 4a). These results indicated MSI-sig high GC patients could benefit from DOS therapy, but not XELOX therapy.

To investigate the mechanism of sensitivity/resistance of different treatment regimens, we further examined the functional differences between sensitive group and non-sensitive group in DOS and XELOX subcohorts, respectively. We firstly compared the proteome of DSG, DNSG, XSG, and XNSG, and identified the overrepresented proteins (ORPs) of each group (twofold changes in each group at a 1% false discovery rate (FDR) at the protein levels) (Supplementary Fig. 5b and Supplementary Data 6 and 7). We then performed pathway enrichment on the overrepresented proteins of each group according to the KEGG pathway annotations shown in Fig. 4b. Surprisingly, we found that Fc

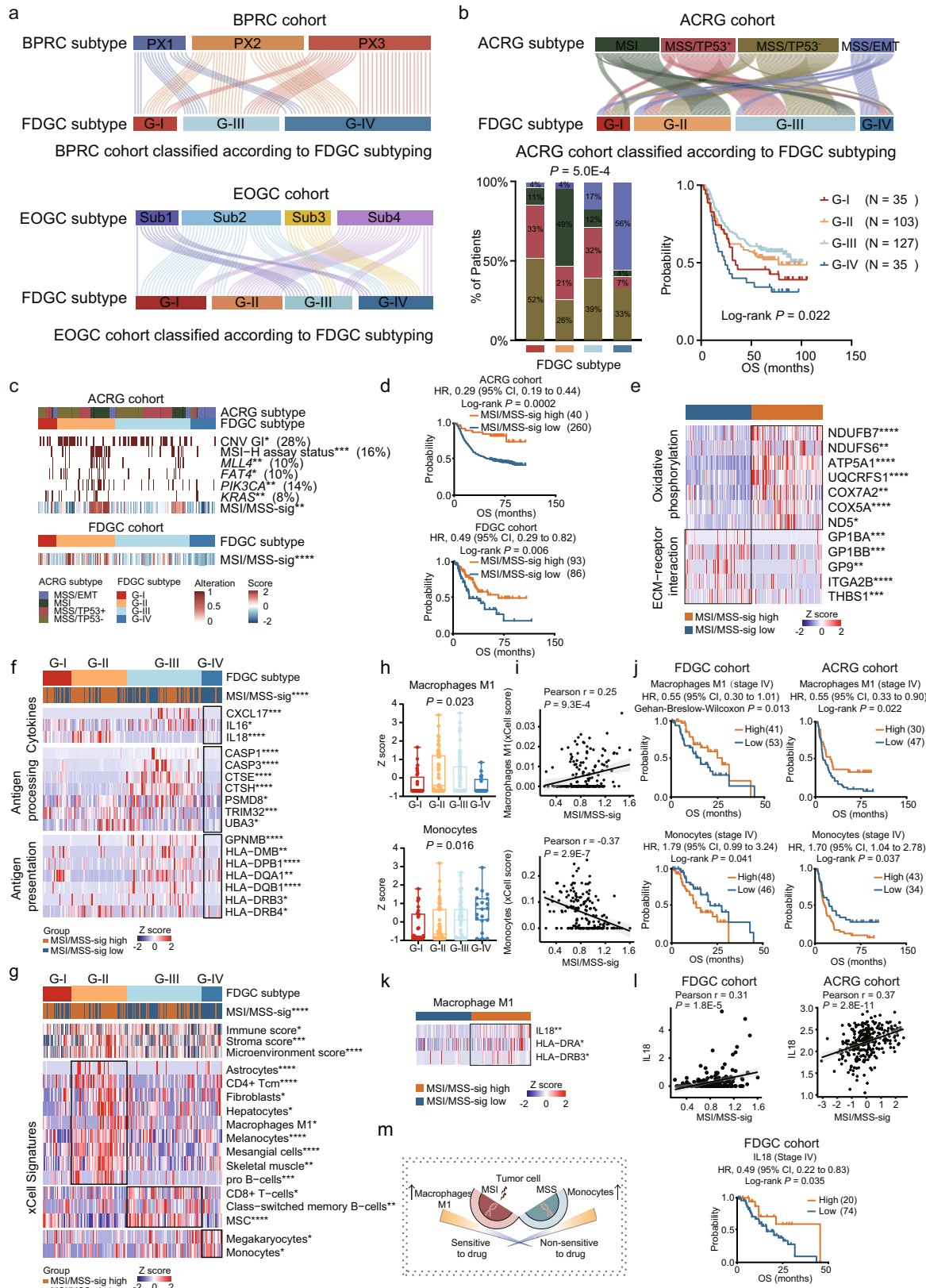

epsilon RI signaling, TNF signaling, T cell receptor (TCR) signaling, and B cell receptor (BCR) signaling pathways were enriched in DSG and XNSG ($P < 0.05$) (Fig. 4b and Supplementary Data 6 and 7). Among these pathways, GSEA analysis showed that TCR signaling pathway was enriched in DSG and the high activation of this pathway represented better prognosis in DOS subcohort ($P = 0.015$; HR = 0.18; 95% CI, 0.04–0.88);

on the contrary, in the XELOX subcohort, TCR signaling pathway was enriched in XNSG and the high expression represented poor prognosis ($P = 0.013$; HR = 5.19; 95% CI, 2.15–12.52) (Fig. 4c). Meanwhile, the TCR signaling pathway enriched in XELOX non-sensitive patients was validated in BPRC cohort received XELOX therapy (Fig. 4d). The xCell analysis also showed that immune cells such as CD8 + Tcm, CD8 + Tem,

**Fig. 3 | The application of FDGC subtyping algorithm in multiple independent cohorts, and the association of MSI/MSS characteristics with clinical outcomes. a** Sankey diagram indicating the comparison of FDGC subtype and BPRC subtype or EOGC subtype. **b** Sankey diagram and barplot indicating the comparison of FDGC subtype and ACRG subtype. Survival analysis of ACRG cohort classified by FDGC subtyping (two-sided log rank test). **c** Heatmap of CNV GI (copy number variation genomic instability index), MSI-H assay status, mutations, and MSI/MSS-sig (microsatellite instability/microsatellite stability gene expression signatures assessment) of ACRG cohort classified by FDGC subtyping (upper). Heatmap of MSI/MSS-sig of FDGC subtype (bottom). P-values are calculated by two-sided Fisher's exact test (categorical variables) and two-way ANOVA test (continuous variables). **d** The association of MSI/MSS-sig level with OS in ACRG and FDGC cohorts (two-sided log rank test). **e** Heatmap illustrating significantly differential expressed proteins in MSI/MSS-sig high and low group (two-sided Wilcoxon rank-sum test). **f** Heatmap illustrating down-regulated proteins in G-IV compared with other subtypes (Kruskal-Wallis test). **g** Heatmap illustrating the dominant cell type compositions of G-II, G-III, and G-IV (two-sided Student's $t$ test). **h** Boxplot showing the xCell score of Macrophages M1 and monocytes among four proteomic subtypes (two-sided Student's $t$ test). n (G-I) = 28, $n$ (G-II) = 56, $n$ (G-III) = 75, $n$ (G-IV) = 20 biologically independent samples examined. Boxplots show median (central line), upper and lower quartiles (box limits), 1.5× interquartile range (whiskers). **i** Correlation between MSI/MSS-sig with Macrophages M1 and monocytes (two-sided Pearson's correlation test). **j** The association of Macrophages M1 and monocytes with OS in FDGC and ACRG cohorts (two-sided Gehan-Breslow-Wilcoxon or two-sided log rank test). **k** The abundance of Macrophages M1 markers in MSI/MSS-sig high and low groups (two-sided Wilcoxon rank-sum test). **l** Correlation between IL18 abundance with MSI/MSS-sig (two-sided Pearson's correlation test), and survival analysis of IL18 expression with OS (two-sided log rank test). **m** Diagram illustrating the potential association between MSS/MSI characteristics with drug response. *$P < 0.05$ is considered statistically significant. *$P < 0.05$, **$P < 0.01$, ***$P < 0.001$, ****$P < 0.0001$. Source data are provided as a Source Data file.

B-cells, dendritic cells (cDC and iDC), were enriched in DSG; similarly, we observed CD8 + Tcm, CD8+ T-cells, and cDC were enriched in XNSG (Fig. 4e). Consistently, we found immune effectors such as dendritic cells and T cells were also enriched in XELOX non-sensitive patients in BPRC cohort (Fig. 4f). Among these immune cells, CD8 + Tcm showed positive correlation with TCR signaling pathway (Supplementary Fig. 5c). Collectively, immune modulation exerts diverse effects in response to DOS and XELOX chemotherapy.

The immune-induced anticancer response was reported to contribute to the efficacy of conventional chemotherapeutic agents[43,44]. The major difference of DOS and XELOX in therapeutic regiments was docetaxel component, of which DOS has docetaxel, while XELOX has not. We hypothesized docetaxel might play an important role in the immune-induced anticancer response. We next surveyed the immunogenic cell death (ICD) prediction scores for chemotherapeutic agents in NCI library base on molecular descriptors obtained with the chemistry development kit[45,46]. Significantly, among 31 anticancer drugs, the docetaxel ranked the top one ICD component, indicating higher capability of docetaxel in inducing immunogenic cell death (Fig. 4g).

To search for the indicators in response to DOS and XELOX therapies, we focused on the overlapped up-regulated proteins and pathways in sensitive or non-sensitive response to DOS and XELOX therapy. We observed the overlapped 2053 ORPs among DSG, DNSG, XSG, and XNSG. Almost half (46.5%, 954) of the 2053 ORPs was found in DSG and XSG, DNSG and XNSG, which might due to the common therapeutic components oxaliplatin and 5-FU in XELOX and DOS therapies; while the remaining 53.5% (1099) was found in DSG and XNSG, DNSG and XSG, which might due to the differential therapeutic component docetaxel (Supplementary Fig. 5b). Remarkably, the higher proportion (64.4%, 708 proteins) of the 1099 proteins was found in both DSG and XNSG, which were involved in immune modulation (TCR signaling pathway and immune system) and ERBB pathway (Fig. 4h and Supplementary Fig. 5b). As shown in Fig. 4i, the univariable cox analysis identified a panel of 12 proteins involved in immune modulation (such as BCL2L1, CD81, CTSE, ENAH, MRC1, ORMDL3, RAB24, REG3A, and VNN1) and ERBB signaling (such as ERBB2, MAP2K2, and PPP3CB) showed positive association with disease-free survival (DFS) in DOS subcohort, while negative association in XELOX subcohort. Among the 12 proteins, ERBB2 was a poor prognostic factor in XELOX subcohort (HR = 1.3, Cox P-value = 0.035). To assess statistical independence of prognostic power of ERBB2, we then performed multivariable cox analysis adjusted for baseline clinical covariates. The result illustrated that ERBB2 could serve as an independent prognostic factor ($P = 0.001$; HR = 1.497; 95% CI, 1.169–1.92) in the multivariable analysis after adjusting for Lauren's type, grade and RECIST in XELOX, but not DOS subcohort (Supplementary Table 4). ERBB2, as known as HER2, was the only drug target approved for the first-line treatment of HER2-positive GC. Therefore, we speculated that XELOX combined with anti-HER2 (Trastuzumab, Herceptin) targeted therapy could improve the therapeutic response.

To validate the speculation that the combination of trastuzumab and XELOX therapy have synergistic effects in vitro, we treated NCI-N87 cells (HER2-positive cell line) with XELOX, trastuzumab (TRA), and combination of XELOX and trastuzumab (XELOX + TRA), respectively. We observed the IC50 values of XELOX and TRA were 36.6 μM and 43.7 μM in NCI-N87 cells (Fig. 4j). As indicated in Fig. 4j, the IC50 value of the XELOX in the combination treatment (XELOX + TRA) was significantly decreased from 36.6 μM to 19.6 μM (two-sided Student's $t$ test, $P = 1.4E-2$). Similarly, the IC50 value of TRA in the combination treatment (XELOX + TRA) was also significantly decreased from 43.7 to 19.6 μM (two-sided Student's $t$ test, $P = 8.2E-3$). These results indicated trastuzumab could increase the sensitivity of HER2-amplified human gastric cancer cells to XELOX therapy.

We then compared the combination of XELOX with anti-HER2 targeted therapy and XELOX chemotherapy alone. After examining the association of MSI/MSS-sig level with clinical outcomes, we observed a better prognosis in MSI-sig high GC patients in XELOX + HER2 subcohort (log rank test, $P < 0.05$), but not in XELOX subcohort (Supplementary Fig. 5d and Fig. 4a). For further exploring the sensitive and non-sensitive mechanism of the combination therapy, we next performed functional enrichment analysis on the overrepresented proteins of XHSG and XHNSG [FC (XHNSG vs XHSG) >2 or <0.5], and observed that GC patients featured by TCR signaling, BCR signaling, and Fc epsilon RI signaling pathways showed non-sensitive to XELOX therapy but sensitive to XELOX combined with anti-HER2 targeted therapy (Supplementary Fig. 5e and Supplementary Data 8). GSEA analysis also showed TCR signaling was enriched in XHSG ($P = 0.033$) and positively correlated with clinical outcome ($P = 0.038$; HR = 0.45; 95% CI, 0.18–1.14) (Fig. 4k). The xCell analysis illustrated the abundance of immune cells such as CD4 + T-cells, CD4 + Tcm, and CD8 + Tcm was significantly increased in XHSG ($P < 0.05$) (Fig. 4l). The proteins involved in antigen processing and presentation and CD8 + Tcm markers were also upregulated in XHSG (Supplementary Fig. 5f). We observed activated immune signaling and aggregated immune cells in XHSG, in contrast with XNSG. To directly address the result, we performed immunohistochemistry (IHC) of T-cell marker CD8+ and CD4+ to evaluate tumor-infiltration lymphocytes (IT-TILs) in FFPE tumor tissue, from patients received XELOX therapy or combined with anti-HER2 therapy. Here, we included sensitive and non-sensitive patients treated with XELOX therapy (XSG and XNSG) or XELOX + HER2 therapy (XHSG and XHNSG). As a result, we observed the expression of CD4 and CD8 was significantly increased in FFPE tumor tissues from patients of XNSG compared with XSG; while in FFPE tumor tissues from patients of XHSG, we observed the expression of CD4 and CD8 was significantly increased, compared with XHNSG

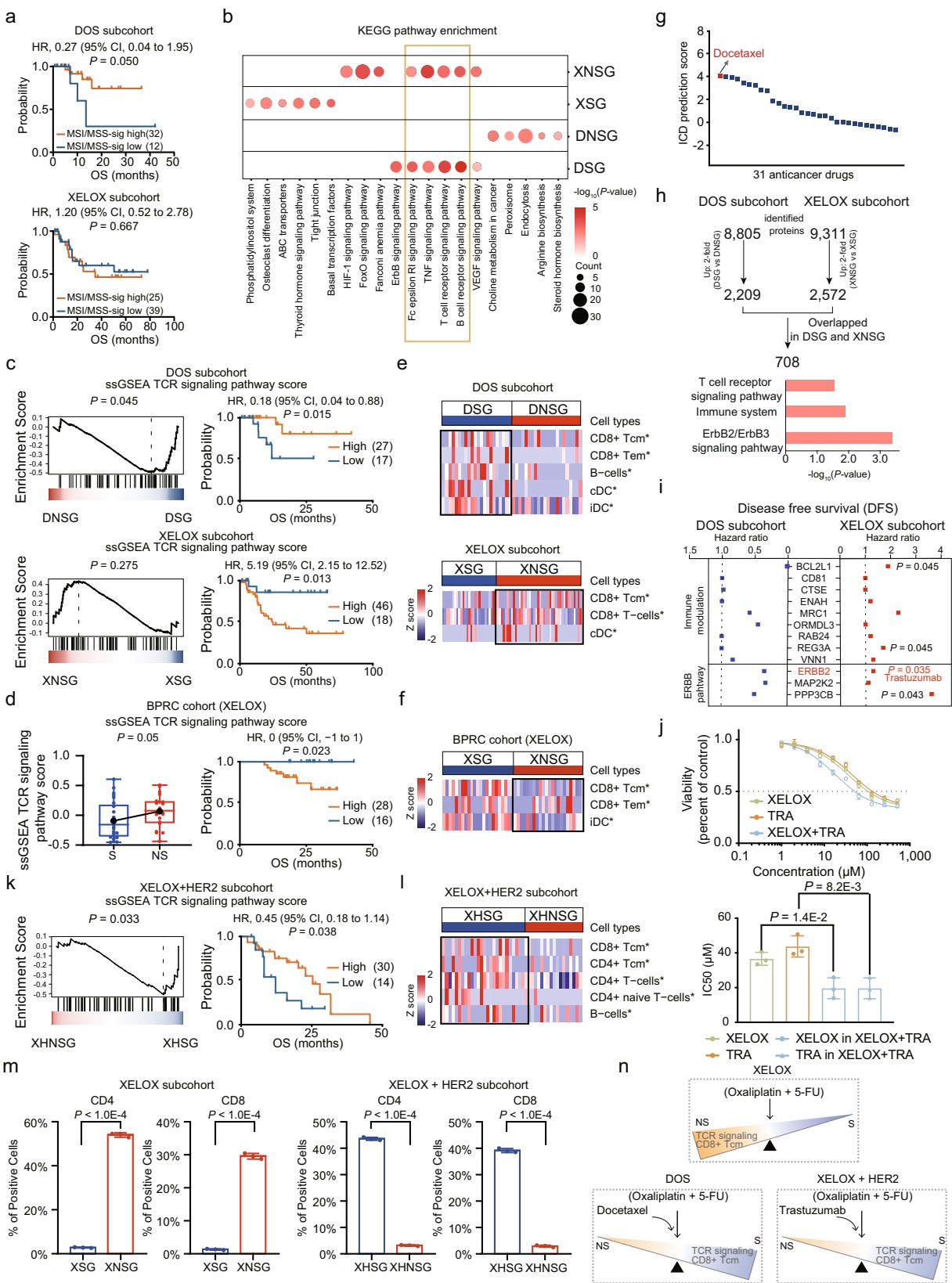

(Supplementary Fig. 5g). Moreover, XHSG had significantly increased percentage of CD4 positive cells (43.7%) and CD8 positive cells (39.2%) than XHNSG (3.2% and 2.9%, respectively) ($P < 1E − 4$); while the percentage of CD4 positive cells and CD8 positive cells were higher in XNSG (54.2% and 29.7%, respectively) compared with XSG (2.9% and 1.4%, respectively) ($P < 0.05$) (Fig. 4m).The result verified the findings

that GC patients with high TCR signaling are unlikely to benefit from XELOX, but instead respond to XELOX+ anti-HER2. Taken together, the combination of trastuzumab with XELOX therapy could resulted in a synergistic antitumor effect, and TCR signaling pathway and CD8 + Tcm emerged as a favorable response marker for the combination therapy (Fig. 4n).

**Fig. 4 | The differential expression of proteins and signaling pathways in sensitive and non-sensitive groups of therapy subcohorts. a** The association of MSI/MSS-sig level with OS in DOS and XELOX subcohorts (two-sided log rank test). **b** Bubble plot showing the KEGG pathway enrichment (two-sided Fisher's exact test) of DSG, DNSG, XSG, and XNSG groups. **c, d, k** The GSEA enrichment analysis of TCR signaling pathway in DOS (**c**), XELOX (**d**, $n$ (S) = 25, $n$ (NS) = 20), XELOX of BPRC cohort (**k**), and XELOX + HER2 subcohorts, respectively (left). Boxplots show median (central line), upper and lower quartiles (box limits), 1.5× interquartile range (whiskers). The survival analysis of ssGSEA TCR signaling pathway score with OS in these subcohorts (two-sided log rank test) (right). **e, f, l** Heatmap illustrating significantly differential cell type compositions between sensitive and non-sensitive groups of these subcohorts (two-sided Student's $t$ test). **g** ICD prediction score of 31 anticancer drugs from NCI library. **h** The overlap of upregulated proteins in DSG and XNSG, and the KEGG pathway enrichment (two-sided Fisher's exact test) of these overlapped proteins. **i** Cox analysis (two-sided Cox test) of the proteins involved in immune modulation and ErbB2/ErbB3 signaling pathway with disease-free survival (DFS). The little boxes indicate the DFS hazard ratios. **j** Dose-response curves of NCI-N87 cells after 72-h treatments with XELOX, trastuzumab, and combination of trastuzumab and XELOX. The comparison of IC50 values of different therapies (two-sided Student's $t$ test). Bars represent the mean of $n$ = 3 independent experiments with error bars indicating SD. **m** The qualification of CD4 and CD8 stained by immunohistochemistry (IHC) in representative examples in the XELOX and XELOX + HER2 subcohorts. Data are analyzed by two-sided Student's $t$ test and shown as mean ± SD ($n$ = 3 independent experiments). **n** Diagram showing the potential connection of immune characteristics and the therapy response. Source data are provided as a Source Data file.

## Activation of ECM/PI3K-AKT pathway related to resistance to anti-HER2 targeted therapy

We then focused on the HER2 subcohort, and separated the subcohort as HER2 sensitive group (HSG) and non-sensitive group (HNSG). There was a significant difference of prognosis in OS ($P$ = 0.039; HR = 0.75; 95% CI, 0.41–1.35) and PFS ($P < 1.0E − 4$; HR = 0.30; 95% CI, 0.17–0.54) between HSG and HNSG as expected (Fig. 5a). We then hypothesized the therapy response of anti-HER2 targeted therapy was related to the expression of HER2. Therefore, we evaluated the expression of HER2 identified in HSG and HNSG by MS, and found no difference between HSG and HNSG. Cox analysis suggested that ERBB2 could not be regarded as an independent prognostic factor in HER2 subcohort. These results reflected the low association of HER2 expression level with trastuzumab resistance (Fig. 5b). For investigating functional characteristics and molecular markers in sensitive or non-sensitive response to anti-HER2 targeted therapy, we performed further comparative analysis of HSG and HNSG. The KEGG pathway enrichment analysis revealed that HSG was featured by TCR signaling pathway, etc.; while HNSG was featured by ECM-receptor interaction, PI3K-AKT pathway, etc. (Fig. 5c, d and Supplementary Data 9). In addition, ECM was negatively correlated with clinical outcome ($P$ = 0.046; HR = 2.6; 95% CI, 1.19–5.67) (Fig. 5e).

To further validate the association of ECM proteins with the response to anti-HER2 targeted therapy, we compared gene expression profiles data from GSE77346 in trastuzumab-sensitive NCI-N87 cell line versus four trastuzumab-resistant cell lines (N87-TR1, N87-TR2, N87-TR3, N87-TR4) by microarray analysis. As a result, we found ECM proteins such as COL4A1, COL6A5, FN1, GP1BA, ITGA4, THBS3, and THBS4 were also overrepresented in trastuzumab-resistant cell lines[47] (Fig. 5f). The PI3K-AKT signaling pathway could be activated by a range of signals, including hormones, growth factors and components of the extracellular matrix (ECM), subsequently regulating cell proliferation and apoptosis. We found PI3K-AKT signaling pathway, showed the highest positive correlation with ECM-receptor interaction pathway (Pearson $r = 0.72$, $P = 2.4E − 12$) (Fig. 5g and Supplementary Fig. 5h). In addition, we observed the subsequent downregulation of apoptosis-related proteins, including BCL2, BCL2L1, CASP3, CASP7, and CDKN2A, in HNSG (Fig. 5h). Based on these findings, we speculated there was a potential synergistic effect of PI3K-AKT inhibition in combination with anti-HER2 therapy.

To validate the hypothesis, we performed further in vitro validation experiment to confirm the synergistic effects of the combination of anti-HER2 and the PI3K-AKT inhibitors. As reported in the previous studies, buparlisib (BKM120) is a commonly used potent, pan-class I PI3K inhibitor approved for clinical trials[48–50]. We treated NCI-N87 cells (HER2-positive cell line) with trastuzumab (TRA), buparlisib (BUP), and combination of trastuzumab and buparlisib (BUP + TRA) with a ratio of 1:1, respectively. We observed the IC50 values of BUP and TRA were 8.19 µM and 46.57 µM in NCI-N87 cells, respectively (Fig. 5i). As indicated in Fig. 5i, the IC50 value of BUP in the combination treatment (BUP + TRA) was significantly decreased to 3.44 µM, compared with single BUP treatment (two-sided Student's $t$ test, $P$ = 3.4E−3). Similarly, the IC50 value of TRA in the combination treatment (BUP + TRA) was also significantly decreased to 3.44 µM, compared with single TRA treatment (two-sided Student's $t$ test, $P < 1.0E-4$). These results suggested BUP could synergize with TRA, resulting an enhanced anti-tumor effect. Taken together, ECM could activate PI3K-AKT pathway and inhibit the apoptosis, thus impairing the anti-tumor effect of trastuzumab (Fig. 5j). Therefore, the PI3K-AKT inhibition combined with anti-HER2 therapy provided a promising therapeutic strategy for HER2-positive GC patients.

Interestingly, we found the negative association of ECM and MSI/MSS-sig, therefore, we wonder whether the response to trastuzumab-based therapy was related to MSI/MSS characteristics. We evaluated the MSI/MSS-sig level in HSG and HNSG, and found GC patients with low MSI/MSS-sig level had poor prognosis, which accounted for 56% of HNSG (higher than HSG (33%)) (log rank test, $P < 0.05$) (Supplementary Fig. 5i). Finally, we proposed the decision-making strategy for the GC therapy. As summarized in Fig. 5k, GC patients with MSS status might not benefit from chemotherapy and targeted therapy; GC patients with MSI status were featured by TCR activation and could benefit from the docetaxel, or trastuzumab + XELOX combined therapies; GC patients with high immune activation showed non-sensitive response to XELOX therapy; GC patients with high ECM level were resistant to anti-HER2 targeted therapy.

## Construction and validation of the predictive models for GC chemotherapy and targeted therapy

Having proposed the decision tree for the personalized chemotherapy and targeted therapy guidance, we next set out to determine whether comparing S-overrepresented and NS-overrepresented proteomes could distinguish sensitive GC patients from non-sensitive GC patients in response to DOS therapy, XELOX therapy, anti-HER2 therapy or combined with chemotherapies (Fig. 6a). We analyzed DSG ($N = 22$) and DNSG ($N = 22$), XSG ($N = 27$) and XNSG ($N = 42$), HSG ($N = 32$) and HNSG ($N = 37$), respectively. We applied Wilcoxon rank-sum tests with a Benjamini-Hochberg (BH) adjusted $P$ value cutoff ($P < 0.05$, BH $P < 0.05$) and found 234, 278, and 194 differentially expressed proteins (DEPs) in the DOS, XELOX, and HER2 subcohorts, respectively (Supplementary Data 6, 7, and 9 and Supplementary Fig. 5j). We employed stepwise logistic regression, which is robust to noise and overfitting, to identify a subset of signatures that accurately discriminates DSG/DNSG, XSG/XNSG, and HSG/HNSG (named as DSG/DNSG-sig, XSG/XNSG-sig, HSG/HNSG-sig). To train and subsequently test the model, samples were partitioned based on sample type (i.e., S or NS) and 80% of samples were used as a training set with the remaining 20% representing the independent testing set. Based on DSG/DNSG-sig ($N = 6$), XSG/XNSG-sig ($N = 14$), HSG/HNSG-sig ($N = 12$) (Supplementary Fig. 6a and Supplementary Data 10), we applied 10-fold cross-validation to the training set yielded three predictive models with high sensitivity (true positive rate) (≥89%) and specificity (true negative rate) (≥88%) in the three subcohorts (Fig. 6b). When applied to the independent test set

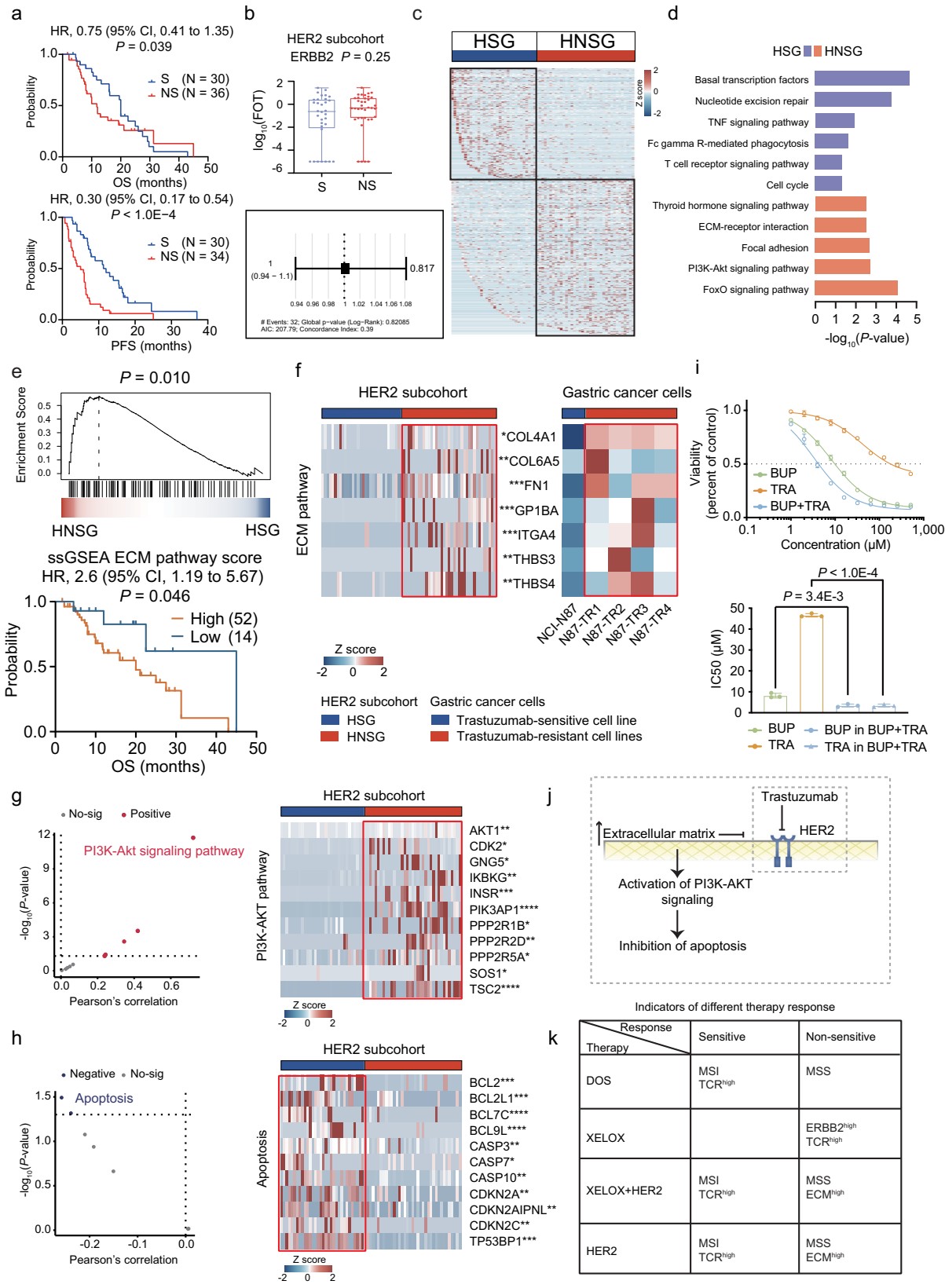

samples, the three predictive models based on DSG/DNSG-sig, XSG/XNSG-sig, HSG/HNSG-sig separately achieved high accuracy of 89%, 93%, and 100% (Fig. 6b).

To evaluate the accuracy of the predictive signatures for the chemotherapeutic response, we designed PRM strategy to quantify these signature proteins in FFPE tumor tissues from the new independent cohort composed of 60 GC patients receiving either DOS ($N = 20$: DSG, $N = 10$; DNSG, $N = 10$), XELOX ($N = 20$: XSG, $N = 10$; XNSG, $N = 10$), or anti-HER2 ($N = 20$: HSG, $N = 10$; HNSG, $N = 10$) therapies (Supplementary Data 11). We selected a set of target peptides that unique to these proteins, including DSG/DNSG-sig (ATP5S, C11orf31, CDC42SE2, CHP2, and AHR), XSG/XNSG-sig (RFC2, NIT1, RAB32, FLG2,

**Fig. 5 | The differential expression of proteins and signaling pathways in HSG/HNSG. a** Kaplan–Meier plots show significant differences between the sensitive group (S) and non-sensitive group (NS) in overall survival (OS) (upper) and progression-free survival (PFS) (bottom) in the HER2 subcohort (two-sided log rank test). **b** Boxplot showing the ERBB2 expression in HSG (n = 32) and HNSG (n = 37) (two-sided Wilcoxon rank-sum test) (upper). Boxplots show median (central line), upper and lower quartiles (box limits), 1.5× interquartile range (whiskers). Cox analysis (two-sided Cox test) of ERBB2 with OS (bottom). **c** Heatmap showing the abundance of differentially expressed proteins in HSG and HNSG. **d** Pathway alterations in HSG and HNSG (two-sided Fisher's exact test). **e** The GSEA enrichment analysis of ECM in HER2 subcohort (Nominal P value, calculated as Phenotype-based permutation test). The survival analysis of ssGSEA ECM pathway score with OS in HER2 subcohort (two-sided log rank test). **f** Heatmap illustrating the abundance of ECM proteins in HER2 subcohort and five gastric cancer cells (two-sided Wilcoxon rank-sum test). **g, h** Left panel: Correlation of ECM pathway score (**g**) and PI3K-AKT pathway score (**h**) with its downstream pathway assessed by ssGSEA (two-sided Pearson's correlation test). Right panel: Heatmap illustrating the protein abundance of PI3K-AKT pathway (**g**) and apoptosis (**h**) related proteins (two-sided Wilcoxon rank-sum test). **i** Dose-response curves of NCI-N87 cells after 72-h treatments with buparlisib (BUP), trastuzumab (TRA), and combination of trastuzumab and buparlisib with a ratio of 1:1. IC50, half-maximal inhibitory concentration. The comparison of IC50 values of different therapies (two-sided Student's t test). Bars represent the mean of n = 3 independent experiments with error bars indicating SD. **j** Diagram showing the potential mechanism of resistance to anti-HER2 targeted therapy. **k** The decision-making strategy for the GC therapy. *P < 0.05 is considered statistically significant. *P < 0.05, **P < 0.01, ***P < 0.001, ****P < 0.0001. Source data are provided as a Source Data file.

FNBP1, GCLC, DYNLRB1, RBBP7, LPXN, LMAN2, NUB1, WAS, FAM82B, and MYCBP), and HSG/HNSG-sig (CAPN5, BAIAP2, SRPX2, COMMD4, SCIN, DSC2, SEPSECS, TECPR1, DDX60L, NPL, SLC39A4, and IRF6) using the library search results (Supplementary Data 11). Based on the PRM quantification, we performed comparative analysis of signature proteins (including DSG/DNSG-sig, XSG/XNSG-sig, and HSG/HNSG-sig) between DSG and DNSG, XSG and XNSG, HSG and HNSG, respectively. As a result, we observed the significantly differential expression of these signature proteins between DSG and DNSG, XSG and XNSG, HSG and HNSG, respectively. In DOS subcohort, we observed signature proteins, including AHR, ATP5S, C11orf31, CDC42SE2, and CHP2, had at least twofold differences between DSG and DNSG (P < 0.05, Wilcoxon rank-sum test). In XELOX subcohort, XSG/XNSG-sig (RFC2, NIT1, RAB32, FLG2, FNBP1, GCLC, DYNLRB1, RBBP7, LPXN, LMAN2, NUB1, WAS, FAM82B, and MYCBP) were significantly increased in XNSG compared with XSG (Fold change (XNSG/XSG) > 2, P < 0.05, Wilcoxon rank-sum test). In HER2 subcohort, we observed HSG/HNSG-sig (CAPN5, BAIAP2, SRPX2, COMMD4, SCIN, DSC2, SEPSECS, TECPR1, DDX60L, NPL, SLC39A4, and IRF6) had a more than twofold increase in HNSG compared with HSG (Fold change (HNSG/HSG): 2.17, 2.22, 4.44, 5.43, 5.71, 4.60, 5.27, 5.03, 4.06, 2.18, 5.40, 3.35; P < 0.05, Wilcoxon rank-sum test) (Supplementary Fig. 6b–d). In addition, the heatmaps showed a clear separation between DSG and DNSG, XSG and XNSG, HSG and HNSG in the new independent cohort, respectively (Fig. 6c). Collectively, predictive power of the signature proteins in different therapies (including DOS, XELOX, and anti-HER2) was validated in an independent cohort by PRM assays.

Furthermore, XSG/XNSG-sig model was also validated in the external clinically annotated BPRC DGC cohort which is accessible in the PRIDE Archive under the accession number PXD008840[27]. Here, 45 patients with GC received XELOX chemotherapy with long-term follow-up. In the BPRC DGC cohort, chemo-non-sensitive group (N = 20, median disease-free survival (mDFS) = 339.5 days) and chemo-sensitive group (N = 25, mDFS = 695.0 days) were defined (Supplementary Fig. 6e), and showed strong correlations with prognosis (log rank test, P < 0.05). The XSG/XNSG-sig revealed in our study was then applied to predict the therapeutic response of XELOX in the BPRC cohort. Significantly, our predictive model resulted in a high sensitivity and specificity of prediction with an AUC of 0.95 in the BPRC DGC cohort (Supplementary Fig. 6e), demonstrating the robustness, accuracy, and stability of the prediction model. Taken together, the accuracy of these models for predicting response of different therapies were verified by multi-center GC cohorts based on the proteomic data.

## Overexpression of CTSE synergistically enhances sensitivity to docetaxel by stabilizing microtubules

Based on the proteomic subtyping analysis, we found that intracellular proteinase such as cathepsin E (CTSE) overrepresented in G-III subtype featured with response to DOS but not XELOX therapy; the result was further validated in the following differential analysis of DOS and XELOX subcohorts. Taken together, the expression level of CTSE positively correlated with the patient chemosensitivity to DOS (Supplementary Fig. 7a).

We first investigated how CTSE levels modulated the response of tumor cells to DOS and XELOX, in cultured MKN45 and MGC803 GC cell lines. Compared with the cells transfected with an empty vector, CTSE overexpression significantly promoted the proliferation of MKN45 and MGC803 cells (Student's t test, P < 0.01) (Supplementary Fig. 7b and Fig. 7a, b), consistent with previous reports[51,52], and excluded the negative effect of CTSE overexpression on cell proliferation. Next, we conducted similar experiments using clinical combined patterns of DOS (docetaxel: oxaliplatin: 5-fluorouracil = 1:1:10) and XELOX (oxaliplatin: 5-fluorouracil = 1:7.7) to treat MKN45 and MGC803 cells[19]. The degree of DOS and XELOX sensitivities in the MKN45 and MGC803 cells were estimated by their half-maximal inhibitory concentration (IC50) values[53,54]. We treated CTSE-overexpressing and empty vector-overexpressing MKN45 and MGC803 cells with DOS and XELOX therapies, respectively; consistent results in the two GC cell lines were observed. The IC50 values of DOS were significantly lower (0.37-fold and 0.53-fold decrease in MKN45 and MGC803, respectively; Student's t test, P < 0.01) in the CTSE-overexpressing groups (IC50, 1.47 and 1.26 nM, respectively) compared with the control groups (IC50, 2.32 and 2.69 nM, respectively) (Fig. 7c), whereas there was no obvious change for XELOX (1.23 and 0.83 fold change, respectively; P > 0.05) (Fig. 7d). Taking into consideration the differences between DOS and XELOX, we further treated each CTSE-overexpressing and empty vector-overexpressing MKN45 and MGC803 cells with docetaxel (DOC), 5-fluorouracil (5-FU), and oxaliplatin (OXA). The CTSE-overexpressing groups were shown to have lower IC50 values for DOC than the control groups (0.31-fold and 0.38-fold decrease in MKN45 and MGC803 cells, respectively; Student's t test, P < 0.05) (Fig. 7e). This, together with the findings that the IC50s of 5-FU (0.91 and 1.03 fold change, respectively; P > 0.05) and OXA (1.05 and 0.84 fold change, respectively; P > 0.05) did not change (Supplementary Fig. 7c, d), suggested that CTSE overexpression is an important determinant of DOC, but not 5-FU or OXA, sensitivity.

To elucidate the molecular mechanism by which CTSE overexpression enhanced the cytotoxic effects of DOC, we performed proteomic analysis using a label-free technique and compared the proteome between the CTSE-overexpressing and control MKN45 cells treated with or without DOC. We performed three repeats in the CTSE-overexpressing and control MKN45 cells, and made a uniformed quality control, which resulted in the identification of 5371 and 5735 GPs, respectively, in MKN45 cells with and without DOC treatment at a 1% global protein FDR (Supplementary Data 12). There were no major differences in the coverage between the CTSE-overexpressing and control MKN45 cells with or without DOC treatment (Supplementary Fig. 7e–g). Differential proteomic analysis of the CTSE-overexpressing and control MKN45 cells without treatment showed that 971 upregulated GPs were enriched in basal transcription factors (P = 4.36E − 3)

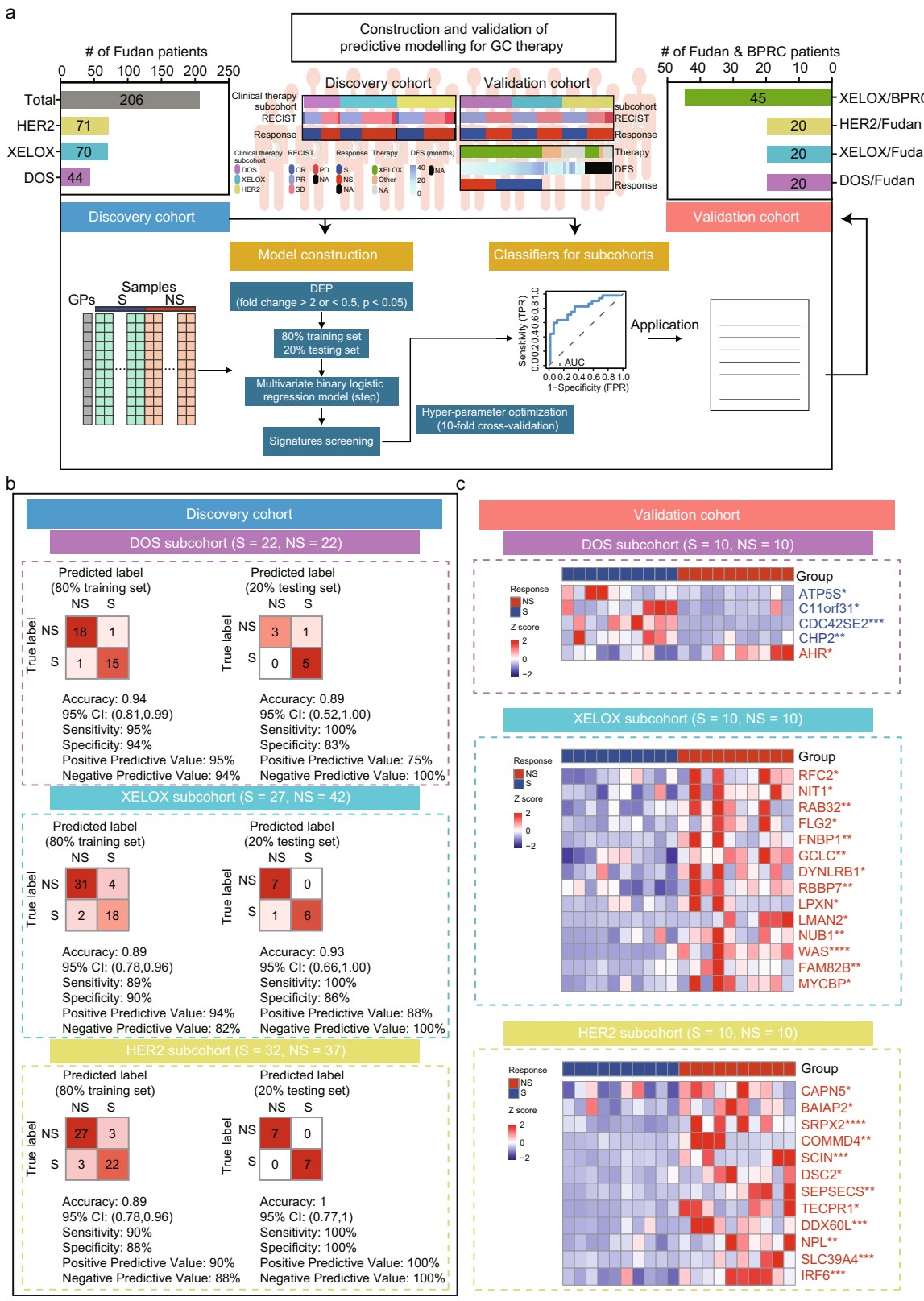

**Fig. 6 | The construction and validation of predictive classifiers. a** Diagram describing a construction and validation of the predictive modeling for sensitive (S) and non-sensitive (NS) groups. **b** The discovery cohort: Classification error matrix using logistic regression classifier of 80% training set and 20% testing set in DOS, XELOX, and HER2 subcohorts. The number of samples identified is noted in each box. **c** Heatmap showing differential expression of DSG/DNSG-sig, XSG/XNSG-sig, and HSG/HNSG-sig based on PRM quantification in an independent cohort (validation cohort) composed of 60 GC patients (two-sided Wilcoxon rank-sum test). Source data are provided as a Source Data file.

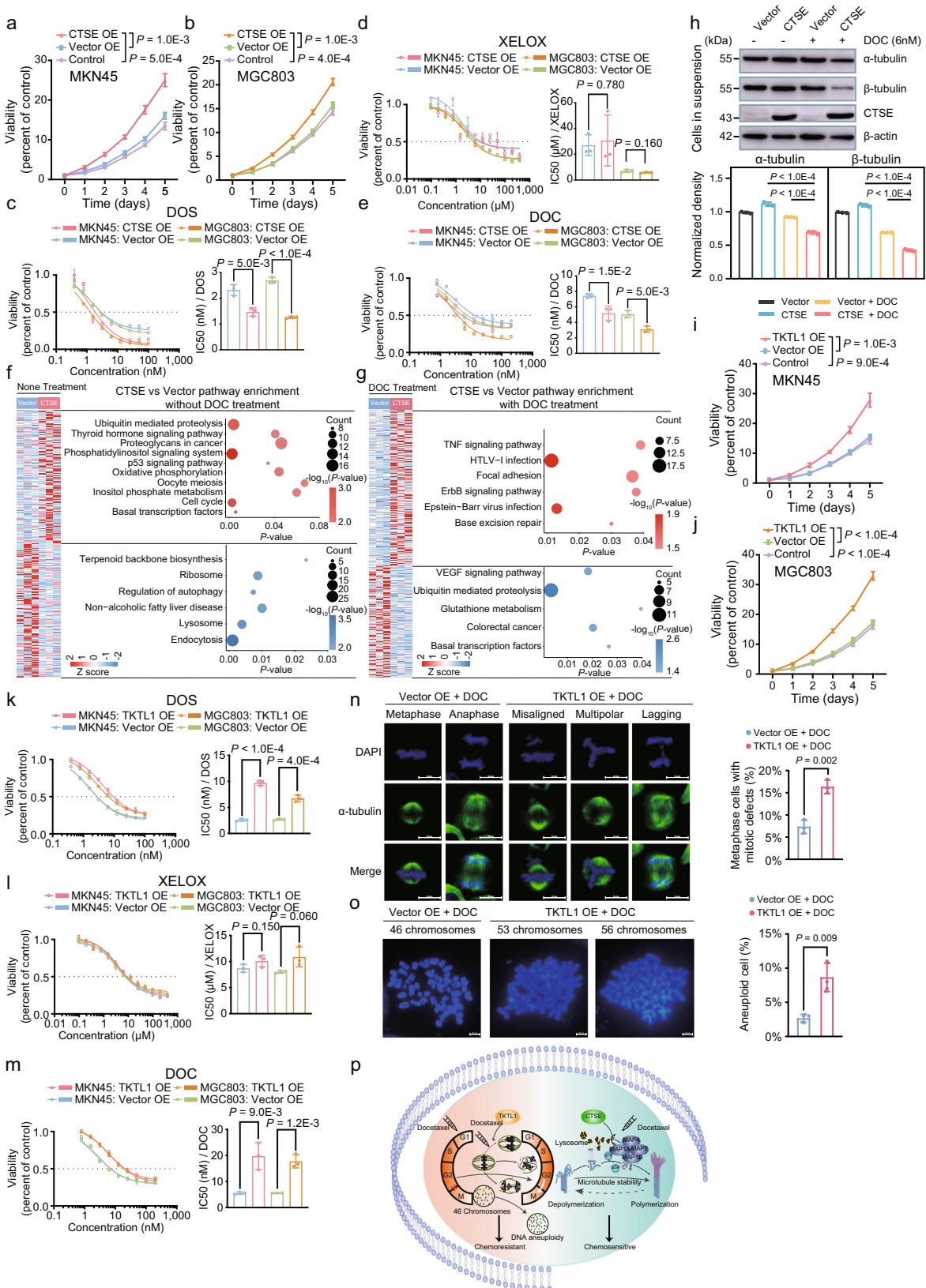

(including TAF2, TAF3, TAF4, TAF13, GTF2H4, etc.), ubiquitin-mediated proteolysis ($P = 4.45E-3$) (including UBE2C, UBE2G1, UBE2G2, CUL2, PIAS2, etc.), and p53 signaling pathway ($P = 3.52E-2$) (including CCNB1, CASP3, TSC2, CDK6, IGFBP3, etc.), (Fig. 7f and Supplementary Data 12). The 969 downregulated GPs were mainly enriched processes involving endocytosis ($P = 2.29E-4$) (including

CHMP2A, CLTA, WASH1, SNX2, SNX6, etc.), lysosome ($P = 3.37E-3$) (including CTSB, CTSC, GUSB, ATP6V0C, LAMP1, etc.), regulation of autophagy ($P = 7.76E-3$) (including GABARAPL1, GABARAPL2, ATG5, ATG7, BECN1, etc.), and ribosome ($P = 9.47E-3$) (including MRPL1, MRPL30, RPL17, MRPS11, RPLP1, etc.) (Fig. 7f and Supplementary Data 12), which potentiated CTSE's promotion of tumor growth.

**Fig. 7 | Effect of CTSE and TKTL1 overexpression on anticancer activity of DOS and XELOX in gastric cancer cell lines MKN45 and MGC803. a, b** Effect of CTSE overexpression on proliferation in gastric cancer (GC) cell lines (MKN45 and MGC803, n = 3 independent experiments, two-sided Student's t test, mean ± SD). **c–e** Dose-response curves of MKN45 and MGC803 cell lines overexpressing CTSE after 72-h treatments with DOS, XELOX, and DOC. IC50, half-maximal inhibitory concentration. *P-values are calculated using two-sided Student's t test. Barplots showing the comparison of IC50 values in each group. **f, g** KEGG pathway analysis showing the differential function in the CTSE group with or without DOC treatment (two-sided Fisher's exact test). **h** Immunoblot analysis of soluble alpha-tubulin and soluble beta-tubulin in MKN45 cells after DOC treatment, and the normalization of qualified western blots (n = 3 independent experiments, two-sided Student's t test, mean ± SD). **i, j** Effect of TKTL1 overexpression on MKN45 and MGC803 cell proliferation (n = 3 independent experiments, two-sided Student's t test, mean ± SD).

**k–m** Dose-response curves of MKN45 and MGC803 cell lines overexpressing TKTL1 after 72-h treatments with DOS, XELOX, and DOC (n = 3 independent experiments, two-sided Student's t test, mean ± SD). IC50, half-maximal inhibitory concentration. Barplots showing the comparison of IC50 values in each group. **n, o** Chromosome segregation defects (**n**) and percentage of aneuploid cells (**o**) in DOC-treated MKN45 cells with or without TKTL1 overexpression. The left panel shows the representative results. The scale bar indicates 10 µm. The right panel shows the statistical results from n = 3 independent experiments in the MKN45 cells (two-sided Student's t test, mean ± SD). **p** Schematic illustration indicating how CTSE inducts microtubule stabilization, and how TKTL1 induces abnormal chromosome segregation. Bars represent the mean of n = 3 independent experiments with error bars indicating SD (for **c–e**, **h**, **k–m**, **n–o**). Source data are provided as a Source Data file.

Further differential proteomic analysis suggested that CTSE overexpression rendered MKN45 cells prone to necrosis with DOC treatment [e.g., the upregulation of the TNF pathway ($P = 3.96E-2$) (including MAP3K7, AKT1, CEBPB, JUN, NFKB1, etc.) and the downregulation of VEGF ($P = 1.90E-2$) (including MAPK3, MAPK14, PIK3CA, PXN, PIK3R2, etc.) and basal transcription factors ($P = 2.68E-2$) (including TAF8, TAF10, MNAT1, GTF2H5, GTF2B, etc.)], which suggested that CTSE has a positive response to DOC (Fig. 7g and Supplementary Data 13). Overexpression of CTSE did not alter the protein expression levels of MAP4, and TUBB3 (Wilcoxon rank-sum test, $P > 0.05$) (Supplementary Fig. 7h). Interestingly, overexpression of CTSE and DOC treatment synergistically reduced the expression levels of the proteins involved in the microtubule assembly bioprocesses [MAP4 (fold change = 0.15), and TUBB3 (fold change = 0.46) (Wilcoxon rank-sum test, $P < 0.05$)] (Supplementary Fig. 7h). According to previous research, microtubule associated protein share the same binding site to microtubule as paclitaxel, and the increase of soluble intracellular tubulin is an indicator of microtubule stability[55,56]. Therefore, we speculated that the low expression of microtubule-associated proteins rendered the microtubules vulnerable to DOC and made the GC cell lines hypersensitive. We also performed immunoblot analysis to verify that DOC exposure in the CTSE group reduced the content of both soluble α- and β-tubulin subunits (Student's t test, $P < 0.001$), indicating the enhanced polymerization of tubulin into microtubules to increase DOC sensitivity (Fig. 7h). The immunohistochemical staining showed significant differences in the CTSE expression between the sensitive and non-sensitive groups in patients treated with DOS and XELOX (Student's t test, $P < 0.0001$) (Supplementary Fig. 7i–l), which verified the aforementioned observations that CTSE levels positively responded to DOS treatment.

### Overexpression of TKTL1 decreases the sensitivity to docetaxel by inducing abnormal chromosome segregation

Transketolase-like-1 (TKTL1), a rate-limiting enzyme in the non-oxidative part of the pentose-phosphate pathway, has been demonstrated to promote carcinogenesis and cell proliferation[57]. In contrast to CTSE, the levels of TKTL1 negatively corresponded to DOS therapy (Supplementary Fig. 7a). A 6.9-fold increase of TKTL1 expression was observed in the DNSG, but not DSG, suggesting that TKTL1 led to a decreased sensitivity to DOS. Consistent with previous findings[58], we confirmed that the overexpression of TKTL1 promoted cell proliferation in both MKN45 and MGC803 cell lines (Student's t test, $P < 0.01$) (Supplementary Fig. 8a and Fig. 7i, j). By comparing the sensitivities of TKTL1-overexpressing and control cells to DOS and XELOX, we showed that the IC50 values of DOS were significantly higher (3.76-fold and 2.50-fold increase in MKN45 and MGC803, respectively; Student's t test, $P < 0.001$) in the TKTL1-overexpressing cells (IC50, 9.71 nM and 6.73 nM, respectively) compared with the control groups (IC50, 2.59 nM and 2.69 nM, respectively), whereas, no significant change in XELOX was observed (1.16 and 1.35 fold change respectively, $P > 0.05$)

(Fig. 7k, l). Taking into consideration the differences between DOS and XELOX, we further treated each TKTL1-overexpressing and empty vector-overexpressing MKN45 and MGC803 cells with DOC, 5-FU, and OXA, and observed that the TKTL1-overexpressing groups had higher IC50 values for DOC than the control groups (3.57-fold and 3.18-fold increase, respectively; Student's t test, $P < 0.01$) (Fig. 7m). This, together with findings that the IC50s of 5-FU (1.16 and 1.24 fold change, respectively; $P > 0.05$) and OXA (0.95 and 0.98 fold change, respectively; $P > 0.05$) did not change (Supplementary Fig. 8b, c), suggested that TKTL1-overexpressing cells were resistant to DOC. DOC is reported to cause cell cycle arrest in mitosis, and increases the apoptosis of cancer cells[59]. Here, we confirmed that DOC induced G2/M arrest and increased the apoptosis in MKN45 cell lines; however, the overexpression of TKTL1 alleviated the G2/M arrest and reduced cell apoptosis following treatment with DOC (Student's t test, $P < 0.01$) (Supplementary Fig. 8d–g). Considering that increased TKTL1 promoted the release of a DOC-mediated mitosis block, we examined the chromosome status in TKTL1-overexpressing and control cells. Severe abnormal chromosome segregation phenotypes were observed in the DOC-treated TKTL1-overexpressing cells, including misaligned, multipolar, and lagging chromosomes (Student's t test, $P < 0.05$) (Fig. 7n). Moreover, increased aneuploidy was observed in DOC-treated TKTL1-overexpressing cells, compared with DOC-treated control cells ($P < 0.05$) (Fig. 7o). The immunohistochemical staining showed significant differences in TKTL1 expression between the sensitive and non-sensitive groups in patients treated with XELOX or DOS (Student's t test, $P < 0.0001$) (Supplementary Fig. 8h–k), which agreed with the findings above that TKTL1 positively and negatively responds to XELOX and DOS, respectively. Taken together, these results indicated that TKTL1 promoted DOC-treated cell survival by causing abnormal chromosome segregation and DNA aneuploidy. Finally, we proposed a model in which CTSE functions as a cell intrinsic enhancer of the chemosensitivity of DOC via microtubule stabilizing effects, while TKTL1 functions as an attenuator by inducing abnormal chromosome segregation (Fig. 7p).

## Discussion

GC is currently the fourth most common malignancy, and the second leading cause of cancer-related deaths worldwide. As it is most often diagnosed at an advanced stage, the preferred treatment for advanced GC is surgery[60,61]. However, for patients with no chance of surgical treatment, the ultimate goal of chemotherapy is to prolong survival and improve quality of life[62]. Recently, significant progress has been made by the iterative updates of first-line chemotherapeutic drugs[19,63], and the integration of HER2-targeting drugs[11,64]. Meanwhile, significant challenges have been added as there are no indicators for their chemotherapeutic effectiveness, and the development of drug resistance remains unresolved. Therefore, it is imperative to make a preclinical diagnosis of tumor response to the first-line therapies XELOX and DOS. Herein, we collected FFPE tissues of 206 patients with GC before the

initiation of different treatments (including DOS, XELOX, and anti-HER2-based therapy), and presented an unprecedented large-scale clinical proteomic landscape, with the aim of identifying reliable predictive markers of the GC patient response to the diverse chemotherapies and targeted therapies.

In the study, the proteomic subtypes were identified with distinct molecular features and clinical outcomes, and associated with therapy subcohorts. Further analysis revealed that the significant distribution difference of therapy subcohorts among proteomic subtypes were mainly derived from HER2 subcohort, due to the selection of HER2-positive patients for anti-HER2 targeted therapy. In addition, HER2 expression detected by IHC and FISH analysis demonstrated the specific molecular pattern in HER2-positive GC patients, supporting the reliability of proteomic subtyping. We also surveyed the robust quantification of the signature proteins identified in four proteomic subtypes. We found 1679 signature proteins (74% of 2279 signature proteins) have immunohistochemistry (IHC) staining data as reported by The Human Protein Atlas (HPA), among which 1467 (87.4%) signature proteins showed medium to high tumor-specific staining in GC samples, demonstrating these signature proteins could be reused and validated in the future research community (Supplementary Fig. 3a). Pathway enrichment analysis of the proteomic subtypes suggested that activation of endocytosis indicated drug sensitivity, while high expression of ECM associated with drug resistance. These results suggested endocytosis activation or ECM inhibition could be applied in the combination of chemotherapy and targeted therapy, thus improving therapy sensitivity and alleviating therapy resistance. Further analysis revealed high expression of THSD4 was significantly associated with poor prognosis and highly correlated with responses of drugs (including 5-FU, oxaliplatin, and docetaxel) in gastric cancer cell lines data from the DepMap. We also validated THSD4 overexpression reduced the anti-tumor effect of chemotherapeutic drugs in vitro. The association between ECM proteins and drug resistance was validated by PRM approach in an independent cohort, indicating these extracellular matrix proteins could serve as indicators to predict chemotherapy response in clinic.

In addition, the proteomic subtyping system have been applied in other gastric cancer cohorts (BPRC cohort, ACRG cohort, and EOGC cohort). The results demonstrated there was a significant concordance between FDGC subtype and EOGC subtype or ACRG subtype, but not between FDGC subtype and BPRC subtype. After our examination of the composition of clinical characteristics, we speculated the composition of Lauren's type, not other clinical characteristics, was the possible reason for no concordance between the FDGC and the BPRC subtypes. Based on the unique features of our proteomic subtypes associated with therapeutic response, we could assess the possibility of response to specific therapy in other cohorts (although there was no exact therapy response). Overall, the result demonstrated the robustness of FDGC subtyping and their consistent and significant association with clinical outcomes despite the various sources of heterogeneity and cohort differences. Importantly, the FDGC subtyping in ACRG cohort provided a clue that GC patients with MSS status had an association with drug resistance and poor survival. We further explored the association of immune microenvironment and MSI/MSS characteristics among four proteomic subtypes. The results showed G-IV had the lowest MSI/MSS-sig level, and was featured by less cytokines and antigen processing and presentation, which made more monocytes aggregated and less macrophages M1 differentiated.

Whether MSI/MSS characteristics could be a predictive marker for response to chemotherapy and targeted therapy remain not clearly reported. Even though XELOX and DOS are two major first-line treatments[19], there is no indication for patients to choose the appropriate treatment strategy. Surprisingly, we found MSS characteristics served as a chemo-resistance indicator and poor prognosis marker in the DOS subcohort, but not in the XELOX subcohort. Furthermore, we

conducted a thorough bioinformatic analysis of the sensitive and non-sensitive patient groups to elucidate the potential mechanisms of drug response, thereby aiding the prediction of sensitivity to chemotherapy and the reversal of drug resistance to improve therapeutic efficacy. Interestingly, our results showed that patients with GC with predominant immune pathways (BCR, TCR, and FcεRI pathways) were inclined to be sensitive to DOS therapy while resistant to XELOX therapy. Among these pathways, TCR signaling pathway showed obvious association with drug response and clinical prognosis. The xCell analysis also revealed that immune cells such as CD8 + Tcm enriched in DSG and XNSG, indicating the distinct immunostimulatory effect in response to DOS and XELOX therapies. The ranked immunogenic cell death (ICD) prediction score of chemotherapeutic regiments showed docetaxel was the top one ICD drug. Surprisingly, we found ERBB pathway was enriched in XNSG and DSG, and univariable and multivariable cox analysis validated ERBB2 was a predictive prognostic marker adjusted for baseline clinical covariates in XELOX subcohort, indicating the clinical implication of XELOX combined with anti-HER2 targeted therapy. The synergistic effects of the combination of anti-HER2 and XELOX therapy was confirmed by the in vitro validation experiment. Further differential analysis focused on the combination therapy and revealed the activation of TCR signaling pathway was associated with sensitive response to XELOX combined with anti-HER2 targeted therapy. These results indicated that TCR signaling pathway exerts diverse effects in response to DOS, XELOX, as well as XELOX combined with anti-HER2 targeted therapy, which assist to determine the appropriate chemotherapy and targeted therapeutic strategy for GC patients.

Clinically, the usage of the anti-HER2 targeted therapy was determined by the expression of HER2 detected by FISH (fluorescence in situ hybridization) and IHC (Immunohistochemistry). Although trastuzumab-based first-line treatments represent the standard approach for HER2-positive GC, not all HER2-positive patients with GC benefit from this treatment, and show variable ORR (-32–68%). Researchers have explored various approaches, such as circulating tumor DNA detection and gene expression analyses, to elucidate the possible mechanisms involved in trastuzumab resistance[65,66]. In this study, consensus clustering analysis identified four proteomic subtypes (G-I to G-IV), among which G-II subtype had the highest proportion of patients receiving anti-HER2 therapy, due to the selection of HER2-positive patients for anti-HER2 targeted therapy, indicating the specific proteome panel of HER2-positive patients. However, HSG and HNSG accounted nearly half in G-II subtype, suggesting glycolysis/gluconeogenesis and pantothenate/CoA biosynthesis (enriched in G-II subtype) were unrelated to anti-HER2 therapy response. On the contrary, G-IV subtype, featured by ECM pathway, had the highest proportion of non-sensitive patients of anti-HER2 therapy. Consistently, further comparative analysis of between sensitive and non-sensitive patients with anti-HER2 targeted therapy also revealed ECM-receptor interaction pathway were enriched in non-sensitive group, and negatively correlated with clinical outcome. Surprisingly, we found the low association of HER2 expression level with trastuzumab resistance. Importantly, apart from steric hindrance to anti-HER2 antibody binding through ECM components in tumor, we found a potential mechanism related to trastuzumab resistance that ECM could activate PI3K-AKT signaling pathway and inhibit the subsequent apoptosis signaling, thus impairing the anti-tumor effect of trastuzumab. Further experiments validated that PI3K-AKT inhibitor buparlisib (BKM120) could synergize with trastuzumab, resulting in an enhanced anti-tumor effect, and provided a promising therapeutic strategy for HER2-positive GC patients.

Current clinically used tumor markers for GC screening have achieved much progress, including HER2[67], human epidermal growth factor receptor (EGFR)[68], mammalian target of rapamycin (mTOR)[69], PD-L1[70,71], and TP53[22]. However, predictive markers for specific

therapeutic strategies in GC are still lacking. The identification of effective biomarkers for clinical diagnosis, prognosis, prediction, and therapy is a continuous effort. For clinical guidance in the response to therapy, in this study, we employed proteomic technology implemented with machine learning statistics to search for drug sensitive and non-sensitive predictive biomarkers in GC as an alternative to clinical therapeutic strategies. Statistical analysis was performed on the discovery set to identify potential chemosensitivity mediators for GC. Furthermore, these signature proteins identified in predictive models for DOS, XELOX, and HER2 subcohorts, were validated by parallel reaction monitoring (PRM) quantification with 100% expression frequency, in the new independent cohort, exhibiting a well distinguish between sensitive patients and non-sensitive patients and indicating the reliability of these results. In addition, the good diagnostic performance of prediction model for XELOX therapy was also confirmed by an external validation (BPRC DGC cohort, AUC = 0.95). Besides proteomic quantification, the expression of these signature proteins was also validated by the IHC staining data reported by HPA, (Supplementary Fig. 6f). Taken together, our study provides a panoramic view of the clinical proteomic landscape of GC which can be specifically coupled to the therapeutic response. Multicenter validation of predictive signatures demonstrated the robustness and accuracy of this prediction model; PRM quantification and HPA IHC staining data validated that these signature proteins in models could be reused and validated in the future research community.

Based on the proteomic analysis, we concluded that CTSE expression levels were sensitive to DOS therapy but non-sensitive to XELOX therapy. As recently reported, cathepsins have emerged as an important class of proteolytic enzymes in cancer development, and cysteine cathepsin inhibitors have been proposed as anticancer agents[72–75]. Data presented in this study confirmed the potential for CTSE to promote GC growth. More importantly, we further demonstrated that CTSE could enhance the cytotoxic response of GC cells to docetaxel, suggesting it as a candidate marker of the drug response. Differential proteomic analysis further revealed the downregulation of microtubule-associated proteins (MAP4 and MAP7), suggesting a potential chemoresistance mechanism by which CTSE overexpression can affect microtubule dynamics. CTSE is widely distributed in digestive cancers, such as esophageal cancer, GC, colorectal cancer, pancreatic cancer, and liver cancer. We speculate that CTSE also has an enhanced antitumor effect on DOC-based chemotherapy in other digestive cancers.

In addition, increased TKTL1 expression[76,77] attenuates the chemosensitivity of patients to DOS therapy, which is completely opposite to the response to DOC-based chemotherapy observed with CTSE. Our previous study revealed that TKTL1 gathered signals from the nutrition and cell cycles, and promoted cell cycle proceeding[78]. In the current study, we further found that high levels of TKTL1 promoted the escape of tumor cells from apoptosis induced by DOC, mediated by abnormal chromosome segregation and DNA aneuploidy. As a result, TKTL1 promoted DOC-treated cells' survival. Taken together, we propose that CTSE functions as a cell intrinsic enhancer of chemosensitivity to DOC via microtubule stabilizing effects, whereas TKTL1 functions as an attenuator by inducing abnormal chromosome segregation.

In summary, our proteomic data described an atlas of chemotherapy and targeted therapy in GC, covering three first-line therapy subcohorts: the DOS, XELOX, and HER2 subcohorts. This study identified the proteomic subtyping correlated with clinical outcomes (overall survival and therapy response), the performance of which was further evaluated in multi-center GC cohorts. Importantly, we provided indicators for these chemotherapy and targeted therapy, including MSI/MSS, ERBB2 expression, the activation of ECM and TCR signals, to help develop rational treatment options for GC patients in clinic, thus providing them more effective strategy. Finally, we extracted S-overrepresented and NS-overrepresented proteomes and constructed predictive models, which could well distinguish sensitive GC patients from non-sensitive GC patients in response to DOS therapy, XELOX therapy, and anti-HER2 therapy. The construction and validation of proteomic-based predictive classifiers for these first-line therapies of GC contributed to personalized chemotherapy and targeted therapy, to a certain extent; thereby, moving us toward the era of proteomic-driven precision medicine. More importantly, we propose that CTSE coupled with TKTL1 can facilitate a more effective clinical decisions to determine the relative benefit of DOC-based therapy for patients. In summary, our study has potentially important clinical implications in GC, and we look forward to further developing potential therapeutic combinations for DOC-based therapy.

## Methods

### Clinical sample acquisition

This study was approved by the Research Ethics Committee of Zhongshan Hospital (B2019-200R). Written informed consent was received from all patients included in this study.

Archival formalin-fixed, paraffin-embedded (FFPE) tissues from patients with GC, from January 2002 to March 2020, were reviewed in the Department of Pathology, Zhongshan Hospital, Fudan University (Shanghai, R. P. China). The discovery cohort consisted of the DOS subcohort (44 cases treated with S-1 and oxaliplatin combined with docetaxel), the XELOX subcohort (70 cases treated with capecitabine and oxaliplatin), and the HER2 subcohort (71 cases treated with the anti-HER2-based therapy), according to the treatment profiles. Another 21 cases were assigned as "Others," of which 3 cases received apatinib or docetaxel therapies, and 18 cases had no chemotherapy information. The independent validation cohort for PRM verification included 60 patients with GC, receiving either DOS ($N$ = 20), XELOX ($N$ = 20), or anti-HER2 ($N$ = 20) therapies. All the chemotherapy regimens were given at standard dosing as described in previous studies. Briefly, the XELOX regimen was administered as follows: capecitabine (1000 mg/m², twice daily on days 1–14) and oxaliplatin (130 mg/m² on day 1)[28]. The DOS regimen was provided as S-1 (tegafur, gimeracil and oteracil porassium capsules; 40 mg/m² orally administered twice a day on days 1–14), oxaliplatin (100 mg/m² on day 1), and docetaxel (40 mg/m² on day 1). The two regimens were repeated every 3 weeks[79]. In the HER2 subcohort, patients generally received the anti-HER2-based therapy either as a XELOX combined anti-HER2 therapy or other chemotherapies combined anti-HER2 therapy, as reported in the previous two phase II trials[8,11]. All the patients of the HER2 subcohort received trastuzumab with a dose of 6 mg/kg every 3 weeks after a first infusion of 8 mg/kg[80]. Medical records were reviewed to obtain the follow-up data. The treatment response was evaluated by CT/MRI scanning following the Response Evaluation Criteria in Solid Tumors (RECIST) (version1.1)[81]. Tumor response was assessed and categorized as a complete response (CR), partial response (PR), stable disease (SD), or progressive disease (PD). In clinic, the ORR, generally defined by the Food and Drug Administration as the sum of PR plus CR, is a direct measurement of drug antitumor activity. Here, the ORR, defined as PR plus CR, was selected for the efficacy evaluation; patients with CR and PR were defined as sensitive (S) and those with SD and PD were defined as non-sensitive (NS). Major clinical parameters were collected including age, gender, grade, Lauren's type, primary site, chemotherapy and targeted therapy, therapy cycle (3 weeks per cycle), RECIST, TNM stage (according to the eighth edition of the American Joint Committee on Cancer staging system), status of cancer recurrence or progression, and status of survival. Archival FFPE tissues with at least 80% tumor purity of GC cases were taken from chemotherapy naive patients. The detailed clinical information of each patient is included in Supplementary Data 1.

## HER2 evaluation

Immunohistochemistry (IHC) and in selected cases fluorescence in situ hybridization (FISH) diagnostics have been implemented to confirm the status of HER2 expression. For all patients, HER2 expression was detected by IHC. IHC staining was carried out using the rabbit monoclonal anti-HER-2/NEU (4B5) antibody (working solution with the concentration of 6 μg/mL for IHC, Ventana Medical Systems, Inc. Tucson, AZ, USA) as the primary antibody against HER2 on a Ventana Benchmark XT automatic staining system, according to the manufacturer's instructions. The amended HER2 IHC scoring system for gastric cancer was used as the criteria for scoring the stained slides[82]. For scoring the IHC image, Histoscore (H-score) was calculated by multiplying the proportion of positive cells in the sample (0–100%) by the average intensity of the positive staining (1+, 2+, or 3+) to obtain a score ranging between 1 and 300 as previously described[83,84]. HER2 amplification levels were measured when the result of IHC was 2+. The PathVysion HER2 DNA Probe kit (Abbot Laboratories, Des Plaines, Illinois, USA) was used to perform FISH analysis, according to the manufacturer's protocol. Any case with IHC 3+ or IHC 2+/FISH+ was considered to be HER2-positive, while cases with IHC 0 or IHC 1+ or IHC 2+/FISH− were considered as HER2-negative (Supplementary Data 1), according to criteria of the European Medicines Agency.

## Cell line

Human HEK293T (Cat# CRL-11268 from ATCC; RRID: CVCL_QW54), MKN45 (Cat# JCRB0254 from Japanese Collection of Research Bioresources (JCRB) Cell Bank, RRID: CVCL_0434), MGC803 (Cat# C6582 from Beyotime Biotechnology, RRID: CVCL_5334), and NCI-N87 (Cat# CRL-5822 from ATCC; RRID: CVCL_1603), were obtained and cultured in DMEM (GIBCO) with 10% FBS (GIBCO) in 5% $CO_2$ at 37 °C. Cells validation using short tandem repeat markers (STR) were performed by Meixuan Biological Science and Technology Ltd. (Shanghai). In detail, these cell lines were firstly tested cell species by PCR method using extracted total genomic DNA, and examined by STR profiling. Then, STR data were analyzed using the DSMZ (German Collection of Microorganisms and Cell Cultures) online STR database (http://www.dsmz.de/fp/cgi-bin/str.html). Cell lines were tested negative for mycoplasma contamination. All cells were grown according to the instruction.

## Protein extraction and trypsin digestion of GC FFPE samples

The biopsy tumor FFPE samples derived from 206 therapy-naïve GC patients were collected, and the tumor regions were determined by pathological examination. For clinical sample preparation, sections (10 μm thick) from FFPE blocks were macro-dissected, deparaffinized with xylene, and washed with ethanol. The ethanol was removed completely and the sections were left to air-dry. For this purpose, a hematoxylin-stained section of the same tumor was used as reference. Areas containing 80% or more tumor were examined independently by two expert gastrointestinal pathologists (C.X. and Y.H.).

Lysis buffer [0.1 M Tris-HCl (pH 8.0), 0.1 M DTT (Sigma, 43815), 1 mM PMSF (Amresco, M145)] was added to the extracted tissues, and subsequently sonicated for 1 min (3 s on and 3 s off, amplitude 25%) on ice. The supernatants were collected, and the protein concentration was determined using the Bradford assay. The extracted tissues were then lysed with 4% sodium dodecyl sulfate (SDS) and kept for 2–2.5 h at 99 °C with shaking at 1800 rpm. The solution was collected by centrifugation at 12,000 × g for 5 min. A fourfold volume of acetone was added to the supernatant and kept in −20 °C for a minimum of 4 h. Subsequently, the acetone-precipitated proteins were washed three times with cooled acetone. Filter-aided sample preparation (FASP) procedure was used for protein digestion[85]. The proteins were resuspended in 200 μL 8 M urea (pH 8.0) and loaded in 30 kD Microcon filter tubes (Sartorius) and centrifuged at 12,800 × g for 20 min. The precipitate in the filter was washed three times by adding 200 μL

50 mM $NH_4HCO_3$. The precipitate was resuspended in 50 μL 50 mM $NH_4HCO_3$. Protein samples underwent trypsin digestion (enzyme-to-substrate ratio of 1:50 at 37 °C for 18–20 h) in the filter, and then were collected by centrifugation at 12,800 × g for 15 min. Additional washing, twice with 200 μL of water, was essential to obtain greater yields. Finally, the centrifugate was pumped out using the AQ model Vacuum concentrator (Eppendorf, Germany).

## LC-MS/MS

Peptide samples were analyzed on a Q Exactive HF-X Hybrid Quadrupole-Orbitrap Mass Spectrometer (Thermo Fisher Scientific, Rockford, IL, USA) coupled with a high-performance liquid chromatography system (EASY nLC 1200, Thermo Fisher Scientific). Peptides, re-dissolved in Solvent A (0.1% formic acid in water), were loaded onto a 2-cm self-packed trap column (100-μm inner diameter, 3-μm ReproSil-Pur C18-AQ beads, Dr. Maisch GmbH) using Solvent A, and separated on a 150-μm-inner-diameter column with a length of 15 cm (1.9-μm ReproSil-Pur C18-AQ beads, Dr. Maisch GmbH) over a 75 min gradient (Solvent A: 0.1% formic acid in water; Solvent B: 0.1 % formic acid in 80 % ACN) at a constant flow rate of 600 nL/min (0–75 min, 0 min, 4% B; 0–10 min, 4–15% B; 10–60 min, 15–30% B; 60–69 min, 30–50% B; 69–70 min, 50–100% B; 70–75 min, 100% B). The eluted peptides were ionized under 2 kV and introduced into the mass spectrometer. MS was operated under a data-dependent acquisition mode. For the MS1 Spectra full scan, ions with m/z ranging from 300 to 1400 were acquired by Orbitrap mass analyzer at a high resolution of 120,000. The automatic gain control (AGC) target value was set as 3E + 06. The maximal ion injection time was 80 ms. MS2 Spectra acquisition was performed in top-speed mode. Precursor ions were selected and fragmented with higher energy collision dissociation with a normalized collision energy of 27%. Fragment ions were analyzed using an ion trap mass analyzer with an AGC target value of 5E + 04, with a maximal ion injection time of 20 ms. Peptides that triggered MS/MS scans were dynamically excluded from further MS/MS scans for 12 s. A single-run measurement was kept for 75 min. All data were acquired using Xcalibur software v2.2 (Thermo Fisher Scientific).

## Peptide and protein identification

MS raw files were processed using the Firmiana proteomics workstation[86]. Briefly, raw files were searched against the NCBI human Refseq protein database (released on 04-07-2013; 32,015 entries) using the Mascot search engine (version 2.4, Matrix Science Inc). The mass tolerances were: 20 ppm for precursor and 50 mmu for product ions collected by Q Exactive HF-X. Up to two missed cleavages were allowed. The database searching considered cysteine carbamidomethylation as a fixed modification, and N-acetylation, and oxidation of methionine as variable modifications. Precursor ion score charges were limited to +2, +3, and +4. For the quality control of protein identification, the target-decoy-based strategy was applied to confirm the FDR of both peptide and protein, which was lower than 1%. Percolator was used to obtain the quality value (q-value), validating the FDR (measured by the decoy hits) of every peptide-spectrum match (PSM), which was lower than 1%. Subsequently, all the peptides shorter than seven amino acids were removed. The cutoff ion score for peptide identification was 20. All the PSMs in all fractions were combined to comply with a stringent protein quality control strategy. We employed the parsimony principle and dynamically increased the q-values of both target and decoy peptide sequences until the corresponding protein FDR was <1%. Finally, to reduce the false positive rate, the proteins with at least one unique peptide were selected for further investigation.

## Label-free-based MS quantification of proteins

The one-stop proteomic cloud platform "Firmiana" was further employed for protein quantification. Identification results and the raw

data from the mzXML file were loaded. Then for each identified peptide, the extracted-ion chromatogram (XIC) was extracted by searching against the MS1 based on its identification information, and the abundance was estimated by calculating the area under the extracted XIC curve. For protein abundance calculation, the nonredundant peptide list was used to assemble proteins following the parsimony principle. The protein abundance was estimated using a traditional label-free, intensity-based absolute quantification (iBAQ) algorithm[87], which divided the protein abundance (derived from identified peptides' intensities) by the number of theoretically observable peptides. A match between runs[88] was enabled to transfer the identification between separate LC-MS/MS runs based on their accurate mass and retention time after retention time alignment. We built a dynamic regression function based on the commonly identified peptides in tumor samples. According to correlation value $R^2$, Firmiana chose linear or quadratic functions for regression to calculate the retention time (RT) of corresponding hidden peptides, and to check the existence of the XIC based on the $m/z$ and calculated RT. Subsequently, the fraction of total (FOT), a relative quantification value was defined as a protein's iBAQ divided by the total iBAQ of all identified proteins in one experiment, and was calculated as the normalized abundance of a particular protein among experiments. Finally, the FOT values were further multiplied by $10^5$ for ease of presentation, and missing values were assigned $10^{-527}$.

## Quality control of the mass spectrometry data

To quality control the MS performance, the HEK293T cell lysate was measured every three days as the quality control standard. The quality control standard was digested and analyzed using the same method and conditions as the GC samples. Pearson's correlation coefficient was calculated for all quality control runs using the R statistical analysis software v.3.5.1 (Supplementary Fig. 1b). The average correlation coefficient among the standards was 0.964, and the maximum and minimum values were 1 and 0.87, respectively. The $\log_{10}$ transformed FOTs for each GC sample (Supplementary Fig. 1d) were plotted to show consistency of data quality. The dynamic range of protein identification of each sample was shown according to the descending sort of protein abundance with a range of 6369–8119 proteins identified in each sample. The protein with highest intensity has the minimum rank number, representing the highest rank; the protein with lowest intensity has the maximum rank number, representing the maximum identification number in one sample.

## The proteomic subtypes generated by consensus clustering analysis and their biological function

The protein expression matrix of the gastric cancer cohort was used to identify the proteomic subtypes using the consensus clustering method implemented in the R package ConsensusClusterPlus v.3.8[89]. All FOTs $< 10^{-5}$ were replaced with $10^{-527}$. Prior to the consensus clustering analysis, we performed a $\log_{10}$ transformation and median centered normalization to facilitate the interpretation of the expression data. Then, the top 1,000 proteins with the highest median absolute deviation were subjected to ConsensusClusterPlus in R v.3.5.1 for unsupervised consensus clustering. The cluster analysis was performed with the following setting: maxK = 10, reps = 10,000, pItem = 0.8, pFeature = 1, clusterAlg = "hc", distance = "pearson" for the clustering runs. A preferred cluster result was selected by considering the profiles of the consensus cumulative distribution function (CDF) and delta area under the CDF curve for clustering solutions between 2 and 10 clusters. As shown in Supplementary Fig. 2a, c, the rank survey profiles of the consensus CDF and the delta area under the CDF curve, along with the consensus membership heat maps, indicated a four-subtype solution for 206 cases and 179 cases of GC using the proteomic data. This showed clear separation and the significant prognostic differences (OS) among 4 clusters (Fig. 2c and Supplementary Fig. 2b). To generate the

abundance heatmap shown in Fig. 2a, the GC samples in each subtype were rearranged from G-I to G-IV, using the signature protein abundance matrix enriched in the signature pathways for each subtype. The signature proteins of each subtype defined here should meet the following criteria: (1) detected in at least 10% of the subtype, (2) differentially expressed of the subtype compared with other subtypes with fold change >2 and $P < 0.05$ (log10-transformed FOT, two-sided Student's $t$ test). KEGG[90] pathway enrichment was performed to determine the biological function of proteomic subtypes.

## Correlation between proteomic subtype and clinical features

The association between the clinical information and proteomic subtypes was examined using Fisher's exact test for categorical data, including gender, grade, Lauren's type, primary site, TNM stage, HER2 status, and response. A two-sided log rank test and Kaplan–Meier survival curves were applied to compare the prognosis among the four proteomic subtypes (G-I to G-IV) using GraphPad Prism 8 software. To evaluate the prognostic power of the proteomic subtypes, we applied univariable Cox analysis and multivariable Cox analysis of the subtypes with known clinical covariates (such as grade and Lauren's type) (Supplementary Table 3). $P$-values $< 0.05$ were considered as significantly different. OS was used as primary endpoint. For proteomic subtype and clinical variables, hazard ratio was calculated from Cox proportional hazards regression analysis. $P$-values $< 0.05$ were considered as significantly different. OS was used as primary endpoint. Clinical variables analyzed with $P$-values $< 0.05$ using univariate analysis was chosen to enter Cox regression multivariate analysis. In the multivariate analysis, proteomics subtyping could also serve as an independent predictive factor after adjusting for clinical covariates.

## Validation of the FDGC subtyping in other independent cohorts

We validated the FDGC subtyping identified in our cohort using other dependent gastric cancer cohorts (BPRC cohort, EOGC, and ACRG cohort) using Rapidminer 9.6.0 (RapidMiner Inc, Boston, USA). Polynomial by Binomial Classification operator uses a binomial classifier and generates binomial classification models for different classes and then aggregates the responses of these binomial classification models for classification of polynomial label. The Fast large margin operator using logistic regression, applied in the subprocess of the Polynomial by Binomial Classification operator, was employed to build the prediction model based on signature proteins of each subtype in the FDGC cohort.

## The evaluation of MSI/MSS characteristics

We calculated the gene expression signature scores using the average of log intensity (also known as the geometric average) of expression of genes in the signature. We compared the association of FDGC subtyping in our cohort and ACRG cohort with the pre-defined published gene expression signatures relevant for MSI/MSS[35] examined by ANOVA test.

## The single sample gene set enrichment analysis (ssGSEA)

All scores were inferred by single sample gene set enrichment analysis (ssGSEA) method from the GSVA R package (v1.34.0) based on the protein expression matrix. The genset (c2.all.v7.4.symbols) of Molecular Signature Databse(MSigDB) was used to ssGSEA[91]. The parameters: min.sz = 10, max.sz = 300 were set and other parameters were used default.

## Differential protein and pathway analysis in subcohorts

The protein expression matrices of the DOS, XELOX, XELOX + HER2, and HER2 subcohorts were used to perform the differential expression analysis of the sensitive group (S) and the non-sensitive group (NS). The overrepresented proteins of S and NS of subcohorts were defined as proteins differentially expressed in the S and NS groups (NS/S > 2 or <0.5); the significantly differentially expressed proteins was defined as

proteins with more than twofold change and Wilcoxon rank-sum test with a Benjamini-Hochberg (BH) adjusted $p$ value cutoff (BH $p$ value < 0.05) (Supplementary Data 6 to 9). Pathway enrichment analysis of the overrepresented proteins was performed by KEGG[90] databases. Pathways with a $P$-value < 0.05 were regarded to be significant enrichment. Gene set enrichment analysis (GSEA) was also used for pathway enrichment analysis[92]. GSEA analysis in each proteomic subtype using the clusterProfiler R package (v3.18.1)[93]. GSEA evaluates and determines whether a priori defined sets of genes show statistically significant, cumulative changes in gene expression that are correlated with a specific phenotype. Samples grouped according to S and NS were subjected to GSEA, respectively. Molecular Signatures Database (MSigDB) of hallmark gene sets (H), curated gene sets (C2) and GO gene sets (C5) were used for enrichment analysis. A $P$ value of 0.05 was used as a cutoff. The enrichment score (ES) in GSEA was calculated by first ranking the proteins from the most to least significant with respect to S and NS, the entire ranked list was then used to assess how the proteins of each gene set were distributed across the ranked list.

### The assessment of drug sensitivity

Gene expression profiles of GC cell lines were from Expression 21Q2 Public dataset (https://depmap.org/portal/download/?releasename= DepMap+Public+21Q2&filename=CCLE_expression.csv). These RNA-Seq files were aligned with STAR and quantified with RSEM, then TPM-normalized and $\log_2(TPM+1)$ transformation. Reported values of genes were $\log_2(TPM+1)$. The drug sensitivity to docetaxel (BRD: BRD-K30577245-001-04-3), oxaliplatin (BRD: BRD-K78960041-001-03-2), and 5-fluorouracil (BRD: BRD-K24844714-001-24-5) using Drug sensitivity (PRISM Repurposing Primary Screen) 19Q4 dataset filtered by gastric cancer cell lines. Reported values of drug sensitivity were $\log_2$ [fold change (treatment group vs control group)]. The Pearson's correlation of gene expression value [$\log_2(TPM+1)$] and drug sensitivity [$\log_2$ fold change] were calculated. All datasets were downloaded from DepMap database (https://depmap.org/portal/).

### Immune cell type composition

The abundance of 64 different cell types were computed via xCell based on proteomic profiles[94]. The Supplementary Data 5 contains the final score computed by xCell of different cell types.

### Construction and validation of predictive models for therapy response

Multiple logistic regression analysis was used to construct the therapeutic response prediction model based on the significantly differentially expressed proteins in S and NS of DOS, XELOX, and HER2 subcohorts using in the R software v3.5.1. And backward stepwise method was utilized to feature selection. Samples was randomly divided into 80% of individuals (the training set) and the remaining 20% (the testing set)[95]. Moreover, the diagnostic value of this model was verified using ROC analysis (pROC R package version 1.16.2 and Caret R package version 6.0–86). Sensitivity, specificity, accuracy, and AUC were used to determine predictive values. The predictive model of XELOX cohort was validated in an external clinically annotated DGC cohort (the Beijing Proteome Research Center) accessible in the PRIDE Archive under the accession number PXD008840[27].

### Targeted PRM analysis

Using the library search results, a set of target peptides that unique to ECM proteins (THSD4, SRPX2, TGFBI, THBS1, and LAMB2), DSG/DNSG-sig (ATP5S, C11orf31, CDC42SE2, CHP2, and AHR), XSG/XNSG-sig (RFC2, NIT1, RAB32, FLG2, FNBP1, GCLC, DYNLRB1, RBBP7, LPXN, LMAN2, NUB1, WAS, FAM82B, and MYCBP), and HSG/HNSG-sig (CAPN5, BAIAP2, SRPX2, COMMD4, SCIN, DSC2, SEPSECS, TECPR1, DDX60L, NPL, SLC39A4, and IRF6) see Supplementary Data 3 and 11 for the list of targeted peptides) was selected and parallel reaction

monitoring (PRM) method was designed. Besides, house-keeping proteins, such as VCP, RPLP0, PSMB4, were also included for the reference. Equal amount of tumor tissue from each sample (an independent cohort with 60 gastric cancer FFPE samples, including 30 sensitive patients and 30 non-sensitive patients) was digested as described in the part of profiling preparation. Peptide samples were injected into the Q Exactive HF-X Hybrid Quadrupole-Orbitrap Mass Spectrometer (Thermo Scientific) operating in PRM mode with quadrupole isolation and HCD fragmentation. The full MS mode was measured at resolution 60,000 with AGC target value of 3E6 and maximum IT of 20 ms, with scanning range of 300 to 1400 m/z. Target ions were submitted to MS/MS in the HCD cell (1.6 m/z isolation width, 27% normalized collision energy). 60 PRM events were performed after MS1 scanning, at resolution 15,000 with AGC target value of 1E6 and maximum IT of 25 ms. Separation was achieved on a 150-μm-inner-diameter column with a length of 15 cm (1.9-μm ReproSil-Pur C18-AQ beads, Dr. Maisch GmbH) in an Easy 1200 nLC HPLC system (Thermo Scientific). Solvent A was 0.1 formic acid in water and solvent B was 0.1% formic acid, 80% ACN in water. Peptides were separated at 600 nL/min across a gradient ranging from 4 to 100% B over 75 min (0–75 min, 0 min, 4% B; 0–10 min, 4–15% B; 10–60 min, 15–30% B; 60–69 min, 30–50% B; 69–70 min, 50–100% B; 70–75 min, 100% B).

Raw data was searched by Skyline-daily (4.2.1.19004, University of Washington, USA). The proteins were quantified with the fragment total area reported by Skyline-daily. We selected peptides and tested their stability of signal and shape of peaks in the pool sample for final quantification, and referred to the ranking offered by skyline.

### Plasmid construction

For the analysis of CTSE in tumor proliferation and tumor response to drug treatment, we constructed stable cell lines overexpressing CTSE-FLAG. The cDNA of *CTSE* was cloned into the pCDH-CMV-EF1-PURO vector via the unique *XbaI* site and the neighboring *EcoRI* site. For convenient detection, a FLAG-tag encoding sequence (GATTA CAAGGATGACGACGATAAG) was inserted before the stop codon (TAG) to express the CTSE-FLAG fusion protein.

The primers used for plasmid construction as following:

Forward primer (5′–3′): GCTCTAGAATGGAATACTTCGGCAC TATC, Reverse primer (5′–3′):

GGAATTCTCACTTATCGTCGTCATCCTTGTAATCAGGTCTGTCA GACAGGCA.

As for TKTL1, PCR-amplified TKTL1 was cloned into pRK7-Flag vector between Hind*III* and EcoR*I*, using the primers as following:

Forward primer (5′–3′): CCCAAGCTTATGGCGGATGCTGAGGC GAGG, Reverse primer (5′–3′): CGGAATTCTTAGTTCAGCAACATGCA TTTCACGGC[78].

To explore the association of THSD4 with tumor proliferation and drug resistance, the cDNA of *THSD4* was cloned into the pcDNA3.1(b)-Flag vector via the unique Nhe*I* site and the neighboring EcoR*I* site.

Forward primer (5′–3′): gggagacccaagctggctagcATGGTTTCCCAT TTCATGGGG, Reverse primer (5′–3′): tgctggatatctgcagaattcTCTG CTCCCCAGGAAGCC).

### Lentivirus production and cell transduction

For CTSE, the double-stranded DNA was cloned into the pCDH-CMV-EF1-PURO vector; 8 μg of packaging plasmids pMD2.G: psPAX2 (1:3) and 8 μg of Lenti-vector containing the target gene were co-transfected into $2.5 \times 10^6$ HEK-293T cells using Lipofectamine 2000 transfection reagent (Invitrogen, 11668-019). The media containing the lentivirus particles were collected after 24 and 48 h, separately, and centrifuged at $1500 \times g$ for 10 min. These 24-h and 48-h supernatants were used independently to infect MKN45 and MGC803 cells in the presence of 10 μg/mL hexadimethrine bromide (Sigma, H9268) for 12 h. After infection, cells were cultured and selected with puromycin for the generation of stably overexpressed cells. Empty pCDH vector was used as a negative control.

For TKTL1 and THSD4, the detailed process of cell transfection was described as the previous publication[78].

## Proteome profiling and differential analysis of GC cell lines

MKN45 cells were divided into four groups according to the following experimental conditions: scramble vector (Vector OE) and CTSE-FLAG overexpression (CTSE OE) with or without docetaxel treatment. Each group contained at least three biological replicates. The protein concentration was determined using the Bradford assay. Cells were boiled in a 99 °C metal bath for 30 min with 100 μL of 50 mM ABC buffer (ammonium bicarbonate) containing SDS at a final concentration of 4%. Protein samples underwent trypsin digestion (enzyme-to-substrate ratio of 1:50 at 37 °C for 18–20 h). All peptide samples were desalinated using a C18 column (50% acetonitrile and 0.1% formic acid) and then analyzed using a Q Exactive HF-X Hybrid Quadrupole-Orbitrap Mass Spectrometer (Thermo Fisher Scientific, Rockford, IL, USA) coupled with a high-performance liquid chromatography system (EASY nLC 1200, Thermo Fisher). MS raw files generated by LC-MS/MS were searched against the NCBI human Refseq protein database (released on 04-07-2013; 32,015 entries) using MaxQuant (version 1.6.2.10) software enabled with Andromeda search engine. Protease was Trypsin/P. Up to 2 missed cleavages were allowed. Carbamidomethyl (C) was considered as a fixed modification. For the proteome profiling data, variable modifications were oxidation (M) and acetylation (Protein N-term). We screened the differentially expressed proteins in CTSE-overexpressing and control MKN45 cells without DOC treatment (FC > 1.5 or 0.7) or with DOC treatment (FC > 2 or 0.5). Pathway enrichment analysis was performed according to KEGG database.

## Western blotting

For western blotting, cells were lysed with 0.5% NP-40 buffer containing 20 mM Tris-HCl (pH 8.0), 100 mM NaCl, 1 mM EDTA, 1 mM PMSF, and NONIDET P-40 SUBSTITUTE. The protein concentration was quantified using Bradford assay. For each sample, 30 μg of protein extract was separated using 10% sodium dodecyl sulfate–polyacrylamide gel electrophoresis and transferred to nitrocellulose membranes. After blocking with 5% milk (BD Science) solution in TBST (Tris buffered saline with Tween) for 1–2 h, the membranes were incubated with TBST containing the appropriate primary antibodies overnight at 4 °C, followed by a 2 h incubation with horseradish peroxidase-conjugated secondary antibodies. The target protein bands were detected using the Chemiluminescent detection reagent. The mouse monoclonal anti-β-actin antibody (1:10,000, Genscript, catalog No: A00702), the rabbit polyclonal anti-THSD4 antibody (1:2000, ABclonal, catalog No: A17773), the rabbit polyclonal anti-CTSE antibody (1:2000, Signalway Antibody, catalog No: 35666), the rabbit polyclonal anti-TKTL1 antibody (1:1000, Novus Biologicals, catalog No: NBP1-31674), the rabbit polyclonal anti-alpha-tubulin antibody (1:2000, Proteintech, catalog No: 11224-1-AP), and the rabbit polyclonal anti-beta-tubulin antibody (1:2000, Proteintech, catalog No: 10094-1-AP) were used, and their specificity was confirmed by western blotting. Western blot quantification was performed using ImageJ software (Version 1.52a, National Institutes of Health, MD, USA).

## Proliferation assay

Cell proliferation was assessed using the Cell Counting Kit-8 (CCK8; Beyotime, C0046). The gastric cancer MKN45, MGC803, and NCI-N87 cell lines were seeded into 96-well plates (~2000 cells per well). The cells were treated with different doses of the chemotherapy treatments [DOS, XELOX, docetaxel (Aosaikang Pharm Co, Ltd, Jiangsu, China), 5-fluorouracil (Shanghai Xudong Haipu Pharmaceutical Co, Ltd, Shanghai, China), oxaliplatin (Jiangsu Hengrui Medicine Co, Ltd, Jiangsu, China), trastuzumab (Shanghai Roche Pharmaceutical Co., Ltd, Shanghai, China), and buparlisib (MedChemExpress Co., Ltd, Shanghai, China)] for 72 h. CCK8, at a final concentration of 10%, was added to each well and incubated for 1.5 h. Absorbance was measured at 450 and 630 nm and the data were analyzed using GraphPad Prism 8 software. The drug sensitivity was estimated by their half-maximal inhibitory concentration (IC50) values. The IC50 values were calculated as the concentrations of single agent.

## Immunohistochemistry staining and evaluation

A standard immunohistochemistry (IHC) protocol was followed to stain the tumor tissue samples using the rabbit polyclonal antibody against CTSE (1:100, Signalway Antibody, catalog No: 35666), and the rabbit polyclonal antibody against TKTL1 (1:500, Novus Biologicals, catalog No: NBP1-31674), the rabbit antibody monoclonal against CD4 (1:250, GeneTech, catalog No: GT219107), the mouse monoclonal antibody against CD8 (1:300, Leica, catalog NO: PA0183). IHC evaluation was analyzed using an IHC profiler compatible plugin with integrated options for the quantitative analysis of digital IHC images stained for cytoplasmic or nuclear proteins[96]. Moreover, the intensity of the cytoplasmic staining and the percentage of positively stained tumor cells were also scored numerically.

## Flow cytometric analysis of the cell cycle and apoptosis

For cell cycle analysis, ~$10^6$ cells were fixed in 4 °C pre-cooled 70% ethanol overnight at 4 °C. Following three washes, cells were incubated for 1 h at 37 °C in PBS with DNase-free RNase A (100 mg/mL) and propidium iodide (50 mg/mL). Next, the cells were analyzed using a Beckman Coulter flow cytometer (Beckman Coulter, Brea, CA, USA). Data were acquired and analyzed using FlowJo version 10.7.1 (Becton Dickinson Life Sciences). Data are presented as means ± SD of three independent experiments.

For cell apoptosis analysis, ~$10^6$ cells were harvested from culture dishes using trypsin (without EDTA). After two washes with cold PBS, cells were resuspended in 100 μL 1 × Binding Buffer (Annexin V-FITC/PI Apoptosis Detection Kit, YEASEN, Shanghai, China). Cells were stained with 5 μL annexin V-FITC and 10 μL PI staining solution (Annexin V-FITC/PI Apoptosis Detection Kit) in the dark, at room temperature for 15 min. Following this incubation, 400 μL 1 × Binding Buffer was added to each sample, and then kept on ice until analysis (within 1 h).

## Metaphase spreads

In order to count the chromosomes of a single cell in metaphase, MKN45 cells were incubated with 50 ng/mL colchicine (Sangon Biotech, A600322-0100) at 37 °C for 2 h. Cells were harvested and suspended in 8 mL prewarmed hypotonic buffer (0.075 mol/L KCl) at 37 °C for 20 min which induced cellular distention. After centrifugation, cells were fixed with Carnoy's buffer (methanol: acetic acid in 3:1 ratio) at 37 °C for 20 min. Fixed cells were centrifuged and then resuspended in Carnoy's buffer, twice. The final supernatants, containing the fixed mitotic cells, were dropped onto ice-cold slides and dried for 10 min. Slides were stained with 1 μg/mL DAPI (Sigma, D8417) for 1 min followed by washing with PBS. Cells were imaged using a Zeiss LSM 880 (Carl Zeiss, Jena, Germany) confocal microscope using an oil 63 × 1.4 NA objective lens.

## Immunocytochemistry and immunofluorescence

MKN45 cells were seeded into glass bottom culture dishes to reach 70–80% confluency. The cells were fixed using 4% paraformaldehyde in PBS (pH 7.4) for 10 min at room temperature, followed by three 5-min washes with ice-cold PBS. Then, the samples were permeabilized with PBS containing 0.1% Triton X-100 for 10 min at room temperature, and washed with PBS three times for 5 min. Cells were incubated with 1% BSA, 22.52 mg/mL glycine in PBST (PBS + 0.1% Tween 20) for 30 min at room temperature to block unspecific binding of the antibodies, prior to incubation with the diluted antibody in 1% BSA in PBST overnight at 4 °C. After three washes in PBS, the samples were incubated with the fluorescent-labeled secondary antibody in 1% BSA for 1 h at

room temperature in the dark. After three further washes in PBS, the samples were incubated with 1 μg/mL DAPI (Sigma, D8417) for 1 min. Finally, the samples were washed three times with PBS, for 5 min each wash, in the dark. Cells were imaged using a Zeiss LSM 880 (Carl Zeiss) confocal microscope using an oil 63 × 1.4 NA objective lens.

### Reporting summary

Further information on research design is available in the Nature Research Reporting Summary linked to this article.

## Data availability

The raw mass spectrometry (MS) proteomics data and parallel reaction monitoring (PRM)-MS proteomics data generated in this study have been deposited in the ProteomeXchange Consortium (dataset identifier: PXD024255) via the iProX partner repository (http://www.iprox. cn/)[97] under Project ID IPX0002116000. All the H&E-stained slides of tumor tissues in this study were deposited in Mendeley (https://data. mendeley.com/datasets/cv6ytf2fz7/1). The gene expression profiles of gastric cancer cell lines associated with drug sensitivity could be accessed at DepMap data portal (https://depmap.org/portal/). The gene expression profiles of gastric cancer cell lines in public dataset Expression 21Q2 in this study are available in the Depmap database (https://depmap.org/portal/download/?releasename=DepMap+Public +21Q2&filename=CCLE_expression.csv). HPA IHC Staining Data could be accessed at https://www.proteinatlas.org/. NCBI human Refseq protein database could be accessed at https://www.ncbi.nlm.nih.gov/ refseq/. Source data are provided with this paper. The remaining data are available within the Article, Supplementary Information or Source Data file. Source data are provided with this paper.

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

## Acknowledgements

This work is supported by the National Key Research and Development Program of China (2017YFA0505102 [C.D.], 2016YFA0502500 [C.D.], 2020YFE0201600 [C.D.], 2018YFE0201603 [C.D.], 2017YFA0505101 [C.D.], 2018YFA0507501 [Z.Y.Q.], and 2017YFC0908404 [Z.Y.Q.]), the National Natural Science Foundation of China (31770886 [C.D.], 31972933 [C.D.], and 31700682 [C.D.]), the Science and Technology Commission of Shanghai Municipality (2017SHZDZX01 [C.D.]), Shanghai Science Technique Planning Foundation (no. 19441904000 [Y.Y.H.]), Shanghai Municipal Key Clinical Specialty (shslczdzk 01302 [Y.Y.H.]), Shanghai Science and Technology Development Fund (No. 19MC1911000 [Y.Y.H.]), the Major Project of Special Development Funds of Zhangjiang National Independent Innovation Demonstration Zone (ZJ2019-ZD-004 [C.D.]), and China Postdoctoral Science Foundation (2020T130114 [J.W.F.], 2019M651268 [Y.Z.W.]).

## Author contributions

Y.Li., C.X., J.Y.Z., Y.Y.H., and C.D. conceived the work and designed the experiments; C.X., D.X.J., Y.L.L., Q.S., X.L.Z., A.S., J.H., and Y.Y.H. collected the tissue samples; Y.Li., B.W., F.J.X., Y.Y.Q., D.X.J., K.L., S.T., Y.Z.W., and Z.Y.Q. performed the experiments and acquired the MS data; J.Q., T.S.L., and K.T.S. provided expertise and technical support; Y.Li., B.W., Y.Y.Q., F.H.M., K.L., J.W.F., X.H.W., and Y.Liu. analyzed the data; Y.Li. and C.D. wrote the original manuscript. All authors contributed to data interpretation, manuscript editing, and revision.

## Competing interests

The authors declare no competing interests.

## Additional information

[1]State Key Laboratory of Genetic Engineering and Collaborative Innovation Center for Genetics and Development, School of Life Sciences, Institute of Biomedical Sciences, Human Phenome Institute, Zhongshan Hospital, Fudan University, Shanghai 200433, China. [2]Department of Pathology, Zhongshan Hospital, Fudan University, Shanghai 200032, China. [3]State Key Laboratory of Cell Differentiation and Regulation, Henan International Joint Laboratory of Pulmonary Fibrosis, Henan center for outstanding overseas scientists of pulmonary fibrosis, College of Life Science, Institute of Biomedical Science, Henan Normal University, Xinxiang 453007, China. [4]Department of Oncology, The Affiliated Hospital of Southwest Medical University, Luzhou 646000, China. [5]Department of Urology, Fudan University Shanghai Cancer Center, Shanghai 200032, China. [6]Department of Oncology, Shanghai Medical College, Shanghai 200032, China. [7]Shanghai Genitourinary Cancer Institute, Shanghai 200032, China. [8]Department of General Surgery, Zhongshan Hospital, Fudan University, Shanghai 200032, China. [9]Department of Oncology, Zhongshan Hospital, Fudan University, Shanghai 200032, China. [10]Institute for Developmental and Regenerative Cardiovascular Medicine, MOE-Shanghai Key Laboratory of Children's Environmental Health, Xinhua Hospital, Shanghai Jiao Tong University School of Medicine, Shanghai 200092, China. [11]Department of Anatomy and Neuroscience Research Institute, School of Basic Medical Sciences, Zhengzhou University, Zhengzhou 450001, China. [12]These authors contributed equally: Yan Li, Chen Xu, Bing Wang, Fujiang Xu, Fahan Ma.
✉e-mail: liu.tianshu@zs-hospital.sh.cn; shen.kuntang@zs-hospital.sh.cn; zhaojy@fudan.edu.cn; hou.yingyong@zs-hospital.sh.cn; chend@fudan.edu.cn

