## [Peer Review File · Nature Communications]

Proteomic Characterization of Gastric Cancer Response to Chemotherapy and Targeted therapies Reveals Potential Therapeutic StrategiesReviewers' Comments:

Reviewer #1:

Remarks to the Author:

In the here presented manuscript, the authors performed proteomics characterization of 206 formalin-fixed paraffin-embedded (FFPE) tumors tissues from patients with gastric cancer (GC) undergoing either chemotherapy (DOS or XELOX) or anti-HER2-based therapy. They identified four proteomic-based subtypes with different molecular features and clinical outcomes including drug response. The authors also revealed the potential resistance mechanisms for different chemo- or targeted-treatments. And finally they developed prognostic models to predict the chemotherapeutic response. This manuscript presented a comprehensive proteomic resource of chemo-/targeted therapy in GC and could be further strengthened in response to the following comments.

Major comments :

1. Why the authors only selected 179 samples with long-term follow-up to identify the proteomic subtypes instead of using all the 206 samples to conduct the unsupervised clustering? It would be better to use the results from all of the 206 samples to evaluate the molecular features or clinical outcomes among different subtypes.
2. How about the tumor purity of these tumor samples? Is there any difference of tumor purity among these proteomic subtypes which may influence the molecular features?
3. The authors suggested THSD4 was significantly associated with poor prognosis and drug resistance. Further validation experiments (for example, overexpressed THSD4 using GC cell lines) should be added to confirm this conclusion.
4. High expression of extracellular matrix proteins are associated with drug resistance, immunohistochemical analysis or PRM analysis using mass spectrometry from sensitive or non-sensitive patients of these extracellular matrix proteins may further validate the role of indicators to predict chemotherapy response.
5. In figure 3i,l, and Supplementary Fig. 3f, why the authors only used patients in stage IV?
6. The authors constructed the predictive models for GC chemo- and targeted therapies using the proteomic data, and finally identified 6, 14, 12 proteins for stratifying patients sensitive or non-sensitive to DOS, XELOX and HER2, respectively. Further targeted quantitative analysis may be conducted to validate its predictive role and improve its usage in clinical.

Minor comments :

1. The authors suggested that the G-I was dominant for endocytosis, but in figure 2d, endocytosis pathway was also enriched for G-III, how to explain this inconsistency?
2. In figure 2d, several proteins were presented repeatedly such as DNM2, DNM3...
3. In line 237-238, it is confused that figure 2g did not show the association of THSD4 expression and the prognosis. ("and the high expression of THSD4 was significantly associated with poor prognosis (log rank test, $P < 0.05$) (Fig. 2g)").
4. How to explain the difference of the percentage of each subtype in these three cohorts, this should be at least discussed.
5. The legend in figure 3c is very confused and should be improved to help readers understand.
6. In line 291, the authors suggested that G-IV had lower immune infiltration, but it could not be observed in Supplementary Fig. 3e.
7. The layout of figure 7 is confused, it should be consistent with the corresponding text.

Reviewer #2:

Remarks to the Author:

In this manuscript, Li et al. performed proteomics profiling of a large gastric tumor cohort with 206 GC patients, including 44 patients treated with DOS, 70 with XELOX, and 71 with HER2. Unsupervised analysis of the proteomics data identified four proteomic subtypes characterized by different clinical and molecular features. Supervised comparison between treatment sensitive and resistant groups

further identified proteins and pathways that are associated with treatment response. In particular, CTSE overexpression was found to enhance DOC sensitivity, whereas TKTL1 overexpression was associated with DOC resistance. Functional perturbation experiments were further performed in cell lines to support a causal connection between these proteins and DOC sensitivity.

Lacking treatment response information is a typical limitation in other cancer proteomics studies. Therefore, I applaud the authors for assembling such a large tumor cohort with treatment response information for proteomic profiling to address a clinically important question. The authors have performed a large amount of informatics analyses, revealing interesting new insights into the biology and treatment of GC. Moreover, the dataset itself, which is deposited in public repositories, will serve as a rich resource for the GC research community. I have a few suggestions for the authors to consider to further strengthen the manuscript.

1. The FDGC subtype has no association with the previously published BPRC subtype ($p=0.42$). However, its concordance with the ACRG subtype is significant ($p=5.0e-4$). This is puzzling because both the FDGC and the BPRC subtypes are based on proteomics profiling, whereas the ACRG subtype is based on mRNA profiling. The authors should provide potential explanations for this result, especially why there is no concordance between the two proteomic subtypes. It would also be useful to further include the GC subtypes from Mun et al. (Cancer Cell, 2019, PMID 30645970) for comparison.
2. There are a couple problems with the construction and validation of predictive classifiers (Figure 6). First, feature selection (DEP in Figure 6a) should be performed within the training set only to avoid overfitting. Second, if I understand correctly, the validation cohort in Figure 6c is not really an independent validation cohort, most of the samples are from the discovery cohort. This is very misleading. Validation cohort should not reuse samples from the discovery cohort. Validation result from the BPRC cohort (assuming there is no sample overlap between the BPRC and the FUDAN cohort) is encouraging. However, due to the small sample size of both the discovery and validation cohorts, it would be important to publish the final model so that it can be tested in the future when new datasets are available. Moreover, to further demonstrate the value of proteomics data in predictive modeling, the authors should check whether drug sensitivity can be accurately predicted using other clinical attributes.
3. The final proteomics data table (supplementary Data 1c) includes 11287 genes, but 1754 genes (16%) were identified in only a single sample, 5316 (47%) were identified in less than 10% of the samples, and 7683 (68%) were identified in less than 50% of the samples. It is not clear whether all proteins in this table are used for downstream data analysis. It is well known that both detection variation and quantification variation are high for proteins with low abundance, and proteins that cannot be robustly quantified by the platform should be excluded from downstream analyses. The authors should define and justify which proteins are considered "quantifiable". This is important both for this study and for future reuse of the data by the research community.
4. The authors should provide clinical data for individual patients, including all features in Table 1 and treatment response data, in a supplementary table to enable effective reuse of the dataset.
5. Figure 1d is confusing. First, there are 12,519 proteins but the largest rank is below 10,000. Second, some proteins with relatively higher intensity ranked higher than proteins with relatively lower intensity. For example, the protein with the second lowest intensity ranked at around 7000 rather than at the very end.
6. Page 8, line 214, what does frequency >10% mean?

Reviewer #3:

Remarks to the Author:

Comments

In this manuscript, Li et al. generated an impressive proteomic dataset consisting of 206 treatment-naive FFPE tumour tissues from GC patients who have received either triplet combination

chemotherapy DOS (docetaxel, oxaliplatin and S-1), doublet chemotherapy XELOX (capecitabine and oxaliplatin) or anti-HER2-based therapy (trastuzumab). Proteomic-based classification identified 4 subtypes with different therapeutic clinical responses and molecular features. Using differential proteomics analysis between therapy-sensitive and non-sensitive groups, the authors explored key pathways that predict and underlie therapeutic resistance, including activation of immune, ECM and PI3K-Akt pathways. They also developed prognostic models to predict chemotherapeutic responses. Finally, the authors highlighted 2 key proteins (CTSE and TKTL1) that may regulate chemo-sensitivity, supported with extensive relevant in vitro experiments.

Overall, given that current large-scale proteomic information of GC remains mostly limited to diffuse GCs and pre-cancerous gastric lesions, this paper provides a highly relevant and novel dataset that will certainly be of interest to the GC community. This paper also seeks to address a scientifically important question of drug resistance predictors and mechanisms, and provides a valuable resource that can enrich our understanding. The data analysis and experiments performed were also generally well planned and executed, with clear descriptions of findings and figures.

Major Comments

1. Many figures employ the use of heatmaps to show differential expression of genes (e.g. Fig 3f, 3k, 4m,5h) or differential enrichment of cell types (e.g. Fig 4e, 4f). However, not all the genes/cell types that are listed are significantly differential (since they are not marked with *). This makes it difficult to interpret and also begs the question of how the genes/cell types listed in the figure are being selected. The authors should consider for e.g. putting exact p-values for each gene/cell type listed so readers can be clear about the difference between groups compared.
2. The utility of performing unsupervised proteomic-based classification to reveal 4 subtypes (as shown in Fig2) is not quite apparent. After all, the main aim seems to be identifying predictors/mechanisms of resistance, and thus a direct comparison of sensitive vs. non-sensitive groups would be most useful (as done in Fig4/5). If the authors choose to delineate 4 subtypes G-I to G-IV as in Fig2, it would be nice if they could then further discuss and analyse differences in predictors/mechanisms of resistance for each subtype since each has its own share of non-responders. For e.g. Fig 2e-2g showing high expression of ECM and its association with drug resistance is likely to only apply for group G-IV. What about G-I, G-II and G-III?
3. Following up on above point, do the authors foresee the proteomic subtyping system (G-I/II/III/IV) to have utility in the clinics? For e.g. different subtypes might have different resistance mechanisms, and hence require different ways to alleviate resistance?
4. The authors propose MSI tumours to be more sensitive to drug treatments in general due to higher activated immune response in Fig3m. However, the association of MSI/MSS-sig level and therapy response is very weak and also only marginally significant at $p=0.045$ (Fig 3d). Since the authors subsequently show that MSI-sig high GCs benefit only from DOS and not XELOX (Fig 4a), would the analysis in Fig3 benefit from analysing different drug treatment groups separately?
5. The combination of anti-HER2 therapy to overcome XELOX resistance is an interesting finding and could be further pursued and/or supported by experimental evidence. For e.g., does combinatorial treatment of anti-HER2 and XELOX have synergistic effects in vitro?
6. It is also not clear how anti-HER2 therapy "alleviates the immune-related XELOX resistance" and "activates anticancer immune response" (page 15). It appears that the authors have only presented evidence that GC patients with high TCR signalling are unlikely to benefit from XELOX, but instead respond to XELOX + anti-HER2. Whether the role of anti-HER2 therapy affects TCR signalling has not been directly addressed.
7. The authors propose ECM proteins as negatively associated with anti-HER2 therapy (Fig 5). However most of the non-responders to anti-HER2 therapy seem to fall in G-II proteomic subtype characterized by glycolysis and panthothenate/CoA biosynthesis (Fig 2b). Can the authors also further explore/discuss the role of glycolytic pathway etc in anti-HER2 therapy resistance?
8. To further support Fig5, does PI3K-AKT inhibition synergize with anti-HER2?

REVIEWER COMMENTS

Reviewer #1, expertise in mass spectrometry-based proteomics (Remarks to the Author):

Comments

In the here presented manuscript, the authors performed proteomics characterization of 206 formalin-fixed paraffin-embedded (FFPE) tumors tissues from patients with gastric cancer (GC) undergoing either chemotherapy (DOS or XELOX) or anti-HER2-based therapy. They identified four proteomic-based subtypes with different molecular features and clinical outcomes including drug response. The authors also revealed the potential resistance mechanisms for different chemo-or targeted-treatments. And finally they developed prognostic models to predict the chemotherapeutic response.

This manuscript presented a comprehensive proteomic resource of chemo-/targeted therapy in GC and could be further strengthened in response to the following comments.

Response:

We appreciate the reviewer for the constructive and insightful comments, which help to improve the quality of this manuscript. Here, we summarized the reviewer's comments as following: (1) about the proteomic subtyping in 206 samples; (2) about the tumor purity of all tumor samples; (3) about further validation experiments, including THSD4-overexpression in GC cell lines, and PRM analysis of extracellular matrix proteins and predictive signature proteins in an independent validation cohort; (4) about the prognosis analysis of monocytes in stage IV; (5) about the extraction of dominant pathways in proteomic subtypes; (6) about the discussion of distribution difference of therapy subcohorts among proteomic subtypes; (7) about figure layout, and other minor points. The point-to-point responses are as follows.

Major comments:

Q1. Why the authors only selected 179 samples with long-term follow-up to identify the proteomic subtypes instead of using all the 206 samples to conduct the unsupervised clustering? It would be better to use the results from all of the 206 samples to evaluate the molecular features or clinical outcomes among different subtypes.

Response to Q1:

Thanks for the comment. According to reviewer's suggestions, we divided the response into two parts to answer: (1) the reason for selecting 179 samples for proteomic subtyping; (2) the proteomic subtyping in 206 samples.

(1) About the reason for selecting 179 samples for proteomic subtyping

Firstly, we apologized for not explaining it clearly. Among the 206 patients in the study, the 179 patients had complete follow-up information. In the previous version, we performed the consensus clustering analysis only in the 179 patients after excluding the patients who were lost to follow-up. The consensus clustering analysis of the proteomic profiles among 179 samples identified four subtypes, which were featured with distinct biological functions and clinical outcomes, especially therapy response. Furthermore, univariate and multivariable cox analysis of overall survival revealed that proteomic subtyping served as an independent predictive factor.

(2) About the proteomic subtyping in 206 samples

Secondly, according to reviewer's suggestion, we performed consensus clustering analysis on the 206 samples based on proteomic profiles, which resulted in a similar four subtype: G-I (N = 29), G-II (N = 60), G-III (N = 97), and G-IV (N = 20) (**Figure RL1a and 1b**). Sankey plot showed high concordance between the two kinds of proteomic subtyping systems, among which only three samples were not branched into the corresponding subtypes (**Figure RL1b**). Furthermore, we performed a survival analysis on 179 patients who had complete follow-up information among four proteomic subtypes. We observed the proteomic subtypes of 206 samples were also associated with prognosis, among which the G-IV subtype had the worst overall survival (log rank test, $P = 0.034$) (**Figure RL1c**). The signature proteins of each subtype were identified with the same criteria ($P < 0.05$; fold change > 2), which resulted in 315, 773, 849, and 499 signature proteins in G-I to G-IV, separately (**Figure RL1d**). The functional enrichment analysis based on these signature proteins determined the dominant bioprocesses of each subtype (**Figure RL1e**). The G-I subtype was dominant for endocytosis ($P = 2.9E-4$); the G-II subtype

was featured by metabolic related pathways, including glycolysis/ gluconeogenesis ($P = 6.6E-3$), pantothenate/CoA biosynthesis ($P = 0.01$), inositol phosphate metabolism ($P = 2.1E-4$). In the G-III subtype, we observed proteins involved in ribosome, lysosome, and Fc epsilon RI signaling pathways were up-regulated ($P < 0.05$). Excitingly, we observed the same biological pathways, including ECM-receptor interaction ($P = 2.0E-3$), focal adhesion ($P = 0.016$), complement/coagulation cascades ($P = 1.2E-5$), and PI3K-AKT signaling pathway ($P = 0.042$), were enriched in G-IV subtype, validating the biological pathways related poor prognosis and therapy resistance. Overall, the overrepresented pathways of four subtypes were consistent between 206 samples and 179 samples for proteomic subtyping.

In the revision, we mainly focused on the proteomic subtyping system of 179 samples with the complete survival information during the further analysis, which showed the similar patterns of molecular features and clinical outcomes with proteomic subtyping system of 206 samples. The proteomic subtyping systems of 206 samples and 179 samples were both shown in the **Fig. 2**, **Supplementary Fig. 2 and 3 in the revision**. The corresponding association of the two subtyping systems was included in the **Supplementary Data 3**. Finally, we updated the relative description in the part of “The proteomic subtypes generated by consensus clustering analysis and their biological function” in the “Method” section (**line 1299–1322 in Page 47–48**) and in the part of “Proteomic subtyping of the GC cohort and their association with a therapeutic response” in the “Result” section (**line 198–271 in Page 8–10**) in the revision.

Figure RL1. The comparison of proteomic subtypes between 206 GC samples and 179 GC samples. **a** The consensus clustering analysis of 206 cases of GC and four subtypes were generated. k was tested from 2 to 10. Consensus matrices, as well as the consensus cumulative distribution function (CDF) plot, delta area (change in CDF area) plot, and tracking plot are shown. **b** Sankey diagram indicating the comparison of proteomic subtypes between 206 GC samples and 179 GC samples. **c** The Kaplan–Meier curves of overall survival (OS) of 179 patients of each proteomic subtype identified in proteomic subtypes of 206 GC samples. P -value was calculated by two-sided log rank test. **d** The signature proteins of each subtype. **e** Bubble plot showing the KEGG pathway enrichment of G-I, G-II, G-III, and G-IV subtypes in both 179 samples and 206 samples.

Q2. How about the tumor purity of these tumor samples? Is there any difference of tumor purity among these proteomic subtypes which may influence the molecular features?

Response to Q2:

Thanks for the comment. We apologized for not providing the exact tumor purity values and explaining it clearly. During the process of sample collection, the histologic sections were obtained from top and bottom portions of tumor tissues and Hematoxylin and eosin (H&E)-stained for review. Each tumor sample was checked by two expert pathologists and with at least $\geq 80\%$ tumor purity. To answer reviewer's questions, we reviewed again all H&E-stained slides of tumor tissues of patients involved in the study. We respond as following:

Firstly, all the H&E-stained slides of tumor tissues were uploaded to the Mendeley data (<https://data.mendeley.com/datasets/cv6ytf2fz7/draft?a=03f59fd3-c90a-4ca4-bb04-ddc4475c5f5e>), demonstrating the high purity ($\geq 80\%$) and quality of tumor tissues. For example, we showed 8 H&E-stained slides of tumor samples below (**Figure RL2**). In the revision, we updated the relevant description in the part of "Data and materials availability" in the "Method" as following (**line 1571–1577 in Page 57**):

"All the H&E-stained slides of tumor tissues were uploaded to the Mendeley data (<https://data.mendeley.com/datasets/cv6ytf2fz7/draft?a=03f59fd3-c90a-4ca4-bb04-ddc4475c5f5e>). The accession number for the MS proteomics data reported in this paper is iProX repository (www.iprox.org): the ProteomeXchange ID PXD024255, and the project ID IPX0002116000 (<https://www.iprox.cn/page/PSV023.html?url=1640781547717wcRH>, password: eiBJ)."

Secondly, in the revision, we supplemented tumor purity values of tumor samples of all the patients in the **Supplementary Data 1**, which included all clinical characteristics for individual patients.

Thirdly, according to reviewer's suggestions, we compared the tumor purity among proteomic subtypes. The result showed there was no difference of tumor purity among four subtypes (ANOVA test, $P > 0.05$) (**Figure RL3, see also Supplementary Fig. 2e in the revision**). In the

revision, we updated the relevant description in the part of “Proteomic subtyping of the GC cohort and their association with a therapeutic response” in the “Results” as following (line 220–223 in Page 8):

“Additionally, an obvious association between proteomic subtyping and therapy subcohort or TNM stage was determined (Fisher's exact test, $P < 0.01$), but this association was not observed with either grade, Lauren's type, the primary site, or tumor purity ($P > 0.05$) (Fig. 2b and Supplementary Fig. 2e).”

In conclusion, we provided the exact tumor purity values and H&E-stained slides of all 206 patients in the supporting materials. The statistical analysis revealed no difference of tumor purity among proteomic subtypes.

Figure RL2. Hematoxylin and eosin (H&E)-stained slides of tumor tissues for quality control.

Figure RL3. Barplot for tumor purity among four proteomic subtypes.

Q3. The authors suggested THSD4 was significantly associated with poor prognosis and drug resistance. Further validation experiments (for example, overexpressed THSD4 using GC cell lines) should be added to confirm this conclusion.

Response to Q3:

Thanks for the constructive comment. In the study, to investigate the association of proteins expression with therapy resistance, we analyzed the correlation of the extracellular matrix proteins significantly upregulated in the G-IV subtype with drug sensitivity (half maximal inhibitory concentration [IC50]) using gastric cancer cell lines data from the Cancer Dependency Map Project (DepMap). Then, we identified 5 proteins up-regulated in G-IV subtype, including THSD4, SRPX2, TGFBI, THBS1, and LAMB2, which showed high correlation with drugs response (5-FU, oxaliplatin, or docetaxel) (Pearson $r > 0.4$, $P < 0.05$) (**Fig. 2e in the revision**). Among these proteins, only THSD4 was resistant to all three drugs (5-FU, oxaliplatin, or docetaxel), and the high expression of THSD4 was significantly associated with poor prognosis (log rank test, $P < 0.05$) (**Fig. 2e and Supplementary Fig. 3i in the revision**). Here, according to reviewer's suggestion, we performed validation experiments as follows, to confirm the result that THSD4 was significantly associated with poor prognosis and drug resistance.

Firstly, we overexpressed THSD4 in two gastric cancer cell lines MKN45 and MGC803 (**Figure RL4a, see also Supplementary Fig. 3j, k in the revision**). We wondered whether THSD4 overexpression could affect cell proliferation. We assessed the cell proliferation in control group, empty vector-overexpressing group, and THSD4-overexpressing group by CCK8 assay. The result revealed that THSD4 overexpression significantly promoted the proliferation of MKN45 and MGC803 cells (Student's t test, $P < 0.05$), compared with the cells transfected with an empty vector (**Figure RL4b, c, see also Supplementary Fig. 3l, m in the revision**).

Secondly, to further explore the association of THSD4 with drug resistance, we treated THSD4-overexpressing and empty vector-overexpressing MKN45 and MGC803 cells with

docetaxel, oxaliplatin, and 5-FU, respectively. The drug sensitivities in the MKN45 and MGC803 cells were estimated by their half-maximal inhibitory concentration (IC₅₀) values. We then compared the IC₅₀ values of docetaxel, oxaliplatin, and 5-FU between THSD4-overexpressing cells and empty vector-overexpressing cells, to assess the change of drug sensitivities. We observed consistent results in the two GC cell lines as follows. In detail, the IC₅₀ values of docetaxel, oxaliplatin, and 5-FU were significantly higher (4.26-fold, 1.44-fold, and 1.38-fold increase in MKN45, respectively; Student's t test, $P < 0.05$) in the THSD4-overexpressing MKN45 cells (IC₅₀, 12.33 nM, 16.90 μM, and 2.75 μM, respectively) compared with the empty vector-overexpressing cells (IC₅₀, 2.89 nM, 11.7 μM, and 2.0 μM, respectively) (**Figure RL4d–f, see also Fig. 2f in the revision**). Consistently, we observed the similar change that IC₅₀ values of docetaxel, oxaliplatin, and 5-FU were significantly higher (3.60-fold, 1.47-fold, and 1.20-fold increase in MGC803) in the THSD4-overexpressing MGC803 cells (IC₅₀, 15.97 nM, 27.18 μM, and 2.35 μM, respectively) compared with the empty vector-overexpressing cells (IC₅₀, 4.44 nM, 18.45 μM, and 1.95 μM, respectively) (**Figure RL4d–f, see also Fig. 2f in the revision**).

In conclusion, the experiments further validated THSD4 overexpression reduced the anti-tumor effect of chemotherapeutic drugs, including docetaxel, oxaliplatin, and 5-FU. Finally, we added the **Figure RL4** into the **Fig. 2 and Supplementary Fig. 3**, and updated the “Results” (see **line 281–296 in Page 11**) and “Methods” (see **line 1453–1469 in Page 52–53**) in the revised manuscript.

Figure RL4. Effect of THSD4 overexpression on anticancer activity of chemotherapeutic drugs in gastric cancer cell lines MKN45 and MGC803. **a** Immunoblot analysis of THSD4 overexpression and the normalization of a qualified western blot. **P*-values were calculated by two-tailed Student's *t* test. **** *P* < 0.0001. **b, c** Effect of THSD4 overexpression on proliferation in gastric cancer (GC) cell lines (MKN45 and MGC803; N = 3 biological repeats). Data presented as mean \pm SD. **P*-values calculated using two-tailed Student's *t* test. **d, e, and f** Dose-response curves of MKN45 and MGC803 cell lines overexpressing THSD4 after 72-h treatments with docetaxel, oxaliplatin, and 5-FU (mean \pm SD, N = 3 biological repeats). IC₅₀, half-maximal inhibitory concentration. **P*-values calculated using two-tailed Student's *t* test.

Q4. High expression of extracellular matrix proteins are associated with drug resistance, immunohistochemical analysis or PRM analysis using mass spectrometry from sensitive or non-sensitive patients of these extracellular matrix proteins may further validate the role of indicators to predict chemotherapy response.

Response to Q4:

Thanks for the comment. In the study, to investigate the association of proteins expression with therapy resistance, we analyzed the correlation of the extracellular matrix proteins significantly upregulated in the G-IV subtype with drug sensitivity (half maximal inhibitory concentration [IC₅₀]) using gastric cancer cell lines data from the Cancer Dependency Map Project (DepMap).

We screened 5 extracellular matrix proteins up-regulated in G-IV subtype, including THSD4, SRPX2, TGFBI, THBS1, and LAMB2, which showed high correlation with drugs response (5-FU, oxaliplatin, or docetaxel) (Pearson $r > 0.4$, $P < 0.05$), indicating the association of these proteins with drug resistance (**Fig. 2e in the revision**).

As mentioned by the reviewer, the parallel reaction monitoring (PRM) assays are powerful targeted approaches to detect and quantify pre-specified proteins with a high throughput using high-resolution mass spectrometers (*Mol Cell Proteomics*, 2012, PMID: 22962056) ¹. We firstly constructed a new independent cohort composed of 60 GC patients receiving either DOS (N = 20), XELOX (N = 20), or anti-HER2 (N = 20) therapies. All samples were histologically scored by two expert gastrointestinal pathologists (C.X. and Y.H) according to the widely accepted Response Evaluation Criteria in Solid Tumors (RECIST) (version1.1) by CT/MRI scanning and grouped into complete response (CR), partial response (PR), stable disease (SD), or progressive disease (PD). In this study, patients with CR and PR were defined as sensitive (S) and those with SD and PD were defined as non-sensitive (NS). The independent cohort was grouped into DOS-sensitive group (DSG, N = 10), DOS-non-sensitive group (DNSG, N = 10), XELOX-sensitive group (XSG, N = 10), XELOX-non-sensitive group (XNSG, N = 10), HER2-sensitive group (HSG, N = 10), and HER2-non-sensitive group (HNSG, N = 10) (**Figure RL5a, Supplementary Data 3 in the revision**).

Next, to validate the association between extracellular matrix proteins and resistance of drugs (5-FU, oxaliplatin, and docetaxel), we employed the targeted MS approach, PRM assays, to quantify these proteins in FFPE tumor tissues from patients receiving DOS therapy (triplet combination chemotherapy of 5-FU, oxaliplatin, and docetaxel). We then selected a set of target peptides that unique to these ECM proteins (including THSD4, SRPX2, TGFBI, THBS1, and LAMB2) using the library search results (**Table RL1, see also Table S1 in the revision**). Besides, house-keeping proteins, such as VCP, RPLP0, PSMB4, were also included for the quality control (*Trends Genet*, 2013, PMID: 23810203) ². The fragment total areas of targeted peptides reported by Skyline-daily (4.2.1.19004, University of Washington, USA) were used to quantify these proteins. As a result, we observed these ECM proteins were higher expressed

in DNSG compared with DSG in PRM-MS experiments (Fold change (DNSG/DSG) > 2, $P < 0.05$, Wilcoxon rank-sum test): THSD4 (Fold change (DNSG/DSG) = 2.49, $P = 5.2E-3$), SRPX2 (Fold change (DNSG/DSG) = 7.10, $P = 0.035$), TGFBI (Fold change (DNSG/DSG) = 1.97, $P = 0.035$), THBS1 (Fold change (DNSG/DSG) = 7.63, $P = 1.5E-3$), and LAMB2 (Fold change (DNSG/DSG) = 2.94, $P = 0.043$) (**Figure RL5b, see also Fig. 2g in the revision**). In conclusion, our data demonstrated the association between these ECM proteins and drug resistance was validated by PRM approach in an independent cohort. In addition, the association of THSD4 with chemotherapy resistance was also validated in gastric cancer cell lines data from the Cancer Dependency Map Project (DepMap). Furthermore, in the revision, according to the reviewer's comments of Q3, we validated THSD4 overexpression reduced the anti-tumor effect of chemotherapeutic drugs *in vitro*, including docetaxel, oxaliplatin, and 5-FU. Collectively, our results illustrated the high expression of extracellular matrix proteins were associated with drug resistance, and these extracellular matrix proteins could serve as indicators to predict chemotherapy response.

In the revision, we have added the analysis result in the part of "Proteomic subtyping of the GC cohort and their association with a therapeutic response" in the "Result" (**line 298–320 in Page 11–12**). Meanwhile, we updated the description of "Targeted PRM analysis" in the "Method" (**line 1410–1438 in Page 51–52**).

Figure RL5. The validation of ECM proteins by Targeted MS analysis. **a** The new independent cohort for PRM validation. **b** Boxplot showing the differential expression. P values calculated by Wilcoxon rank-sum test.

Table RL1. Targeted peptides from ECM proteins.

Group	Protein	Sequence	m/z [Da]
ECM proteins	LAMB2	QLDALLEALK	557.32905
ECM proteins	LAMB2	AMDYDLLLLR	555.28645
ECM proteins	TGFB1	EGVYTVFAPTNEAFR	850.91708
ECM proteins	TGFB1	YGTLFTMDR	552.26273
ECM proteins	TGFB1	VLDELK	409.23695
ECM proteins	THBS1	MENAELDVPIQSVFTR	924.96118
ECM proteins	THBS1	TIVTTLQDSIR	623.85358
ECM proteins	THBS1	FQDLVDAVR	531.78209
ECM proteins	THSD4	FSPHRPDNLVPPAPQPPR	506.27236
ECM proteins	THSD4	FYEWEPFAEVK	722.84286
ECM proteins	THSD4	SWFLTEWSER	670.8163
ECM proteins	SRPX2	EQQLSANIIEELR	771.90974

Q5. In figure 3i,I, and Supplementary Fig. 3f, why the authors only used patients in stage IV?

Response to Q5:

Thanks for the comment. In the study, we evaluated the tumor immune microenvironment among four subtypes (G-I to G-IV) by xCell analysis, and observed G-IV was featured with lower xCell score of immune cells, such as CD4+ Tcm, macrophages M1, and CD8+ T-cells. In contrast with macrophages M1, precursor monocytes were aggregated in the G-IV subtype (**Fig. 3g**). We further investigated the association of monocytes/macrophages M1 with the clinical outcome. Among the 206 patients in the study, the 179 patients had complete follow-up information. We firstly explored the association of monocytes with prognosis in 179 patients with follow-up information. The survival analysis revealed that there was no association of monocytes with overall survival in 179 patients (log-rank $P > 0.05$) (**Figure RL6a**). We wondered whether the association of monocytes with overall survival could be affected by different TNM stages. Then, we validated the association of monocytes with prognosis among patients in stage II, III, and IV, respectively. The results revealed that, in the patients of stage IV, but not of stage II and III, higher xCell score of monocytes reflected poor prognosis (log-rank $P = 0.041$; HR = 1.79; 95%CI, 0.99 to 3.24). Consistently, we confirmed the association of monocytes with prognosis in the patients of stage IV (log-rank $P = 0.037$; HR = 1.70; 95%CI, 1.04 to 2.78), but not of stage II and III, in the independent ACRG cohort (**Figure RL6a, b**). The result indicated the significance to explore the prognosis power of monocytes in the patients of stage IV.

As for the IL18, a secreted pro-inflammatory factor of macrophages M1, the survival analysis of IL18 revealed the positive association with prognosis in patients of stage IV. This result further validated the prognosis power of monocytes/macrophages M1 in stage IV. Therefore, the survival curves presented in the previous **Fig. 3i, I, and Supplementary Fig. 3f (corresponding to Fig. 3j, I, and Supplementary Fig. 4g in the revision)** were for patients

in stage IV. Thank reviewer for pointing out. We have revised the relative description in the revision, making the result more precise. Please see **line 404–422 in Page 15–16**.

Figure RL6. The association of monocytes score with overall survival. **a** The survival analysis of monocytes xCell score with OS according to different TNM stages in FDGC cohort. **b** The survival analysis of monocytes score with OS according to different TNM stages in ACRG cohort.

Q6. The authors constructed the predictive models for GC chemo- and targeted therapies using the proteomic data, and finally identified 6, 14, 12 proteins for stratifying patients sensitive or non-sensitive to DOS, XELOX and HER2, respectively. Further targeted quantitative analysis may be conducted to validate its predictive role and improve its usage in clinical.

Response to Q6:

Thanks for the constructive comment. As reviewer mentioned, we employed stepwise logistic regression and identified a subset of signatures that accurately discriminated DSG/DNSG, XSG/XNSG, and HSG/HNSG (named as DSG/DNSG-sig, XSG/XNSG-sig, HSG/HNSG-sig). According to reviewer's suggestion, we employed a targeted MS approach, parallel reaction

monitoring (PRM) in the revision, which has been adopted in classifier's validation in recent proteomic research (*Cell*, 2020, PMID: 32795414; *J Extracell Vesicles*, 2020, PMID: 32363013)^{3, 4}, to validate predictive power of the signature proteins.

For the validation, we firstly constructed a new independent cohort composed of 60 GC patients receiving either DOS (N = 20), XELOX (N = 20), or anti-HER2 (N = 20) therapies. All samples were histologically scored by two expert gastrointestinal pathologists (C.X. and Y.H) according to the widely accepted Response Evaluation Criteria in Solid Tumors (RECIST) (version1.1) by CT/MRI scanning and grouped into complete response (CR), partial response (PR), stable disease (SD), or progressive disease (PD). In this study, patients with CR and PR were defined as sensitive (S) and those with SD and PD were defined as non-sensitive (NS). The independent cohort was grouped into DOS-sensitive group (DSG, N = 10), DOS-non-sensitive group (DNSG, N = 10), XELOX-sensitive group (XSG, N = 10), XELOX-non-sensitive group (XNSG, N = 10), HER2-sensitive group (HSG, N = 10), and HER2-non-sensitive group (HNSG, N = 10) (**Figure RL7a, see also Supplementary Data 11 in the revision**).

Then, we selected a set of target peptides that unique to these signature proteins, including DSG/DNSG-sig (ATP5S, C11orf31, CDC42SE2, CHP2, and AHR), XSG/XNSG-sig (RFC2, NIT1, RAB32, FLG2, FNBP1, GCLC, DYNLRB1, RBBP7, LPXN, LMAN2, NUB1, WAS, FAM82B, and MYCBP), and HSG/HNSG-sig (CAPN5, BAIAP2, SRPX2, COMMD4, SCIN, DSC2, SEPSECS, TECPR1, DDX60L, NPL, SLC39A4, and IRF6) using the library search results (**Table RL2, see also Table S1 in the revision**). Besides, house-keeping proteins, such as VCP, RPLP0, PSMB4, were also included for the quality control (*Trends Genet*, 2013, PMID: 23810203)², indicating the stability of MS platform. Followed by this, PRM strategy was designed to quantify these signature proteins in FFPE tumor tissues from the independent cohort. The fragment total areas of targeted peptides reported by Skyline-daily (4.2.1.19004, University of Washington, USA) were used for protein quantification.

Based on the PRM quantification, we performed comparative analysis of signature proteins (including DSG/DNSG-sig, XSG/XNSG-sig, and HSG/HNSG-sig) between DSG and DNSG,

XSG and XNSG, HSG and HNSG, respectively. As a result, we observed the significantly differential expression of these signature proteins between DSG and DNSG, XSG and XNSG, HSG and HNSG, respectively. In DOS subcohort, we observed signature proteins, including AHR, ATP5S, C11orf31, CDC42SE2, and CHP2, had at least 2-fold differences between DSG and DNSG ($P < 0.05$, Wilcoxon rank-sum test). In XELOX subcohort, XSG/XNSG-sig (RFC2, NIT1, RAB32, FLG2, FNBP1, GCLC, DYNLRB1, RBBP7, LPXN, LMAN2, NUB1, WAS, FAM82B, and MYCBP) were significantly increased in XNSG compared with XSG (Fold change (XNSG/XSG) > 2 , $P < 0.05$, Wilcoxon rank-sum test). In HER2 subcohort, we observed HSG/HNSG-sig (CAPN5, BAIAP2, SRPX2, COMMD4, SCIN, DSC2, SEPSECS, TECPR1, DDX60L, NPL, SLC39A4, and IRF6) had a more than 2-fold increase in HNSG compared with HSG (Fold change (HNSG/HSG): 2.17, 2.22, 4.44, 5.43, 5.71, 4.60, 5.27, 5.03, 4.06, 2.18, 5.40, 3.35; $P < 0.05$, Wilcoxon rank-sum test) (**Figure RL7b, d, and f, see also Supplementary Fig. 6b, c, d in the revision**). In addition, the heatmaps showed a clear separation between DSG and DNSG, XSG and XNSG, HSG and HNSG in the new independent cohort, respectively (**Figure RL7c, e, and g, see also Fig. 6c in the revision**). Overall, predictive power of the signature proteins in different therapies (including DOS, XELOX, and anti-HER2) was validated in a new independent cohort by PRM assays. Finally, the PRM-MS-based proteomics data have been deposited to the iProX repository and available via the same ProteomeXchange ID PXD024255 and project ID IPX0002116000.

In the revision, we have added the descriptions in the part of “Construction and validation of the predictive models for GC chemo- and targeted therapies” in the “Result” (**line 619–645 in Page 23–24**). Meanwhile, we have updated the description of “Targeted PRM analysis” in the “Method” (**line 1410–1438 in Page 51–52**).

Figure RL7. The validation of DSG/DNSG-sig, XSG/XNSG-sig, and HSG/HNSG-sig by Targeted MS analysis in a new independent cohort of 60 GC patients. **a** An independent cohort for PRM validation. **b, d, and f** Boxplot showing the differential expression (Log₁₀-transformed) of the DSG/DNSG-sig (**b**), XSG/XNSG-sig (**d**), and HSG/HNSG-sig (**f**). *P* values calculated by Wilcoxon rank-sum test. **c, e, and g** Heatmaps showing DSG/DNSG-sig (**c**), XSG/XNSG-sig (**e**), and HSG/HNSG-sig (**g**) validated by targeted MS in an independent cohort of 60 GC patients.

Table RL2. Targeted peptides that unique to signature proteins.

Group	Protein	Sequence	m/z [Da]
a. DSG_DNSG-sig model	AHR	LASLLPFPQDVINK	777.94803
a. DSG_DNSG-sig model	AHR	NDFSGEVDFR	593.2622
a. DSG_DNSG-sig model	AHR	SFFDVALK	463.75358
a. DSG_DNSG-sig model	ATP5S	DYNHLPTGPLDK	457.23033
a. DSG_DNSG-sig model	ATP5S	TALPSLELK	486.29223
a. DSG_DNSG-sig model	C11orf31	AEAAVVAVAEK	529.29855
a. DSG_DNSG-sig model	C11orf31	FPEPQEVVEELK	722.37176
a. DSG_DNSG-sig model	C11orf31	LEAPELPVK	498.29255
a. DSG_DNSG-sig model	C11orf31	NAAALSQALR	507.78859
a. DSG_DNSG-sig model	CDC42SE2	GGYGGGMPANVQMLVDTK	961.95833
a. DSG_DNSG-sig model	CHP2	IIESFFPDGSQR	698.34931
b. XSG_XNSG-sig model	DYNLRB1	GVQGIIVNTEGIPIK	818.98537
b. XSG_XNSG-sig model	DYNLRB1	STMDNPTTTQYASLMHSFILK	1193.5759
b. XSG_XNSG-sig model	DYNLRB1	DIDPQNDLTFLR	723.86482
b. XSG_XNSG-sig model	FAM82B	LLVYEALEYAK	656.36311
b. XSG_XNSG-sig model	FAM82B	QIQTEAAQLLTSFSEK	897.46675
b. XSG_XNSG-sig model	FAM82B	ESEDAELLWR	624.29973
b. XSG_XNSG-sig model	FLG2	NPDDPDTVDMHMLDR	991.95196
b. XSG_XNSG-sig model	FLG2	SVVTVIDVFYK	635.35899
b. XSG_XNSG-sig model	FNBP1	FEAWLAEVEGR	653.82494
b. XSG_XNSG-sig model	FNBP1	TEIELSYAK	527.27613
b. XSG_XNSG-sig model	FNBP1	ADYSSILQK	512.76921
b. XSG_XNSG-sig model	GCLC	SLFFPDEAINK	640.829
b. XSG_XNSG-sig model	GCLC	DPLTLFEELK	1091.56403
b. XSG_XNSG-sig model	GCLC	YDSIDSYLSK	595.7828
b. XSG_XNSG-sig model	LMAN2	LTVMTDLEDK	1164.58275
b. XSG_XNSG-sig model	LMAN2	LPTGYFFGASAGTGDLSDNHDIIISMK	910.76176
b. XSG_XNSG-sig model	LMAN2	DNVDDPTGNFR	625.27584
b. XSG_XNSG-sig model	LPXN	TSAAAQLDELMAHLTEMQAK	720.35421
b. XSG_XNSG-sig model	LPXN	DFLAMFSPK	528.2661
b. XSG_XNSG-sig model	MYCBP	HHLGAATPENPEIELLR	633.0017
b. XSG_XNSG-sig model	MYCBP	VLVALYEEPEKPNALDFLK	1138.11934
b. XSG_XNSG-sig model	MYCBP	SGVLDLTK	467.26625
b. XSG_XNSG-sig model	NIT1	LLEEYQLAR	618.33505
b. XSG_XNSG-sig model	NIT1	DPAETLHLSEPLGGK	782.40285
b. XSG_XNSG-sig model	NIT1	IDLNYLR	453.75561
b. XSG_XNSG-sig model	NUB1	IAETFGLQENYIK	763.39894
b. XSG_XNSG-sig model	NUB1	VDNLLQLGFTAQEAR	837.94435
b. XSG_XNSG-sig model	NUB1	TTGIATIEVFLPPR	757.93262
b. XSG_XNSG-sig model	RAB32	ATIGVDFALK	517.79737
b. XSG_XNSG-sig model	RAB32	VLVIGELGVGK	542.34155
b. XSG_XNSG-sig model	RAB32	DNINIEEAAR	572.78302
b. XSG_XNSG-sig model	RBBP7	EMFEDTVEER	642.77431
b. XSG_XNSG-sig model	RBBP7	GEFGGFGSVTGK	571.77703
b. XSG_XNSG-sig model	RFC2	LNEIVGNEDTVSR	723.36492
b. XSG_XNSG-sig model	RFC2	EGNVPNIIAGPPGTGK	817.45021
b. XSG_XNSG-sig model	RFC2	LTDAILTR	515.79832

b. XSG_XNSG-sig model	WAS	LIYDFIEDQGGLEAVR	919.47159
b. XSG_XNSG-sig model	WAS	SGPLPPVPLGIAPPPPTPR	620.69603
b. XSG_XNSG-sig model	WAS	GAPAVQQNIPSTLLQDHENQR	1158.58841
c. HSG_HNSG-sig model	BAIAP2	ELGDVLFQMAEVHR	548.61244
c. HSG_HNSG-sig model	BAIAP2	EGDLITLLVPEAR	713.40058
c. HSG_HNSG-sig model	BAIAP2	GYFDALVK	456.74482
c. HSG_HNSG-sig model	CAPN5	DFFFQNPQYIFEVK	888.43511
c. HSG_HNSG-sig model	CAPN5	ADPDNLQALHTLHLR	857.45585
c. HSG_HNSG-sig model	CAPN5	GENLAIGFDIYK	670.34826
c. HSG_HNSG-sig model	COMMD4	HSVDGESLSSELQQLGLPK	675.34723
c. HSG_HNSG-sig model	COMMD4	VDYTLSSSLLQSVVEEPMVHLR	801.7451
c. HSG_HNSG-sig model	COMMD4	FQVLLAELK	530.82347
c. HSG_HNSG-sig model	DDX60L	DLSIAVQMMK	568.29536
c. HSG_HNSG-sig model	DSC2	LTDPTGWVTIDENTGSIK	973.98939
c. HSG_HNSG-sig model	DSC2	VIPDDLAQQNLIVSNTEAPGDDK	1226.61809
c. HSG_HNSG-sig model	DSC2	GPGVDQEPR	477.73553
c. HSG_HNSG-sig model	IRF6	LQISTPDIKDNIVAQLK	948.54465
c. HSG_HNSG-sig model	NPL	AEELLDGILDK	608.32642
c. HSG_HNSG-sig model	NPL	DILINFLK	488.29747
c. HSG_HNSG-sig model	SCIN	IQEGEPEEFWNSLGGK	974.95191
c. HSG_HNSG-sig model	SCIN	ATEVPLSWDSFNK	747.36458
c. HSG_HNSG-sig model	SCIN	DYQTSPLLETQAEDHPPR	1048.99899
c. HSG_HNSG-sig model	SEPSECS	ITNSLVLDIIK	614.87919
c. HSG_HNSG-sig model	SLC39A4	ADLVAEESPELLNPEPR	939.97643
c. HSG_HNSG-sig model	SRPX2	EQQLSANIIEELR	771.90974
c. HSG_HNSG-sig model	TECPR1	LVTSGPWLEVPPIALR	874.50847

Minor comments:

Q7. The authors suggested that the G-I was dominant for endocytosis, but in figure 2d, endocytosis pathway was also enriched for G-III, how to explain this inconsistency?

Response to Q7:

Thanks for the comments. In the study, we performed a comparative analysis of proteomic profiling among four subtypes, which resulted in 301 (G-I), 611 (G-II), 925 (G-III), and 467 (G-IV) signature proteins (fold change > 2; $P < 0.05$). The KEGG pathway enrichment illustrated distinct bioprocess among the four subtypes based on these signature proteins. According to reviewer's comments, we firstly surveyed the hypergeometric test in pathway enrichment, usually p-value is equal or smaller than 0.05 to be considered strongly enriched in the annotation categories. We found endocytosis pathway was more significantly enriched in G-I ($P = 4.73E-4$) than G-III ($P = 0.042$) (**Figure RL8a**). Furthermore, ssGSEA analysis showed endocytosis pathway score of the G-I subtype was higher than G-III (two-sided Student's t test, $P < 0.05$) (**Figure RL8b**). Therefore, we defined endocytosis pathway as the dominant pathway

in G-I subtype. We apologized for not refining more dominant pathway as the feature of subtypes, thus misleading the reader in understanding the feature of subtypes in the previous version. In the revision, we presented the dominant pathways of four proteomic subtypes as shown in **Figure RL8c** (see also **Fig.2d** in the revision).

Figure RL8. The endocytosis pathway in proteomic subtype. **a** The KEGG pathway enrichment result of endocytosis in G-I and G-III. **b** Boxplot for endocytosis pathway score assessed by ssGSEA analysis between G-I and G-III. **c** Bubble plot showing the KEGG pathway enrichment of G-I, G-II, G-III, and G-IV.

Q8. In figure 2d, several proteins were presented repeatedly such as DNM2, DNM3...

Response to Q8:

Thanks for the comment. We apologized for the repeat presentation. We have removed the repeated proteins such as DNM2 and DNM3 from **Fig. 2d** in the revision.

Q9. In line 237-238, it is confused that figure 2g did not show the association of THSD4 expression and the prognosis. (“and the high expression of THSD4 was significantly associated with poor prognosis (log rank test, $P < 0.05$) (Fig. 2g)”).

Response to Q9:

Thanks for the comment. We apologized that Supplementary Fig. 2d, but not Fig. 2g, should be cited here in the previous version. In the previous Supplementary Fig. 2d, we observed the high expression of THSD4 was significantly associated with poor prognosis (log rank test, $P < 0.05$). In the revision, we accordingly rearranged subfigures, and moved the Supplementary Fig. 2d into **Supplementary Fig. 3i**. Therefore, we corrected “Fig. 2g” as “**Supplementary Fig. 3i**” in the revised manuscript as following (Please see **line 278–281 in Page 10–11**):

“Among these proteins, only THSD4 was related to the resistance of all three drugs (5-FU, oxaliplatin, or docetaxel), and the high expression of THSD4 was significantly associated with poor prognosis (log rank test, $P < 0.05$) (Fig. 2e and Supplementary Fig. 3i).”

Q10. How to explain the difference of the percentage of each subtype in these three cohorts, this should be at least discussed.

Response to Q10:

Thanks for the comment. First of all, we have noticed the distribution difference of three therapy subcohorts among four proteomic subtypes. According to reviewer’s comments, we conducted a further statistical analysis of the association of proteomic subtypes with therapy subcohorts. The result revealed the significant distribution difference of therapy subcohorts among proteomic subtypes, among which HER2 subcohort was mainly enriched in G-II subtype (Fisher’s exact test, $P < 0.05$) (**Figure RL9a**). After statistical analysis, we found the difference of the percentage of each subtype in these three subcohorts were mainly derived from HER2 subcohort, while no significant difference was observed in DOS and XELOX subcohorts (**Figure RL9b and c**).

We further explored why only anti-HER2 targeted therapy, but not chemotherapies, were distributed differentially among proteomic subtypes. We surveyed many clinical and biological researches about anti-HER2 targeted therapy (*Lancet*, 2010, PMID: 20728210; *Bmc Cancer*, 2016, PMID: 26857702; *Eur J Cancer*, 2015, PMID: 25661103)^{5,6,7}. These researches showed

that only HER2-positive gastric cancer patients were selected to receive anti-HER2 targeted therapy. Therefore, we speculated the distribution difference among proteomic subtypes could be caused by the specific proteomic feature related to HER2 expression, due to the selection of HER2-positive patients for anti-HER2 targeted therapy. Then, we evaluated HER2 expression detected by immunohistochemistry (IHC) and FISH analysis. Any case with IHC 3+ or IHC 2+/FISH+ was considered to be HER2-positive, while cases with IHC 0 or IHC 1+ or IHC 2+/FISH- were considered as HER2-negative (**Supplementary Data 1 and Table 1**). As expected, we observed the higher proportion (71%) of HER2-positive patients in G-II subtype compared with other subtypes (G-I: 21%, G-III: 43%, G-IV: 40%) (**Figure RL9d**). This result demonstrated the specific molecular pattern in HER2-positive GC patients, which could be identified at proteome level. As for chemotherapies (DOS and XELOX therapy), there was no distribution difference among proteomic subtypes, due to no selection of patients prior to therapy. These results supported the reliability and importance of proteomic subtyping of this study.

Finally, we have updated the result in **Supplementary Fig. 2e in the revision**, and the relevant contents in parts of “Result” (**line 223–240 in Page 8–9**) and “Discussion” (**line 792–798 in Page 29**) in the revision.

Figure RL9. The difference of the percentage of each subtype in these three subcohorts. **a** The barplot for each subcohort among four subtypes. **b** The barplot for HER2 subcohort among four subtypes. **c** The barplot for DOS and XELOX subcohorts among four subtypes. **d** The barplot for HER2-positive status and HER2-negative status among four subtypes.

Q11. The legend in figure 3c is very confused and should be improved to help readers understand.

Response to Q11:

Thanks for the comment. We apologized for the unclear description of legend in Fig. 3c. We updated a detailed statement of Fig. 3c in the part of Figure Legend in the revision (Please see **line 1012–1016 in Page 37**). We corrected in parts of “Figure Legends” in the revision as following:

“**c** Heatmap of CNV GI (genomic instability index), MSI-H assay status, mutations, and MSI/MSS-sig (MSI/MSS gene expression signatures assessment) of ACRG cohort classified by FDGC subtyping (upper). Heatmap of MSI/MSS-sig of FDGC cohort (bottom). *P*-value was calculated by Fisher's exact test (categorical variables) and ANOVA test (continuous variables). **P* < 0.05, ***P* < 0.01, ****P* < 0.001, *****P* < 0.0001.”

Q12. In line 291, the authors suggested that G-IV had lower immune infiltration, but it could not be observed in Supplementary Fig. 3e.

Response to Q12:

Thanks for the comment. We apologized for the unclear statement. The reviewer is correct that lower immune infiltration in G-IV was not observed in Supplementary Fig. 3e. In this study, we observed MSI/MSS-sig score and immune score showed significant difference among four proteomic subtypes (ANOVA test, *P* < 0.05) (**Fig. 3g**). Among four proteomic subtypes, G-IV had the lowest MSI/MSS-sig score. As for immune infiltration, we found proteomic subtypes were featured with different cell types, among which macrophages M1 were enriched in G-II and precursor monocytes were aggregated in the G-IV (**Figure RL10, see also Fig. 3h in the revision**). Further analysis of this study focused on the association of macrophages M1 or monocytes with clinical outcomes and drug resistance. Therefore, we removed the Supplementary Fig. 3e, and related inappropriate description in the revision. We have updated the figure and its related statement as “We found that G-IV had the lowest MSI/MSS-sig score (**Fig. 3g**). In addition, the proteomic subtypes were featured with different cell types, among which macrophages M1 were enriched in G-II. In contrast with macrophages M1, precursor

monocytes were aggregated in the G-IV subtype (Fig. 3g, h).” in the revision (Please see **line 393–396 in Page 15**).

Figure RL10. The differential score of Macrophages M1 and Monocytes among four subtypes.

Q13. The layout of figure 7 is confused, it should be consistent with the corresponding text.

Response to Q13:

Thanks for the constructive comment. We apologized for the confusion caused by the inconsistency between figure layout and its text. In the revised Fig. 7 (**Figure RL11**), we rearranged the subgraphs corresponding to the manuscript. The previous Fig. 7a was moved to Supplementary Fig. 7a in the revision (**Figure RL12**). In addition, we corrected of the corresponding figure legend (Please see **line 1106–1135 in Page 40–41**) and these subgraphs' citation in the revised manuscript.

Figure RL11. Effect of CTSE and TKTL1 overexpression on anticancer activity of DOS and XELOX in gastric cancer cell lines MKN45 and MGC803. **a, b** Effect of CTSE overexpression on proliferation in gastric cancer (GC) cell lines (MKN45 and MGC803; N = 3 biological repeats). Data presented as mean \pm SD. **P*-values calculated using two-tailed Student's *t* test. **c–e** Dose-response curves of MKN45 and MGC803 cell lines overexpressing CTSE after 72-h treatments

with DOS, XELOX, and DOC (mean \pm SD, N = 3 biological repeats). IC50, half-maximal inhibitory concentration. **P*-values calculated using two-tailed Student's *t* test. **f, g** KEGG pathway analysis showing the differential function in the CTSE group with or without DOC treatment. **h** Immunoblot analysis of soluble alpha-tubulin and soluble beta-tubulin in MKN45 cells after DOC treatment, and the normalization of qualified western blots. **P*-values calculated using two-tailed Student's *t* test. ****P* < 0.001; **** *P* < 0.0001. **i, j** Effect of TKTL1 overexpression on MKN45 and MGC803 cell proliferation (N = 3 biological repeats). Data presented as mean \pm SD, **P*-values calculated using two-tailed Student's *t* test. **k–m** Dose-response curves of MKN45 and MGC803 cell lines overexpressing TKTL1 after 72-h treatments with DOS, XELOX, and DOC (mean \pm SD, N = 3 biological repeats). IC50, half-maximal inhibitory concentration. **P*-values calculated using two-tailed Student's *t* test. **n** Chromosome segregation defects in DOC-treated MKN45 cells with or without TKTL1 overexpression. The left panel shows the representative results of chromosome segregation. The scale bar indicates 10 μ m. The right panel shows the statistical results from 3 biological repeats in the MKN45 cells (mean \pm SD). **P*-values calculated using two-tailed Student's *t* test. **P* < 0.05. **o** The percentage of aneuploid cells in the DOC-treated MKN45 cells, with or without TKTL1 overexpression. The left panel shows the representative results of normal and abnormal chromosome numbers. The scale bar indicates 10 μ m. The right panel shows the statistical results from 3 biological repeats in the MKN45 cells (mean \pm SD). **P*-values calculated using two-tailed Student's *t* test. **P* < 0.05. **p** Schematic illustration indicating how CTSE induces microtubule stabilization, and how TKTL1 induces abnormal chromosome segregation. Abbreviations are as follows: DSG, DOS-sensitive group; DNSG, DOS-non-sensitive group; XSG, XELOX-sensitive group; XNSG, XELOX-non-sensitive group; DOS, the triplet chemotherapy (docetaxel, oxaliplatin and S-1); DOC, docetaxel; IC50, half-maximal inhibitory concentration; OE, overexpression; SD, standard deviation.

Figure RL12. A possible mechanism by which CTSE modulates docetaxel sensitivity via microtubule stabilizing effects. **a** Heatmap of several key proteins in DSG/DNSG and XSG/XNSG. Barplot of fold change of these proteins. *P*-values were calculated by Wilcoxon rank-sum test. *P* < 0.05 considered statistically significant. **b** Immunoblot analysis of CTSE overexpression and the normalization of a qualified western blot. **P*-values were calculated by

two-tailed Student's t test. **** $P < 0.0001$. **c, d** Dose-response curves of GC cell lines (MKN45 and MGC803) overexpressing CTSE after 5-FU and OXA treatments, with an endpoint measurement at 72 h (mean \pm SD, N = 3 biological repeats). **e, f** Barplot showing the numbers of identified proteins in MKN45 cells overexpressing CTSE with or without DOC treatment. **g** A Venn diagram showing the protein overlap of vector OE, CTSE OE, vector OE treated with DOC, and CTSE OE treated with DOC. **h** Effects of CTSE overexpression on the regulation of microtubule associated proteins in MKN45 cells after DOC treatment (mean \pm SD, N = 3 biological repeats). * P -values were calculated by two-tailed Student's t test based on log₁₀-transformed FOT. * $P < 0.05$. **i–l** Immunohistochemistry (IHC) staining and qualification of CTSE in representative examples in the DOS and XELOX subcohorts. The scale bar indicates 50 μ m. * P -values were calculated by two-tailed Student's t test. **** $P < 0.0001$. Abbreviations are as follows: OXA, oxaliplatin; 5-FU, 5-fluorouracil; DOC, docetaxel; SD, standard deviation; IHC, immunohistochemistry.

Reviewer #2, expertise in proteomics and bioinformatics (Remarks to the Author):

Comments

In this manuscript, Li et al. performed proteomics profiling of a large gastric tumor cohort with 206 GC patients, including 44 patients treated with DOS, 70 with XELOX, and 71 with HER2. Unsupervised analysis of the proteomics data identified four proteomic subtypes characterized by different clinical and molecular features. Supervised comparison between treatment sensitive and resistant groups further identified proteins and pathways that are associated with treatment response. In particular, CTSE overexpression was found to enhance DOC sensitivity, whereas TKTL1 overexpression was associated with DOC resistance. Functional perturbation experiments were further performed in cell lines to support a causal connection between these proteins and DOC sensitivity.

Lacking treatment response information is a typical limitation in other cancer proteomics studies. Therefore, I applaud the authors for assembling such a large tumor cohort with treatment response information for proteomic profiling to address a

clinically important question. The authors have performed a large amount of informatics analyses, revealing interesting new insights into the biology and treatment of GC. Moreover, the dataset itself, which is deposited in public repositories, will serve as a rich resource for the GC research community. I have a few suggestions for the authors to consider to further strengthen the manuscript.

Response:

We appreciate the reviewer for the positive evaluation and constructive comments. Here, we made point-to-point responses to all comments. In addition, the key results were updated in the revised version, including (1) the application of FDGC subtyping algorithm in EOGC cohort, (2) the construction of the new validation cohort composed of 60 GC patients receiving either DOS (N = 20), XELOX (N = 20), or anti-HER2 (N = 20) therapies, and (3) the PRM analysis of the predictive signatures. The point-to-point responses are as follows.

Q14. The FDGC subtype has no association with the previously published BPRC subtype ($p=0.42$). However, its concordance with the ACRG subtype is significant ($p=5.0e-4$). This is puzzling because both the FDGC and the BPRC subtypes are based on proteomics profiling, whereas the ACRG subtype is based on mRNA profiling. The authors should provide potential explanations for this result, especially why there is no concordance between the two proteomic subtypes. It would also be useful to further include the GC subtypes from Mun et al. (Cancer Cell, 2019, PMID 30645970) for comparison.

Response to Q14:

Thanks for the constructive comments. Here, we divided the response into two parts to answer: (1) the reason for no concordance between FDGC and the BPRC subtypes; (2) the application of FDGC subtyping algorithm in EOGC cohort (Cancer Cell, 2019, PMID: 30645970).

(1) About there is no concordance between FDGC and the BPRC subtypes

To find the reason for no concordance between FDGC and the BPRC subtypes, we surveyed the composition of different clinical groups, such as TNM stage, gender, and Lauren's type in

FDGC and BPRC cohorts. The result showed there was obvious difference in the composition of Lauren's type, but not in other clinical groups, between two cohorts. In terms of Lauren's type, the FDGC cohort was composed of 21% diffuse-type, 29% mixed-type, and 50% intestinal-type gastric cancer patients; while BPRC cohort was composed of 100% diffuse-type gastric cancer patients. Therefore, the possible reason for no concordance between the FDGC and the BPRC subtypes might be BPRC subtype was only for diffuse-type gastric cancer (DGC); while FDGC subtype was for gastric cancer with multiple Lauren's types, including diffuse-type, intestinal-type, and mixed-type. Although no concordance in the composition of Lauren's type in two cohorts, the proteomic features associated with therapy response and prognosis revealed in FDGC subtypes were applicable in the BPRC cohort. In the comparison of FDGC and the BPRC subtypes, we found the PX3 subtype accounted for the highest proportion in G-IV subtype (50%) compared with other subtypes (20% in G-I, 26% in G-III, respectively), indicating PX3 subtype was mostly clustered into G-IV subtype (**Fig.3a and Supplementary Fig. 4a in the revision**). As reported in BPRC subtype, PX3 subtype was characterized with the enrichment of pathways including ECM organization, EMT, and complement, had the worst prognosis and was resistant to chemotherapy. Consistently, in our study, G-IV subtype was characterized by ECM-receptor interaction, focal adhesion, and complement/coagulation cascades, had the worst prognosis and highest proportion (80%) of non-sensitive patients. Collectively, these results demonstrated the robustness of our proteomic subtyping system related to therapy response.

As for ACRG cohort, we also surveyed the composition of Lauren's type. As a result, we found ACRG cohort included diffuse-type, intestinal-type, and mixed-type, similar to FDGC cohort. As concluded above, BPRC cohort was composed of 100% diffuse-type gastric cancer patients. Therefore, the composition of Lauren's type in FDGC cohort and ACRG cohort was similar, but the composition of Lauren's type in FDGC cohort and BPRC cohort was different. Under this background, we applied FDGC subtyping algorithm in BPRC cohort and ACRG cohort separately, and obtained novel subtypes corresponding to these two cohort (named FDGC subtypes in BPRC cohort and ACRG cohort, respectively). After comparison of FDGC subtypes and BPRC subtype (previously published) or ACRG subtype (previously published), we found

FDGC subtype had no association with BPRC subtype ($P = 0.42$) (**Supplementary Fig. 4a in the revision**), while had significant association with ACRG subtype ($P = 5.0E-4$) (**Fig.3b in the revision**), demonstrating the concordance of the FDGC subtype with ACRG subtype, but not BPRC subtype. As reviewer mentioned, FDGC and the BPRC subtypes are based on proteomics profiling, whereas the ACRG subtype is based on mRNA profiling. Taking this into consideration, we speculated that the composition of Lauren's type has more significant contribution in molecular subtyping.

(2) About the application of FDGC subtype in EOGC cohort

According to reviewer's comments, we explored the proteomic data from a research related to early-onset gastric cancers (EOGC) published on Cancer Cell by Mun *et al.* in 2019 (Cancer Cell, 2019, PMID: 30645970) ⁸. In the research, 80 patients with EOGCs (EOGC cohort) were included for proteogenomic analysis. Consistent to FDGC cohort, the EOGC cohort included diffuse-type, intestinal-type, and mixed-type. According to reviewer's suggestions, we further applied FDGC subtyping algorithm in EOGC cohort. The EOGC cohort was clustered into four subtypes based on the signature proteins of FDGC subtype, as expected (**Figure RL13a**). As shown in **Figure RL13b and c (see also Fig. 3a and Supplementary Fig. 4b in the revision)**, the Sankey plot illustrated significant concordance between FDGC subtype (G-I to G-IV) and EOGC subtype (Sub1 to Sub4) (Chi-Squared Test, $P = 8.5E-9$). After further comparison of FDGC subtype and EOGC subtype, we observed the Sub2 accounted for most (75%) of G-I subtype. As reported in the research related to EOGC subtype, the Sub2 had the best survival. Consistently, in our study, the G-I had the best survival, which indicated the similar survival patterns between FDGC subtype and EOGC subtype. These results verified reviewer's comment, demonstrating the robustness of our proteomic subtyping in other GC cohorts based on proteomics profiling. We thank the reviewer for the constructive suggestion again. In the revision, the application of FDGC subtype in EOGC cohort was shown in **Fig. 3 and Supplementary Fig. 4**. We have updated the supplementary description in the part of "Result" (**line 322–356 in Page 12–13**) and "Discussion" (**line 816–824 in Page 30**) in the revision.

Figure RL13. The comparison of FDGC subtype and EOGC subtype. **a** Clustering of EOGC proteomic data by the signature proteins derived from FDGC subtypes. **b, c** Sankey diagram and barplot indicating the comparison of FDGC subtype and EOGC subtype.

Q15. There are a couple problems with the construction and validation of predictive classifiers (Figure 6). First, feature selection (DEP in Figure 6a) should be performed with the training set only to avoid overfitting. Second, if I understand correctly, the validation cohort in Figure 6c is not really an independent validation cohort, most of the samples are from the discovery cohort. This is very misleading. Validation cohort should not reuse samples from the discovery cohort. Validation result from the BPRC cohort

(assuming there is no sample overlap between the BPRC and the FUDAN cohort) is encouraging. However, due to the small sample size of both the discovery and validation cohorts, it would be important to publish the final model so that it can be tested in the future when new datasets are available. Moreover, to further demonstrate the value of proteomics data in predictive modeling, the authors should check whether drug sensitivity can be accurately predicted using other clinical attributes.

Response to Q15:

Thanks for the suggestions of construction and validation of predictive classifiers.

Firstly, we confirmed that feature selection was performed within the training set, as reviewer mentioned. Secondly, we apologized for the sample reuse from the discovery cohort for validation. Thirdly, the BPRC cohort was from a publication titled with “A proteomic landscape of diffuse-type gastric cancer” in *Nature Communications* (*Nat Commun*, 2019, PMID: 29520031)⁹. There was no sample overlap of gastric cancer patients between the BPRC cohort and our FUDAN cohort.

At last, for validating the predictive models, we constructed a new cohort composed of 60 GC patients receiving either DOS (N = 20), XELOX (N = 20), or anti-HER2 (N = 20) therapies. All samples were histologically scored by two expert gastrointestinal pathologists (C.X. and Y.H) according to the widely accepted Response Evaluation Criteria in Solid Tumors (RECIST) (version1.1) by CT/MRI scanning and grouped into complete response (CR), partial response (PR), stable disease (SD), or progressive disease (PD). In this study, patients with CR and PR were defined as sensitive (S) and those with SD and PD were defined as non-sensitive (NS). The independent cohort was grouped into DOS-sensitive group (DSG, N = 10), DOS-non-sensitive group (DNSG, N = 10), XELOX-sensitive group (XSG, N = 10), XELOX-non-sensitive group (XNSG, N = 10), HER2-sensitive group (HSG, N = 10), and HER2-non-sensitive group (HNSG, N = 10) (**Figure RL14a, see also Supplementary Data 11 in the revision**).

We employed a targeted MS approach, parallel reaction monitoring (PRM) in the revision, which has been adopted in classifier's validation in recent proteomic research (*Cell*, 2020, PMID: 32795414; *J Extracell Vesicles*, 2020, PMID: 32363013)^{3,4}. During the PRM, we selected a set of target peptides that unique to these proteins, including DSG/DNSG-sig (ATP5S, C11orf31, CDC42SE2, CHP2, and AHR), XSG/XNSG-sig (RFC2, NIT1, RAB32, FLG2, FNBP1, GCLC, DYNLRB1, RBBP7, LPXN, LMAN2, NUB1, WAS, FAM82B, and MYCBP), and HSG/HNSG-sig (CAPN5, BAIAP2, SRPX2, COMMD4, SCIN, DSC2, SEPSECS, TECPR1, DDX60L, NPL, SLC39A4, and IRF6) using the library search results (**Table RL3, see also Table S1 in the revision**). Besides, house-keeping proteins, such as VCP, RPLP0, PSMB4, were also included for the reference. Followed by this, PRM strategy was designed to quantify these signature proteins in FFPE tumor tissues from the independent cohort. The fragment total areas of targeted peptides reported by Skyline-daily (4.2.1.19004, University of Washington, USA) were used for protein quantification.

Based on the PRM quantification, we performed comparative analysis of signature proteins (including DSG/DNSG-sig, XSG/XNSG-sig, and HSG/HNSG-sig) between DSG and DNSG, XSG and XNSG, HSG and HNSG, respectively. As a result, we observed the significantly differential expression of these signature proteins between DSG and DNSG, XSG and XNSG, HSG and HNSG, respectively. In DOS subcohort, we observed signature proteins, including AHR, ATP5S, C11orf31, CDC42SE2, and CHP2, had at least 2-fold differences between DSG and DNSG ($P < 0.05$, Wilcoxon rank-sum test). In XELOX subcohort, XSG/XNSG-sig (RFC2, NIT1, RAB32, FLG2, FNBP1, GCLC, DYNLRB1, RBBP7, LPXN, LMAN2, NUB1, WAS, FAM82B, and MYCBP) were significantly increased in XNSG compared with XSG (Fold change (XNSG/XSG) > 2 , $P < 0.05$, Wilcoxon rank-sum test). In HER2 subcohort, we observed HSG/HNSG-sig (CAPN5, BAIAP2, SRPX2, COMMD4, SCIN, DSC2, SEPSECS, TECPR1, DDX60L, NPL, SLC39A4, and IRF6) had a more than 2-fold increase in HNSG compared with HSG (Fold change (HNSG/HSG): 2.17, 2.22, 4.44, 5.43, 5.71, 4.60, 5.27, 5.03, 4.06, 2.18, 5.40, 3.35; $P < 0.05$, Wilcoxon rank-sum test) (**Figure RL14b, d, and f, see also Supplementary Fig. 6b, c, d in the revision**). In addition, the heatmaps showed a clear separation between DSG and DNSG, XSG and XNSG, HSG and HNSG in the new independent cohort, respectively

(Figure RL14c, e, and g, see also Fig. 6c in the revision). Overall, predictive power of the signature proteins in different therapies (including DOS, XELOX, and anti-HER2) was validated in a new independent cohort by PRM assays. Finally, the PRM-MS-based proteomics data have been deposited to the iProX repository and available via the same ProteomeXchange ID PXD024255 and project ID IPX0002116000.

As for clinical attributes for predicting drug sensitivity, there was none clinical attributes used for predicting response of chemotherapies (DOS and XELOX) and anti-HER2 targeted therapy after surveying clinical and biological researches so far. Therefore, it would be of great importance to develop models with high accuracy for predicting therapeutic response through mass spectrometry (MS)-based high throughput proteomic strategy, thus promoting the development of personalized medicine.

Finally, in the revision, we have added the descriptions in the part of “Construction and validation of the predictive models for GC chemo- and targeted therapies” in the “Result” (**line 619–645 in Page 23–24**). Meanwhile, we have updated the description of “Targeted PRM analysis” in the “Method” (**line 1410–1438 in Page 51–52**).

Figure RL14. The validation of DSG/DNSG-sig, XSG/XNSG-sig, and HSG/HNSG-sig by Targeted MS analysis in an independent cohort of 60 GC patients. **a** An independent cohort for PRM validation. **b, d, and f** Boxplot showing the differential expression (Log₁₀-transformed) of the DSG/DNSG-sig (**b**), XSG/XNSG-sig (**d**), and HSG/HNSG-sig (**f**). *P* values calculated by Wilcoxon rank-sum test. **c, e, and g** Heatmaps showing DSG/DNSG-sig (**c**), XSG/XNSG-sig (**e**), and HSG/HNSG-sig (**g**) validated by targeted MS in an independent cohort of 60 GC patients.

Table RL3. Targeted peptides that unique to signature proteins

Group	Protein	Sequence	m/z [Da]
a. DSG_DNSG-sig model	AHR	LASLLPFPQDVINK	777.94803
a. DSG_DNSG-sig model	AHR	NDFSGEVDVFR	593.2622
a. DSG_DNSG-sig model	AHR	SFFDVALK	463.75358
a. DSG_DNSG-sig model	ATP5S	DYNHLPTGPLDK	457.23033
a. DSG_DNSG-sig model	ATP5S	TALPSLELK	486.29223
a. DSG_DNSG-sig model	C11orf31	AEAADVVAEAK	529.29855
a. DSG_DNSG-sig model	C11orf31	FPEPQEVVEELK	722.37176
a. DSG_DNSG-sig model	C11orf31	LEAPELPVK	498.29255
a. DSG_DNSG-sig model	C11orf31	NAAALSQALR	507.78859
a. DSG_DNSG-sig model	CDC42SE2	GGYGGGMPANVQMQLVDTK	961.95833
a. DSG_DNSG-sig model	CHP2	IIESFFPDGSQR	698.34931
b. XSG_XNSG-sig model	DYNLRB1	GVQGIIVNTEGIPIK	818.98537
b. XSG_XNSG-sig model	DYNLRB1	STMDNPTTTQYASLMHSFILK	1193.5759
b. XSG_XNSG-sig model	DYNLRB1	DIDPQNDLTFLR	723.86482
b. XSG_XNSG-sig model	FAM82B	LLVYEALEYAK	656.36311
b. XSG_XNSG-sig model	FAM82B	QIQTEAAQLLTSFSEK	897.46675
b. XSG_XNSG-sig model	FAM82B	ESEDAELLWR	624.29973
b. XSG_XNSG-sig model	FLG2	NPDDPDTVDMHMLDR	991.95196
b. XSG_XNSG-sig model	FLG2	SVVTVIDVFYK	635.35899
b. XSG_XNSG-sig model	FNBP1	FEAWLAEVEGR	653.82494
b. XSG_XNSG-sig model	FNBP1	TEIELSYAK	527.27613
b. XSG_XNSG-sig model	FNBP1	ADYSSILQK	512.76921
b. XSG_XNSG-sig model	GCLC	SLFFPDEAINK	640.829
b. XSG_XNSG-sig model	GCLC	DPLTLFEELK	1091.56403
b. XSG_XNSG-sig model	GCLC	YDSIDSYLSK	595.7828
b. XSG_XNSG-sig model	LMAN2	LTVMTDLEDK	1164.58275
b. XSG_XNSG-sig model	LMAN2	LPTGYFFGASAGTGDLSDNHDIIISMK	910.76176
b. XSG_XNSG-sig model	LMAN2	DNVDDPTGNFR	625.27584
b. XSG_XNSG-sig model	LPXN	TSAAAQLDELMAHLTEMQAK	720.35421
b. XSG_XNSG-sig model	LPXN	DFLAMFSPK	528.2661
b. XSG_XNSG-sig model	MYCBP	HHLGAATPENPEIELLR	633.0017
b. XSG_XNSG-sig model	MYCBP	VLVALYEEPEKPNALDFLK	1138.11934
b. XSG_XNSG-sig model	MYCBP	SGVLDLTK	467.26625
b. XSG_XNSG-sig model	NIT1	LLEEYQLAR	618.33505
b. XSG_XNSG-sig model	NIT1	DPAETLHLSEPLGGK	782.40285
b. XSG_XNSG-sig model	NIT1	IDLNYLR	453.75561
b. XSG_XNSG-sig model	NUB1	IAETFGLQENYIK	763.39894
b. XSG_XNSG-sig model	NUB1	VDNLLQLGFTAQEAR	837.94435
b. XSG_XNSG-sig model	NUB1	TTGIATIEVFLPPR	757.93262
b. XSG_XNSG-sig model	RAB32	ATIGVDFALK	517.79737
b. XSG_XNSG-sig model	RAB32	VLVIGELGVGK	542.34155
b. XSG_XNSG-sig model	RAB32	DNINIEEAAR	572.78302
b. XSG_XNSG-sig model	RBBP7	EMFEDTVEER	642.77431
b. XSG_XNSG-sig model	RBBP7	GEFGGFGSVTGK	571.77703
b. XSG_XNSG-sig model	RFC2	LNEIVGNEDTVSR	723.36492
b. XSG_XNSG-sig model	RFC2	EGNVPNIIAGPPGTGK	817.45021
b. XSG_XNSG-sig model	RFC2	LTDAILTR	515.79832

b. XSG_XNSG-sig model	WAS	LIYDFIEDQGGLEAVR	919.47159
b. XSG_XNSG-sig model	WAS	SGPLPPVPLGIAPPPPTPR	620.69603
b. XSG_XNSG-sig model	WAS	GAPAVQQNIPSTLLQDHENQR	1158.58841
c. HSG_HNSG-sig model	BAIAP2	ELGDVLFQMAEVHR	548.61244
c. HSG_HNSG-sig model	BAIAP2	EGDLITLLVPEAR	713.40058
c. HSG_HNSG-sig model	BAIAP2	GYFDALVK	456.74482
c. HSG_HNSG-sig model	CAPN5	DFFFQNPQYIFEVK	888.43511
c. HSG_HNSG-sig model	CAPN5	ADPDNLQALHTLHLR	857.45585
c. HSG_HNSG-sig model	CAPN5	GENLAIGFDIYK	670.34826
c. HSG_HNSG-sig model	COMMD4	HSVDGESLSSELQQLGLPK	675.34723
c. HSG_HNSG-sig model	COMMD4	VDYTLSSSLLQSVVEEPMVHLR	801.7451
c. HSG_HNSG-sig model	COMMD4	FQVLLAELK	530.82347
c. HSG_HNSG-sig model	DDX60L	DLSIAVQMMK	568.29536
c. HSG_HNSG-sig model	DSC2	LTDPTGWWTIDENTGSIK	973.98939
c. HSG_HNSG-sig model	DSC2	VIPDDLAQQNLIVSNTEAPGDDK	1226.61809
c. HSG_HNSG-sig model	DSC2	GPGVDQEPR	477.73553
c. HSG_HNSG-sig model	IRF6	LQISTPDIKDNIVAQLK	948.54465
c. HSG_HNSG-sig model	NPL	AEELLDGILDK	608.32642
c. HSG_HNSG-sig model	NPL	DILINFLK	488.29747
c. HSG_HNSG-sig model	SCIN	IQEGEPEEFWNSLGGK	974.95191
c. HSG_HNSG-sig model	SCIN	ATEVPLSWDSFNK	747.36458
c. HSG_HNSG-sig model	SCIN	DYQTSPLETTQAEDHPPR	1048.99899
c. HSG_HNSG-sig model	SEPSECS	ITNSLVLDIIK	614.87919
c. HSG_HNSG-sig model	SLC39A4	ADLVAEESPELLNPEPR	939.97643
c. HSG_HNSG-sig model	SRPX2	EQQLSANIIEELR	771.90974
c. HSG_HNSG-sig model	TECPR1	LVTSGPWLEVPPIALR	874.50847

Q16. The final proteomics data table (supplementary Data 1c) includes 11287 genes, but 1754 genes (16%) were identified in only a single sample, 5316 (47%) were identified in less than 10% of the samples, and 7683 (68%) were identified in less than 50% of the samples. It is not clear whether all proteins in this table are used for downstream data analysis. It is well known that both detection variation and quantification variation are high for proteins with low abundance, and proteins that cannot be robustly quantified by the platform should be excluded from downstream analyses. The authors should define and justify which proteins are considered “quantifiable”. This is important both for this study and for future reuse of the data by the research community.

Response to Q16:

Thanks for the comment. In this study, not all identified proteins were used for downstream data analysis. Firstly, during the subsequent data analysis, the up-regulated proteins of each group should first meet the criteria with expression frequency of more than 10% patients in the group. Secondly, the average frequency of signature proteins identified in each subtype was 56%

(Figure RL15a). We surveyed these proteins in corresponding subtypes in BPRC cohort. As a result, these signature proteins of subtypes showed a similar average frequency (59%) of corresponding subtypes in BPRC cohort **(Figure RL15b)**. That means these signature proteins could be validated in other GC cohorts, indicating their robust quantification. In addition, among 2,279 signature proteins identified in four proteomic subtypes, 1679 signature proteins (74%) have immunohistochemistry (IHC) staining data as reported by The Human Protein Atlas (HPA). Among these signature proteins, 1,467 (87.4%) signature proteins showed medium to high tumor-specific staining in GC samples **(Figure RL15c, also see Supplementary Fig.3a in the revision)**. Thirdly, the average frequency of signature proteins identified in predictive models for DOS, XELOX, and HER2 subcohorts was as high as to 60% **(Figure RL15d)**. Among 26 signature proteins (81.2%) identified in predictive models with IHC staining data in HPA, 20 (77.7%) signature proteins were evaluated with medium to high intensity in tumor-specific staining images in GC samples **(Figure RL15d)**. The representative IHC images for these signature proteins of predictive models were shown in **Figure RL15e (Supplementary Fig.6f in the revision)**. Furthermore, these signature proteins were also validated by parallel reaction monitoring (PRM) assays in an independent cohort in the revision. Based on PRM quantification, these proteins were identified in 100% samples, which demonstrated the robustness and application of these signature proteins. Therefore, the quantifications of signature proteins nominated in this study were verified by immunohistochemistry (IHC) staining data as reported by HPA in GC samples, as well as in proteomic data in other independent GC cohorts, demonstrating these results could be reused and validated in the future research community. Finally, in the revision, we have updated these results in **Supplementary Fig.3a** and **Supplementary Fig.6f**, added these descriptions in the “Discussion” **(line 798–803 in Page 29, line 894–906 in Page 32–33)**.

Figure RL15. The identification frequency of signature proteins of proteomic subtypes and predictive models. **a, b** Frequency distribution histogram showing the mean values of signature proteins of G-I to G-IV in FDGC cohort (**a**) and BPRC cohort (**b**). **c** These signature proteins of proteomic subtypes were mostly validated by Human Protein Atlas (HPA) Immunohistochemistry (IHC) Staining Data. **d** Frequency distribution histogram showing the

mean value of signature proteins of predictive models. **e** These signature proteins of predictive models were mostly validated by HPA IHC Staining Data. Representative IHC images for these signature proteins of predictive models.

Q17. The authors should provide clinical data for individual patients, including all features in Table 1 and treatment response data, in a supplementary table to enable effective reuse of the dataset.

Response to Q17:

Thanks for the comment. In the revision, the detailed clinical characteristics for individual patients were provided in **Supplementary Data 1**, which could be effectively reused. In addition, tumor purity of each sample was added in **Supplementary Data 1**.

Q18. Figure 1d is confusing. First, there are 12,519 proteins but the largest rank is below 10,000. Second, some proteins with relatively higher intensity ranked higher than proteins with relatively lower intensity. For example, the protein with the second lowest intensity ranked at around 7000 rather than at the very end.

Response to Q18:

Thanks for the comment. We apologized for the confusion caused by the inconsistent description related to figure presentation. In detail, Fig. 1d showed the dynamic range of the protein identification of each sample, while the upper description (12,519 proteins) of the figure was protein accumulative identification of 206 samples. Proteomics measurement resulted in 6,369–8,119 proteins in each sample. In Fig. 1d, the maximum limitation (14,000) of x axis was set according to the protein identification in total (12,519 proteins). However, the dynamic range was plotted based on the protein identification of each sample in Fig. 1d. In the revision, we firstly presented the dynamic range of protein identification of each sample according to the descending sort of protein abundance with a range of 6,369–8,119 proteins identified in each sample. We observed the maximum rank number of each sample belonged to a range of 6,369–8,119, which was close to the maximum limitation of x axis (narrowed to 9,000) in the revised

Fig. 1d. Meanwhile, we updated the description as “Range: 6,369–8,119 proteins” on the top of the curve (**Figure RL16a, see also Fig. 1d in the revision**). In addition, the dynamic range of proteins accumulate identified in 206 samples were shown in **Figure RL16b**. The total protein identification number 12,519 was nearly closed to the maximum limitation (14,000) of x axis.

Secondly, the dynamic range of protein identification of each sample was plotted according to the descending sort of protein abundance in this sample. That means proteins with higher intensity had higher rank. For example, we define protein with highest intensity as highest rank (rank number 1), and defined protein with second highest intensity as rank number 2, and so on. Each point represents the rank and intensity of each protein in each sample, which make up a line of smooth curve. In this figure, we presented the dynamic range of proteins identified in each sample of all 206 samples, and plotted 206 lines of smooth curves. In one curve (corresponding to one sample), the protein with lowest intensity had the maximum rank number, which represents the maximum identification number in one sample. As reviewer mentioned, the protein with the second lowest intensity in **Figure RL16a** represents the maximum identification number (7,118) of sample GC-40, indicating the lowest intensity of this sample; the protein with the lowest intensity represents the maximum identification number (8,119) of sample GC-116. Based on the average protein abundance of proteins totally identified in 206 samples, we concluded proteins with top five intensities were HBB, HBA1/2, AGR2, and PFN1; and proteins with last five intensities were ZFP28, ZSCAN20, ZNF236, BCORL1, and ZNF536. Here, to be more clearly, we have updated a more detailed description related to this figure in the “Figure legend” and “Methods” as following: (1) “Shown are the dynamic range of protein identification of each sample according to the descending sort of protein abundance in this sample.” (**line 960–964 in Page 35**). (2) “The dynamic range of protein identification of each sample was shown according to the descending sort of protein abundance with a range of 6,369–8,119 proteins identified in each sample. The protein with highest intensity has the minimum rank number, representing the highest rank; the protein with lowest intensity has the maximum rank number, representing the maximum identification number in one sample.” (**line 1293–1297 in Page 47**).

Figure RL16. Overview of the proteomic profile of patients with GC. Shown are the dynamic range of the protein identification of each sample (a), and protein accumulation of 206 samples (b). Range: 6,369–8,119 proteins, Total: 12,519 proteins. Proteins were quantified as a normalized intensity-based fraction of total (FOT) value and log10 transformed. The highest- and lowest-abundance proteins are shown in the box.

Q19. Page 8, line 214, what does frequency >10% mean?

Response to Q19:

Thanks for the comment. As described in the manuscript, “Comparative analysis of proteomic profiling resulted in 301 (G-I), 611 (G-II), 925 (G-III), and 467 (G-IV) GPs ($P < 0.05$; fold change > 2 ; frequency $\geq 10\%$)”, of which “frequency $\geq 10\%$ ” was one of the criteria for the signature proteins defined in each subtype, meaning detected in at least 10% patients of the subtype. In the revision, we updated the description in “Proteomic analysis of the gastric cancer cohort” in the part of “Result” (Please see **line 242–243 in Page 9**).

Reviewer #3, expertise in gastric cancer subtypes and therapy (Remarks to the Author):

Comments

In this manuscript, Li et al. generated an impressive proteomic dataset consisting of 206 treatment-naïve FFPE tumour tissues from GC patients who have received either triplet combination chemotherapy DOS (docetaxel, oxaliplatin and S-1), doublet chemotherapy XELOX (capecitabine and oxaliplatin) or anti-HER2-based therapy (trastuzumab).

Proteomic-based classification identified 4 subtypes with different therapeutic clinical responses and molecular features. Using differential proteomics analysis between therapy-sensitive and non-sensitive groups, the authors explored key pathways that predict and underlie therapeutic resistance, including activation of immune, ECM and PI3K-Akt pathways. They also developed prognostic models to predict chemotherapeutic responses. Finally, the authors highlighted 2 key proteins (CTSE and TKTL1) that may regulate chemo-sensitivity, supported with extensive relevant in vitro experiments.

Overall, given that current large-scale proteomic information of GC remains mostly limited to diffuse GCs and pre-cancerous gastric lesions, this paper provides a highly relevant and novel dataset that will certainly be of interest to the GC community. This paper also seeks to address a scientifically important question of drug resistance predictors and mechanisms, and provides a valuable resource that can enrich our understanding. The data analysis and experiments performed were also generally well planned and executed, with clear descriptions of findings and figures.

Response:

We thank the reviewer for the careful read and thoughtful comments on previous manuscript. According to reviewer's comments, we made corresponding revision to address all the points. The reviewer's comments were summarized as following: (1) about the differential xCell score of cell types and differential expression of proteins presented in heatmaps; (2) about the importance of proteomic-based classification in clinic; (3) about the analysis of therapy response within proteomic subtypes; (4) about the difference of MSI/MSS-sig score associated with response to different treatments; (5) about the validation of synergistic effects of Herceptin with XELOX, or PI3K-AKT inhibitor *in vitro*; (6) about the direct addressment of the result that GC patients with high TCR signalling are unlikely to benefit from XELOX, but instead respond to XELOX + anti-HER2; (6) about the discussion of anti-HER2 therapy resistance. Finally, we made a point to point response and updated these results in the revision.

Major Comments

Q20. Many figures employ the use of heatmaps to show differential expression of genes (e.g. Fig 3f, 3k, 4m,5h) or differential enrichment of cell types (e.g. Fig 4e, 4f). However, not all the genes/cell types that are listed are significantly differential (since they are not marked with *). This makes it difficult to interpret and also begs the question of how the genes/cell types listed in the figure are being selected. The authors should consider for e.g. putting exact p-values for each gene/cell type listed so readers can be clear about the difference between groups compared.

Response to Q20:

Thanks for the comment. We apologized for the unclear presentation. In the study, to explore the association of tumor immune microenvironment with therapy response, we evaluated the abundance of 64 different cell types via xCell based on proteomic profiles (*Genome Biol*, 2017, PMID: 29141660)¹⁰. According to the reviewer's comment, we updated these heatmaps, all proteins or cell types presented in which were significantly differential and marked with * in the revision (**Figure RL17a, b, c, d, e, and f, also see Fig.3f, 3k, Fig. 4e, 4f, supplementary Fig. 5f, and Fig. 5h in the revision**). The p-values less than 0.05, 0.01, 0.001, 0.0001 were marked with *, **, ***, ****, as presented in heatmap, respectively. The corresponding detailed descriptions were revised in the "Figure Legends" sections in the manuscript. According to reviewer's suggestion, we also have provided exact p values in the **Supplementary Data 5, 6, 7, 8, and 9 in the revision**.

Figure RL17. Heatmaps related to differential proteins or cell types. **a** Heatmap illustrating down-regulated proteins in G-IV compared with other subtypes. $*P < 0.05$ was considered statistically significant. $*P < 0.05$, $**P < 0.01$, $***P < 0.001$, $****P < 0.0001$. **b** The abundance of Macrophages M1 markers in MSI/MSS-sig high and MSI/MSS-sig low groups. $*P < 0.05$ was

considered statistically significant. **c, d** Heatmap illustrating cell type compositions between sensitive and non-sensitive groups of DOS and XELOX subcohorts (**c**), and XELOX subcohort of BPRC cohort (**d**). $*P < 0.05$ was considered statistically significant. **e** Heatmap illustrating the protein abundance related to antigen processing and presentation and CD8+ Tcm markers (xCell signature). $*P < 0.05$ was considered statistically significant. **f** Heatmap illustrating the protein abundance of apoptosis related proteins. $*P < 0.05$ was considered statistically significant.

Q21. The utility of performing unsupervised proteomic-based classification to reveal 4 subtypes (as shown in Fig2) is not quite apparent. After all, the main aim seems to be identifying predictors/mechanisms of resistance, and thus a direct comparison of sensitive vs. non-sensitive groups would be most useful (as done in Fig4/5). If the authors choose to delineate 4 subtypes G-I to G-IV as in Fig2, it would be nice if they could then further discuss and analyse differences in predictors/mechanisms of resistance for each subtype since each has its own share of non-responders. For e.g. Fig 2e-2g showing high expression of ECM and its association with drug resistance is likely to only apply for group G-IV. What about G-I, G-II and G-III?

Response to Q21:

Thanks for the comment. According to the reviewer's comments, we responded into two parts as follows: (1) the reason and utility of proteomic-based classification in this study; (2) the further discuss and analysis differences in predictors/mechanisms of resistance for each subtype.

(1) About the reason and utility of proteomic-based classification

In this study, we constructed a GC cohort that covered three clinical therapy subcohorts, including DOS subcohort, XELOX subcohort, and HER2 subcohort, which represented three therapy regimens (DOS, XELOX, and anti-HER2 therapies, respectively). For presenting a comprehensive molecular landscape for predicting the therapeutic response in GC, we performed proteomic-based classification before direct comparison of sensitive vs. non-

sensitive groups in each therapy subcohort. Here, we summarized the key findings as follows, which was the unique conclusions based on proteomic-based classification, not direct comparison.

1) We observed significant distribution difference of therapy subcohorts among proteomic subtypes, which was mainly derived from HER2 subcohort (mainly enriched in G-II subtype); while no significant difference was observed in DOS and XELOX subcohorts (**Figure RL18a–c, also see Supplementary Fig. 2e**). We speculated the distribution difference among proteomic subtypes could be caused by the specific proteomic feature related to HER2 expression, due to the selection of HER2-positive patients for anti-HER2 targeted therapy. Then, we evaluated HER2 expression detected by immunohistochemistry (IHC) and FISH analysis. Any case with IHC 3+ or IHC 2+/FISH+ was considered to be HER2-positive, while cases with IHC 0 or IHC 1+ or IHC 2+/FISH- were considered as HER2-negative (**Supplementary Data 1 and Table 1**). As expected, we observed the higher proportion (71%) of HER2-positive patients in G-II subtype compared with other subtypes (G-I: 21%, G-III: 43%, G-IV: 40%) (**Figure RL18d, also see Supplementary Fig. 2e**). This result demonstrated the specific molecular pattern in HER2-positive GC patients, which could be identified at proteome level. As for chemotherapies (DOS and XELOX therapy), there was no distribution difference among proteomic subtypes, due to no selection of patients prior to therapy. These results supported the reliability and importance of proteomic subtyping of this study.

2) In our previously published research, we constructed the BPRC (Beijing Proteome Research Center) DGC (diffuse-type gastric cancer) cohort, which was classified into three groups (PX1–3) based on the proteomic profile. In addition, as reported, other molecular classifications have been proposed, such as EOGC subtypes (Early-Onset Gastric cancer) (Cancer Cell, 2019, PMID: 30645970) and ACRG subtypes (Asian Cancer Research Group) (*Nat Med*, 2015, PMID: 25894828). In this study, we performed proteomic-based classification and identified four proteomic subtypes, which could be linked with other published GC subtypes (as mentioned above) (**Fig. 3a, b, and Supplementary Fig. 4a, b in the revision**). Based on the unique features of our proteomic subtypes associated with therapeutic response, we could assess the

possibility of response to specific therapy in other cohorts (although there was no exact therapy response).

3) Through comparisons between our proteomic subtypes and other published subtypes (ACRG subtypes), we observed MSS/EMT subtype originated from ACRG subtyping accounted for largest proportion (56%) of the G-IV subtype (**Fig. 3b in the revision**). Further evaluation of microsatellite instability (MSI) revealed that G-IV subtype was characterized with lower MSI/MSS-sig score, while higher MSI/MSS-sig score was enriched in the G-II subtype (**Supplementary Fig. 4c in the revision**). Furthermore, we found the association of microsatellite instability with tumor immune microenvironment, thus affecting the clinical outcome of GC patients with chemo-/targeted therapies (**Fig. 3f–m in the revision**). Based on this finding, we concluded that MSI/MSS-sig level showed significant association with the response to chemo- and targeted therapy.

Overall, the key findings concluded from proteomic-based classification deepened our understanding of the landscape included different therapy regimens, provided the opportunity of linking our subtypes with other published subtypes produced in other independent cohorts, and important clues for us to validate in the further comparative analysis. Moreover, after proteomic-based classification, we indeed performed direct comparison of sensitive vs. non-sensitive groups in each therapy subcohort. Furthermore, through analysis of the association between MSI/MSS-sig level and clinical outcomes in different therapies, we found MSI-sig high GC patients could benefit from DOS therapy, but not XELOX therapy.

(2) About the further discuss and analysis differences in predictors/mechanisms of resistance for each subtype

As shown in the pie charts, we observed different proportions of sensitive patients (S) and non-sensitive patients (NS) in each subtype: 60.71% S (N = 17) and 39.29% NS (N = 11) in G-I subtype; 46.43% S (N = 26) and 50% NS (N = 28) in G-II subtype; 42.67% S (N = 32) and 56% NS (N = 42) in G-III subtype; 20% S (N = 4) and 80% NS (N = 16) in G-IV subtype (**Figure RL18e**). In this study, we concluded that G-IV was characterized by ECM-receptor interaction,

and the activation of ECM-receptor interaction pathway was associated with drug resistance. According to the reviewer's suggestion, to explore whether high expression of ECM and its association with drug resistance was applied for G-I, G-II and G-III subtypes (besides G-IV subtype), we performed comparative analysis between S and NS patients in each subtype, respectively. GSEA analysis revealed that ECM pathway was significantly enriched in NS patients in G-I (NES = 1.77, $P = 0.006$), G-II (NES = 1.51, $P = 0.023$) and G-III (NES = 1.51, $P = 0.027$) subtypes (**Figure RL18f, see also Supplementary Fig. 3h in the revision**), indicating the consistent association of ECM pathway and drug resistance in different proteomic subtypes.

In addition, G-I subtype, with highest proportion (61%) of sensitive patients, was featured by endocytosis pathway. We further analyzed the association of endocytosis pathway with clinical outcomes in the revision. ssGSEA analysis revealed that the activation of endocytosis pathway indicated a better prognosis (log rank test, $P < 0.05$) (**Figure RL18g, see also Supplementary Fig. 3d in the revision**). We then explored the association of proteins involved in endocytosis bioprocess with prognosis. As shown in **Figure RL18h (see also Supplementary Fig. 3e in the revision)**, a group of endocytosis related proteins, such as DNM2, EPS15, WIPF1, ACAP2, and CHMP6, showed positive association with overall survival (hazard ratios range: 0.10-0.54, $P < 0.05$). Then, we performed GSEA analysis between S and NS in G-I subtype. As a result, we found endocytosis was enriched in S patients compared with NS patients in G-I subtype (NES = -1.61, $P = 0.005$). To explore whether activated endocytosis and its association with drug sensitivity was applied for G-II, G-III and G-IV subtypes (besides G-I subtype), we conducted the same GSEA analysis between S and NS patients in each subtype, respectively. Consistently, we observed endocytosis pathway was enriched in S patients in G-II (NES = -1.54, $P = 0.018$), G-III (NES = -1.64, $P = 0.005$) and G-IV subtypes (NES = -1.56, $P = 0.009$) (**Figure RL18i, see also Supplementary Fig. 3h in the revision**), indicating the significant association of endocytosis pathway and drug sensitivity.

Overall, proteomic-based classification identified high expression of ECM associated with drug resistance, while activation of endocytosis indicated drug sensitivity. We found the same resistance/sensitivity mechanisms in different proteomic subtypes. Finally, we have updated

these results in figures (Supplementary Fig. 3d, e, h) and in the revised manuscript (line 242–271 in Page 9–10).

Figure RL18. The enrichment of HER2 subcohort among four subtypes, and pathway enrichment of ECM and endocytosis pathways between S and NS groups in four subtypes. The difference of the percentage of each subtype in these three subcohorts. **a** The barplot for each subcohort among four subtypes. **b** The barplot for HER2 subcohort among four subtypes. **c** The barplot for DOS and XELOX subcohorts among four subtypes. **d** The barplot for HER2-positive status and HER2-negative status among four subtypes. **e** Pie charts showing the

distribution of sensitive patients and non-sensitive patients in each subtype. **f** GSEA enrichment showing ECM pathway was enriched in NS patients in four subtypes. **g** The Kaplan–Meier curves of endocytosis pathway score with OS. *P*-value was calculated by two-sided log rank test, with *P* < 0.05 considered statistically significant. **h** Left panel: The upregulation of endocytosis related proteins in G-I subtype compared with other subtypes (Student's *t* test, *P* < 0.05; fold change > 2). Right panel: the red lines indicate the overall survival hazard ratios of endocytosis related proteins, and the endpoints represent lower or upper of the 95% confidence intervals. *Cox *P*-value < 0.05, **Cox *P*-value < 0.01. **i** GSEA enrichment showing endocytosis pathway was enriched in S patients in four subtypes.

Q22. Following up on above point, do the authors foresee the proteomic subtyping system (G-I/II/III/IV) to have utility in the clinics? For e.g. different subtypes might have different resistance mechanisms, and hence require different ways to alleviate resistance?

Response to Q22:

Thanks for the reviewer's comments. In this study, we constructed a GC cohort that covered three clinical therapy subcohorts, including DOS subcohort, XELOX subcohort, and HER2 subcohort, which represented three therapy regimens (DOS, XELOX, and anti-HER2 therapies, respectively) (**Fig. 1**). Here, we firstly performed proteomic-based classification related to the three therapy subcohorts, and identified four proteomic subtypes (G-I to G-IV subtypes) (**Fig. 2**). **(1)** Based on this proteomic-based classification, we found the different distribution of GC patients receiving different therapy regimens and their response, which was featured by distinct proteome panels. Exactly, we observed significant distribution difference of therapy subcohorts among proteomic subtypes, which was mainly derived from HER2 subcohort (mainly enriched in G-II subtype); while no significant difference was observed in DOS and XELOX subcohorts (**Supplementary Fig. 2e**). The distribution difference among proteomic subtypes could be caused by the specific proteomic feature related to HER2 expression, due to the selection of HER2-positive patients for anti-HER2 targeted therapy, which was further validated by HER2 evaluation by IHC and FISH analysis (**Supplementary Fig. 2e**). This result demonstrated the

specific molecular pattern in HER2-positive GC patients, which could be identified at proteome level. As for chemotherapies (DOS and XELOX therapy), there was no distribution difference among proteomic subtypes, due to no selection of patients prior to therapy. These results supported the reliability and importance of proteomic subtyping of this study. **(2)** In addition, we could link our proteomic subtypes with other published GC subtypes (such as BPRC subtypes, EOGC subtypes, and ACRG subtypes) **(Fig. 3)**, and further assess the possibility of response to specific therapy in other cohorts (although there was no exact therapy response). **(3)** The comparison between our proteomic subtypes and ACRG subtypes revealed G-IV subtype was characterized with lower MSI/MSS-sig score, while higher MSI/MSS-sig score was enriched in the G-II subtype **(Supplementary Fig. 4c in the revision)**. Followed by this, we found the association of microsatellite instability with tumor immune microenvironment, thus affecting the clinical outcome of GC patients with chemo-/targeted therapies **(Fig. 3f–m in the revision)**. Based on this finding, we concluded that MSI/MSS-sig level showed significant association with the response to chemo- and targeted therapy. Furthermore, through analysis of the association between MSI/MSS-sig level and clinical outcomes in different therapies, we found MSI-sig high GC patients could benefit from DOS therapy, but not XELOX therapy. **(4)** As the reviewer mentioned in Q21, we also performed GSEA enrichment analysis between S and NS patients in G-I, G-II, G-III, and G-IV, respectively. As a result, we concluded ECM pathway was enriched in NS patients of G-I, G-II, G-III, and G-IV; while endocytosis was enriched in S patients of G-I, G-II, G-III, and G-IV **(Figure RL18f, I, see also Supplementary Fig. 3h in the revision)**, indicating the consistent association of ECM pathway or endocytosis with drug resistance or sensitivity in different proteomic subtypes. Furthermore, to validate the association between extracellular matrix proteins and resistance of drugs (5-FU, oxaliplatin, and docetaxel), we employed the targeted MS approach, PRM assays, to quantify these proteins in FFPE tumor tissues from patients receiving DOS therapy (triplet combination chemotherapy of 5-FU, oxaliplatin, and docetaxel) in a new independent cohort. Our data demonstrated the association between these ECM proteins and drug resistance was validated by PRM approach in an independent cohort. Collectively, our results illustrated the high expression of extracellular matrix proteins associated with drug resistance, and these extracellular matrix proteins could serve as indicators to predict chemotherapy response in clinic.

Next, we examined the functional differences between sensitive group (S) and non-sensitive group (NS) in DOS, XELOX, and HER2 subcohorts, respectively. The comparative proteomic analysis of DSG and DNSG, XSG and XNSG, HSG and HNSG, revealed the patients with high TCR signaling were unlikely to benefit from XELOX, but instead the combination of anti-HER2-based therapy; while the activation of ECM and the downstream PI3K-AKT pathway impaired the anti-tumor effect of trastuzumab. We performed further validation experiment to confirm the synergistic effects of the combination of trastuzumab with XELOX, or PI3K-AKT inhibitors *in vitro*. Furthermore, we employed stepwise logistic regression and developed prognostic models with high accuracy (≥ 0.89) to predict the therapeutic response.

To evaluate the accuracy of the predictive signatures for the chemotherapeutic response, we constructed a new cohort composed of 60 GC patients (50% sensitive and 50% non-sensitive patients) receiving either DOS (N = 20: DSG, N = 10; DNSG, N = 10), XELOX (N = 20: XSG, N = 10; XNSG, N = 10), or anti-HER2 (N = 20: HSG, N = 10; HNSG, N = 10) therapies (Supplementary Data 11). We selected a set of target peptides that unique to these proteins, including DSG/DNSG-sig (ATP5S, C11orf31, CDC42SE2, CHP2, and AHR), XSG/XNSG-sig (RFC2, NIT1, RAB32, FLG2, FNBP1, GCLC, DYNLRB1, RBBP7, LPXN, LMAN2, NUB1, WAS, FAM82B, and MYCBP), and HSG/HNSG-sig (CAPN5, BAIAP2, SRPX2, COMMD4, SCIN, DSC2, SEPSECS, TECPR1, DDX60L, NPL, SLC39A4, and IRF6) using the library search results (**Table RL4, also see Table S1 in the revision**). Followed by this, PRM strategy was designed to quantify these signature proteins in FFPE tumor tissues from the independent cohort. The fragment total areas of targeted peptides reported by Skyline-daily (4.2.1.19004, University of Washington, USA) were used for protein quantification. Based on the PRM quantification, we performed comparative analysis of signature proteins (including DSG/DNSG-sig, XSG/XNSG-sig, and HSG/HNSG-sig) between DSG and DNSG, XSG and XNSG, HSG and HNSG, respectively. As a result, we observed the significantly differential expression of these signature proteins between DSG and DNSG, XSG and XNSG, HSG and HNSG, respectively. In DOS subcohort, we observed signature proteins, including AHR, ATP5S, C11orf31, CDC42SE2, and CHP2, had at least 2-fold differences between DSG and DNSG (*P*

< 0.05, Wilcoxon rank-sum test). In XELOX subcohort, XSG/XNSG-sig (RFC2, NIT1, RAB32, FLG2, FNBP1, GCLC, DYNLRB1, RBBP7, LPXN, LMAN2, NUB1, WAS, FAM82B, and MYCBP) were significantly increased in XNSG compared with XSG (Fold change (XNSG/XSG) > 2, $P < 0.05$, Wilcoxon rank-sum test). In HER2 subcohort, we observed HSG/HNSG-sig (CAPN5, BAIAP2, SRPX2, COMMD4, SCIN, DSC2, SEPSECS, TECPR1, DDX60L, NPL, SLC39A4, and IRF6) had a more than 2-fold increase in HNSG compared with HSG (Fold change (HNSG/HSG): 2.17, 2.22, 4.44, 5.43, 5.71, 4.60, 5.27, 5.03, 4.06, 2.18, 5.40, 3.35; $P < 0.05$, Wilcoxon rank-sum test) (**Figure RL19b, d, and f, see also Supplementary Fig. 6b, c, d in the revision**). In addition, the heatmaps showed a clear separation between DSG and DNSG, XSG and XNSG, HSG and HNSG in the new independent cohort, respectively (**Figure RL19c, e, and g, see also Fig. 6c in the revision**). Overall, predictive power of the signature proteins in different therapies (including DOS, XELOX, and anti-HER2) was validated in a new independent cohort by PRM assays.

Finally, in the revision, we have added the descriptions in the part of “Construction and validation of the predictive models for GC chemo- and targeted therapies” in the “Result” (**line 619–645 in Page 23–24**), and “Discussion” (**line 742-762 in Page 27-28**). Meanwhile, we have updated the description of “Targeted PRM analysis” in the “Method” (**line 1410–1438 in Page 51–52**). The PRM-MS-based proteomics data have been deposited to the iProX repository and available via the same ProteomeXchange ID PXD024255 and project ID IPX0002116000.

Figure RL19. The validation of DSG/DNSG-sig, XSG/XNSG-sig, and HSG/HNSG-sig by Targeted MS analysis in an independent cohort of 60 GC patients. **a** An independent cohort for PRM validation. **b, d, and f** Boxplot showing the differential expression (Log₁₀-transformed) of the DSG/DNSG-sig (**b**), XSG/XNSG-sig (**d**), and HSG/HNSG-sig (**f**). *P* values calculated by Wilcoxon rank-sum test. **c, e, and g** Heatmaps showing DSG/DNSG-sig (**c**), XSG/XNSG-sig (**e**), and HSG/HNSG-sig (**g**) validated by targeted MS in an independent cohort of 60 GC patients.

Table RL4. Targeted peptides that unique to signature proteins

Group	Protein	Sequence	m/z [Da]
a. DSG_DNSG-sig model	AHR	LASLLPFPQDVINK	777.94803
a. DSG_DNSG-sig model	AHR	NDFSGEVDFR	593.2622
a. DSG_DNSG-sig model	AHR	SFFDVALK	463.75358
a. DSG_DNSG-sig model	ATP5S	DYNHLPTGPLDK	457.23033
a. DSG_DNSG-sig model	ATP5S	TALPSLELK	486.29223
a. DSG_DNSG-sig model	C11orf31	AEAADVVAEK	529.29855
a. DSG_DNSG-sig model	C11orf31	FPEPQEVVEELK	722.37176
a. DSG_DNSG-sig model	C11orf31	LEAPELPVK	498.29255
a. DSG_DNSG-sig model	C11orf31	NAAALSQALR	507.78859
a. DSG_DNSG-sig model	CDC42SE2	GGYGGGMPANVQMQLVDTK	961.95833
a. DSG_DNSG-sig model	CHP2	IIESFFPDGSQR	698.34931
b. XSG_XNSG-sig model	DYNLRB1	GVQGIIVNTEGIPIK	818.98537
b. XSG_XNSG-sig model	DYNLRB1	STMDNPTTTQYASLMHSFILK	1193.5759
b. XSG_XNSG-sig model	DYNLRB1	DIDPQNDLTFLR	723.86482
b. XSG_XNSG-sig model	FAM82B	LLVYEALEYAK	656.36311
b. XSG_XNSG-sig model	FAM82B	QIQTEAAQLLTSFSEK	897.46675
b. XSG_XNSG-sig model	FAM82B	ESEDAELLWR	624.29973
b. XSG_XNSG-sig model	FLG2	NPDDPDTVDMHMLDR	991.95196
b. XSG_XNSG-sig model	FLG2	SVVTVIDVFYK	635.35899
b. XSG_XNSG-sig model	FNBP1	FEAWLAEVEGR	653.82494
b. XSG_XNSG-sig model	FNBP1	TEIELSYAK	527.27613
b. XSG_XNSG-sig model	FNBP1	ADYSSILQK	512.76921
b. XSG_XNSG-sig model	GCLC	SLFFPDEAINK	640.829
b. XSG_XNSG-sig model	GCLC	DPLTLFEELK	1091.56403
b. XSG_XNSG-sig model	GCLC	YDSIDSYLSK	595.7828
b. XSG_XNSG-sig model	LMAN2	LTVMTDLEDK	1164.58275
b. XSG_XNSG-sig model	LMAN2	LPTGYFFGASAGTGDLSDNHDIIISMK	910.76176
b. XSG_XNSG-sig model	LMAN2	DNVDDPTGNFR	625.27584
b. XSG_XNSG-sig model	LPXN	TSAAAQLDELMAHLTEMQAK	720.35421
b. XSG_XNSG-sig model	LPXN	DFLAMFSPK	528.2661
b. XSG_XNSG-sig model	MYCBP	HHLGAATPENPEIELLR	633.0017
b. XSG_XNSG-sig model	MYCBP	VLVALYEEPEKPNALDFLK	1138.11934
b. XSG_XNSG-sig model	MYCBP	SGVLDLTK	467.26625
b. XSG_XNSG-sig model	NIT1	LLEEYQLAR	618.33505
b. XSG_XNSG-sig model	NIT1	DPAETLHLSEPLGGK	782.40285
b. XSG_XNSG-sig model	NIT1	IDLNYLR	453.75561
b. XSG_XNSG-sig model	NUB1	IAETFGLQENYIK	763.39894
b. XSG_XNSG-sig model	NUB1	VDNLLQLGFTAQEAR	837.94435
b. XSG_XNSG-sig model	NUB1	TTGIATIEVFLPPR	757.93262
b. XSG_XNSG-sig model	RAB32	ATIGVDFALK	517.79737
b. XSG_XNSG-sig model	RAB32	VLVIGELGVGK	542.34155
b. XSG_XNSG-sig model	RAB32	DNINIEEAAR	572.78302
b. XSG_XNSG-sig model	RBBP7	EMFEDTVEER	642.77431
b. XSG_XNSG-sig model	RBBP7	GEFGGFGSVTGK	571.77703
b. XSG_XNSG-sig model	RFC2	LNEIVGNEDTVSR	723.36492
b. XSG_XNSG-sig model	RFC2	EGNVPNIIAGPPGTGK	817.45021
b. XSG_XNSG-sig model	RFC2	LTDAILTR	515.79832

Q23. The authors propose MSI tumours to be more sensitive to drug treatments in general due to higher activated immune response in Fig3m. However, the association of MSI/MSS-sig level and therapy response is very weak and also only marginally significant at $p=0.045$ (Fig 3d). Since the authors subsequently show that MSI-sig high GCs benefit only from DOS and not XELOX (Fig 4a), would the analysis in Fig3 benefit from analysing different drug treatment groups separately?

Response to Q23:

We thank the reviewer for the excellent suggestion! According to reviewer's suggestion, we distinguished patients receiving different treatments (DOS and XELOX) in the analysis of exploring the association of MSI/MSS-sig level and therapy response in the revision. As shown in **Figure RL20 (see also Supplementary Fig. 5a in the revision)**, we observed that the GC patients of S group in DOS subcohort had significantly higher MSI-sig level than NS group ($P < 0.05$), while no obvious difference between S and NS groups in XELOX subcohort ($P > 0.05$). That indicated MSI/MSS-sig level showed significant association with response of DOS therapy, but not XELOX therapy. As the reviewer's proposed and expected, we noticed that MSI/MSS-sig level exhibited more significant difference ($P = 0.02$) in response to DOS therapy during the analysis of separate therapy, compared with the marginal significance ($P = 0.049$) for the difference in response to overall treatments. This finding supported our previous result that the MSI/MSS-sig level showed significant association with the response to therapy. In addition, the reviewer's suggestion also promoted us finding a novel point that patients with high MSI-sig could benefit only from DOS but not XELOX. We thank the reviewer again sincerely. In the revision, we analyzed the association of MSI/MSS-sig level with DOS and XELOX therapy, separately, and have updated the relevant figure (**Supplementary Fig. 5a**) and description (**line 430–432 in Page 14–15**) in the revision as following: "Surprisingly, we observed DSG had higher MSI-sig level than DNSG ($P = 0.02$), while no obvious difference between XSG and XNSG ($P > 0.05$) (Supplementary Fig. 5a).".

Figure RL20. Boxplot for the MSI/MSS-sig score between S and NS groups in DOS subcohort (a) and XELOX subcohort (b), respectively.

Q24. The combination of anti-HER2 therapy to overcome XELOX resistance is an interesting finding and could be further pursued and/or supported by experimental evidence. For e.g., does combinatorial treatment of anti-HER2 and XELOX have synergistic effects *in vitro*?

Response to Q24:

Thanks for the positive comment and valuable suggestion to improve the quality of our manuscript. In the study, to search for the indicators in response to DOS and XELOX therapies, we focused on the overlapped up-regulated proteins and pathways in sensitive or non-sensitive response to DOS and XELOX therapy. Among the overlapped proteins, we found ERBB2 could serve as an independent prognostic factor in the multivariable analysis after adjusting for Lauren's type, grade and RECIST in XELOX, but not DOS subcohort. ERBB2, as known as HER2, was the only drug target approved for the first-line treatment of HER2-positive GC. Therefore, we speculated that XELOX combined with anti-HER2 (Trastuzumab, Herceptin) targeted therapy could partially relieve the XELOX resistance.

According to reviewer's suggestion, we performed further *in vitro* validation experiment to confirm the synergistic effects of the combination of anti-HER2 and XELOX therapy. We treated NCI-N87 cells (HER2-positive cell line) with XELOX, trastuzumab, and combination of trastuzumab and XELOX, respectively. The drug sensitivity was estimated by their half-maximal

inhibitory concentration (IC₅₀) values. We observed the IC₅₀ values of XELOX and trastuzumab were 36.6 μ M and 43.7 μ M in NCI-N87 cells (**Figure RL21a**). As indicated in **Figure RL21b**, the IC₅₀ value of the trastuzumab and XELOX combined therapy was decreased to 19.6 μ M. These results indicated trastuzumab could increase the sensitivity of HER2-amplified human gastric cancer cells to XELOX therapy.

In the revision, we have updated the results and relevant description in **Fig. 4j**, and **line 496–502 in Page 18** in the “Result” section.

Figure RL21. The synergistic effects *in vitro* of the combination of trastuzumab and XELOX therapy. **a** Dose-response curves of NCI-N87 cells after 72-h treatments with XELOX, trastuzumab, and combination of trastuzumab and XELOX, (mean \pm SD, N = 3 biological repeats). IC₅₀, half-maximal inhibitory concentration; TRA, trastuzumab. **P*-values calculated using two-tailed Student’s *t* test. **b** The comparison of IC₅₀ values of XELOX, trastuzumab, trastuzumab and XELOX combined therapies.

Q25. It is also not clear how anti-HER2 therapy “alleviates the immune-related XELOX resistance” and “activates anticancer immune response” (page 15). It appears that the authors have only presented evidence that GC patients with high TCR signalling are unlikely to benefit from XELOX, but instead respond to XELOX + anti-HER2. Whether the role of anti-HER2 therapy affects TCR signalling has not been directly addressed.

Response to Q25:

We apologize for the unclear statement related to “alleviates the immune-related XELOX resistance” and “activates anticancer immune response”. During the comparative analysis of the combination of XELOX with anti-HER2 targeted therapy and XELOX chemotherapy alone, we observed that TCR signaling was activated and abundance of immune cells were increased in GC patients of XNSG or XHSG, compared with XSG or XHNSG respectively. The result indicated that GC patients with high TCR signaling are unlikely to benefit from XELOX, but instead respond to XELOX + anti-HER2. Therefore, we updated the relative statement as following: 1) Abstract section (**line 36–38 in Page 2**): “Further comparative analysis and validation experiment revealed the patients with high TCR signaling were unlikely to benefit from XELOX, but instead respond to the combination of anti-HER2-based therapy”, instead of “The combination of anti-HER2-based therapy relieved the immune-related XELOX resistance”; 2) Instruction section (**line 136–138 in Page 5**): “Further comparative analysis revealed the patients with high TCR signaling were unlikely to benefit from XELOX, but instead the combination of anti-HER2-based therapy”, instead of “The combination of XELOX with anti-HER2 targeted therapy partially relieved the immune-related XELOX resistance”; 3) Result section (**line 493–494 in Page 18**): “Therefore, we speculated that XELOX combined with anti-HER2 (Trastuzumab, Herceptin) targeted therapy could improve the therapeutic response”, instead of “Therefore, we speculated that XELOX combined with anti-HER2 (Trastuzumab, Herceptin) targeted therapy could partially relieved the XELOX resistance”; 4) Result section (**line 533–536 in Page 20**): “the combination of trastuzumab with XELOX therapy could resulted in a synergistic antitumor effect”, instead of “the combination of trastuzumab with XELOX therapy could activate anticancer immune response”.

To directly address the result that patients with high TCR signaling or increased immune cells would benefit from the combination therapy of XELOX and anti-HER2 but not XELOX therapy, we performed immunohistochemistry (IHC) of T-cell marker CD8+ and CD4+ to evaluate tumor-infiltration lymphocytes (IT-TILs) in FFPE tumor tissue, from patients receiving XELOX therapy or combined with anti-HER2 therapy in the revision. Here, we included sensitive and non-sensitive patients treated with XELOX therapy (XSG and XNSG) or XELOX + HER2 therapy (XHSG and XHNSG). The rabbit antibody against CD4 (GeneTech, catalog No: GT219107),

and the mouse antibody against CD8 (Leica, catalog NO: PA0183), were used for IHC. IHC evaluation was analyzed using an IHC profiler compatible plugin with integrated options for the quantitative analysis of digital IHC images stained for cytoplasmic or nuclear proteins. As a result, we observed the expression of CD4 and CD8 was significantly increased in FFPE tumor tissues from patients of XNSG compared with XSG. While in FFPE tumor tissues from patients of XHSG, we observed the expression of CD4 and CD8 was significantly increased, compared with XHNSG (**Figure RL22a, c, e, g, see also Supplementary Fig. 5g**). Moreover, the intensity of the cytoplasmic staining and the percentage of positively stained tumor cells were also scored numerically. The result showed XHSG had significantly increased percentage of CD4 positive cells (43.7%) and CD8 positive cells (39.2%) than XHNSG (3.2% and 2.9%, respectively) ($P < 1E-4$); while the percentage of CD4 positive cells and CD8 positive cells were higher in XNSG (54.2% and 29.7%, respectively) compared with XSG (2.9% and 1.4%, respectively) ($P < 0.05$) (**Figure RL22b, d, f, h, see also Fig. 4m**). Collectively, these results verified the findings in the manuscript that GC patients with high TCR signaling are unlikely to benefit from XELOX, but respond to XELOX + anti-HER2 instead.

In the revision, we corrected the relative description shown in **line 36–38 in Page 2, line 136–138 in Page 5, line 493–494 in Page 18, line 533–536 in Page 20**, and have updated the result in the part of “T cell receptor signaling pathway exerts diverse effects in response to DOS and XELOX chemotherapy” in “**Result**” section (see **line 520–537 in Page 19–20**).

Figure RL22. Immunohistochemistry (IHC) staining (**a, c, e, g**) and qualification (**b, d, f, h**) of CD4 and CD8 in representative examples in the XELOX and XELOX + HER2 subcohorts. The scale bar indicates 50 μ m. **P*-values were calculated by two-tailed Student's *t* test. **** *P* < 0.0001.

Q26. The authors propose ECM proteins as negatively associated with anti-HER2 therapy (Fig 5). However most of the non-responders to anti-HER2 therapy seem to fall in G-II proteomic subtype characterized by glycolysis and panthothenate/CoA biosynthesis (Fig 2b). Can the authors also further explore/discuss the role of glycolytic pathway etc in anti-HER2 therapy resistance?

Response to Q26:

Thanks for the comment. In this study, the statistical analysis revealed the significant distribution difference of therapy subcohorts among proteomic subtypes, among which HER2

subcohort was mainly enriched in G-II subtype (Fisher's exact test, $P < 0.05$) (**Figure RL23a, b**). The distribution difference of HER2 subcohort among proteomic subtypes could be due to the selection of HER2-positive patients for anti-HER2 targeted therapy. HER2-positive or negative status was evaluated by immunohistochemistry (IHC) and FISH (**Supplementary Data 1 and Table 1**). As expected, we observed the higher proportion (71%) of HER2-positive patients was enriched in G-II subtype compared with other subtypes (G-I: 21%, G-III: 43%, G-IV: 40%) (**Figure RL23c**). This result demonstrated the specific molecular pattern in HER2-positive GC patients, which could be identified at proteome level. Then, we surveyed the proportion of sensitive/non-sensitive patients of HER2 subcohort (HSG and HNSG) among four proteomic subtypes (shown in **Supplementary Fig. 2e in the revision**). The result suggested that in G-II subtype, HSG and HNSG accounted for 46% and 54%, respectively, indicating the response of anti-HER2 therapy showed no difference in G-II subtype (**Figure RL23d**). Therefore, glycolysis/gluconeogenesis and pantothenate/CoA biosynthesis were not the feature pathway of HSG or HNSG. To verify this finding, we compared the protein expression of the proteins involved in glycolysis and pantothenate/CoA biosynthesis between HSG and HNSG. The result showed there was no difference of these proteins expression between HSG and HNSG (**Figure RL23e**).

To investigate functional characteristics in sensitive or non-sensitive response to anti-HER2 targeted therapy, we performed further comparative analysis of HSG and HNSG. The KEGG pathway enrichment analysis revealed that HNSG was featured by ECM-receptor interaction. In addition, ECM was negatively correlated with clinical outcome. Further downstream analysis revealed that activation of ECM/PI3K-AKT pathway was related to resistance to anti-HER2 targeted therapy. Consistently, we observed the similar pathway enrichment (ECM-receptor interaction, etc.) in G-IV subtype, and the G-IV subtype had the largest proportion (75%) of HNSG. Finally, we have added the relevant description of anti-HER2 resistance related pathways in the "Discussion" part (**line 867–877 in Page 31–32**).

Figure RL23. The distributions of HER2 subcohort and its response among four proteomic subtypes. **a** Sankey plot showing the association of S/NS of HER2 subcohort with proteomic subtypes. **b** The barplot for each subcohort among four subtypes. **c** The barplot for HER2-positive status and HER2-negative status among four subtypes. **d** The barplot for S/NS of HER2 subcohort among four subtypes. **e** The heatmap showing the abundance of proteins involved in glycolysis and pantothenate/CoA biosynthesis between HSG and HNSG.

Q27. To further support Fig5, does PI3K-AKT inhibition synergize with anti-HER2?

Response to Q27:

Thanks for the comment. In this study, the comparative proteomic analysis of sensitive/non-sensitive groups of HER2 subcohort (HSG and HNSG) revealed that ECM and PI3K-AKT signaling pathway was activated in HNSG. Further analysis demonstrated that PI3K-AKT signaling pathway showed the highest positive correlation with ECM-receptor interaction pathway (Pearson $r = 0,72$, $P = 2.4E-12$). Based on this finding, we also speculated there was potential synergistic effect of PI3K-AKT inhibition in combination with anti-HER2 therapy, as reviewer mentioned. To support the result concluded in Fig. 5g, we firstly surveyed the relevant studies of PI3K-AKT inhibition in combination with anti-HER2 therapy within recent years (Nat Rev Cancer, 2009, PMID: 19629070; *Cancer Cell*, 2007, PMID: 17936563; *Oncogene*, 2011, PMID: 21278786)^{11, 12, 13}, which verified the potential value of targeting PI3K-AKT inhibition in anti-HER2 therapy.

To validate the hypothesis, we performed further *in vitro* validation experiment to confirm the synergistic effects of the combination of anti-HER2 and PI3K-AKT inhibitors in the revision. As reported in the previous study, isorhamnetin, a flavonoid compound extracted from *Hippophae rhamnoides L* and other plants (*Pharmacol Res*, 2006, PMID: 16765054) ¹⁴, can effectively suppress the proliferation, cellular morphology, and metastasis of tumor cells, both *in vivo* and *in vitro* via inactivating the PI3K/AKT signaling cascade (*Front Pharmacol*, 2021, PMID: 33679411) ¹⁵. In the revision, we treated NCI-N87 cells (HER2-positive cell line) with trastuzumab, isorhamnetin, and combination of trastuzumab and isorhamnetin, respectively. The drug sensitivity was estimated by their half-maximal inhibitory concentration (IC₅₀) values. We observed the IC₅₀ values of isorhamnetin and trastuzumab were 24.7 μM and 43.7 μM in NCI-N87 cells, respectively (**Figure RL24a**). As indicated in **Figure RL24b**, the IC₅₀ value of the trastuzumab and PI3K-AKT inhibitor combined therapy was decreased to 11.2 μM. These results suggested isorhamnetin could synergize with trastuzumab, resulting an enhanced anti-tumor effect, and provided a promising therapeutic strategy for HER2-positive GC patients. Overall, the PI3K-AKT inhibition combined with anti-HER2 therapy could represent an improved therapy strategy for HER2-positive GC patients.

In the revision, we have updated the results and relevant description in **Fig. 5i**, and **line 570–583 in Page 21** in the “Result” section.

Figure RL24. The synergistic effects *in vitro* of the combination of anti-HER2 and PI3K-AKT inhibition. a Dose-response curves of NCI-N87 cells after 72-h treatments with isorhamnetin, trastuzumab, and combination of trastuzumab and isorhamnetin (mean ± SD, N = 3 biological repeats). IC₅₀, half-maximal inhibitory concentration; ISO, isorhamnetin. *P-

values calculated using two-tailed Student's t test. **b** The comparison of IC50 values of isorhamnetin, trastuzumab, trastuzumab and isorhamnetin combined therapies.

Reference

1. Gallien S, Duriez E, Crone C, Kellmann M, Moehring T, Domon B. Targeted Proteomic Quantification on Quadrupole-Orbitrap Mass Spectrometer. *Mol Cell Proteomics* **11**, 1709-1723 (2012).
2. Eisenberg E, Levanon EY. Human housekeeping genes, revisited. *Trends Genet* **29**, 569-574 (2013).
3. Hoshino A, *et al.* Extracellular Vesicle and Particle Biomarkers Define Multiple Human Cancers. *Cell* **182**, 1044-+ (2020).
4. Zheng X, *et al.* A circulating extracellular vesicles-based novel screening tool for colorectal cancer revealed by shotgun and data-independent acquisition mass spectrometry. *J Extracell Vesicles* **9**, (2020).
5. Bang YJ, Van Cutsem E, Feyereislova A, Investigators TT. Trastuzumab in combination with chemotherapy versus chemotherapy alone for treatment of HER2-positive advanced gastric or gastro-oesophageal junction cancer (TOGA): a phase 3, open-label, randomised controlled trial (vol 376, pg 687, 2010). *Lancet* **376**, 1302-1302 (2010).
6. Gong JF, *et al.* Optimal regimen of trastuzumab in combination with oxaliplatin/capecitabine in first-line treatment of HER2-positive advanced gastric cancer (CGOG1001): a multicenter, phase II trial. *Bmc Cancer* **16**, (2016).
7. Ryu MH, *et al.* Multicenter phase II study of trastuzumab in combination with capecitabine and oxaliplatin for advanced gastric cancer. *Eur J Cancer* **51**, 482-488 (2015).
8. Mun DG, *et al.* Proteogenomic Characterization of Human Early-Onset Gastric Cancer. *Cancer Cell* **35**, 111-+ (2019).
9. Ni XT, *et al.* A region-resolved mucosa proteome of the human stomach. *Nat Commun* **10**, (2019).
10. Aran D, Hu ZC, Butte AJ. xCell: digitally portraying the tissue cellular heterogeneity landscape. *Genome Biol* **18**, (2017).
11. Serra V, *et al.* PI3K inhibition results in enhanced HER signaling and acquired ERK

- dependency in HER2-overexpressing breast cancer. *Oncogene* **30**, 2547-2557 (2011).
12. Berns K, *et al.* A functional genetic approach identifies the PI3K pathway as a major determinant of trastuzumab resistance in breast cancer. *Cancer Cell* **12**, 395-402 (2007).
 13. Engelman JA. Targeting PI3K signalling in cancer: opportunities, challenges and limitations. *Nat Rev Cancer* **9**, 550-562 (2009).
 14. Teng BS, Lu YH, Wang ZT, Tao XY, Wei DZ. In vitro anti-tumor activity of isorhamnetin isolated from *Hippophae rhamnoides* L. against BEL-7402 cells. *Pharmacol Res* **54**, 186-194 (2006).
 15. Zhai TY, *et al.* Isorhamnetin Inhibits Human Gallbladder Cancer Cell Proliferation and Metastasis via PI3K/AKT Signaling Pathway Inactivation (vol 12, 628621, 2021). *Front Pharmacol* **12**, (2021).

Reviewers' Comments:

Reviewer #1:

Remarks to the Author:

The authors addressed all the raised concerns.

Reviewer #2:

Remarks to the Author:

The authors have adequately addressed my previous concerns.

Reviewer #3:

Remarks to the Author:

The authors have adequately addressed my major concerns highlighted in the original review. There are 2 follow-up questions:

1) For the drug synergism experiments in Figure RL21 and Figure RL24, the authors should clarify and state explicitly the ratio of drug used in the combination. It would only make sense to report a single IC50 value for the drug combination treatment if a 1:1 ratio was used (i.e. same IC50 for each drug).

2) In Figure RL24, the choice of isorhamnetin as a pharmacological means of inhibiting PI3K/AKT signalling cascade is less convincing given that many specific PI3K/AKT pathway inhibitors are available (both approved for clinical use or in clinical trials). Can the authors support their findings with more mainstream modes of PI3K/AKT inhibition?

The point-to-point response are as follows.

Reviewer #1 (Remarks to the Author):

The authors addressed all the raised concerns.

Response:

Thanks for the reviewer's suggestions for the improvement of the article. We are very happy that the reviewer is satisfied with our article and response, and supports our publication in Nature Communications.

Reviewer #2 (Remarks to the Author):

The authors have adequately addressed my previous concerns.

Response:

Thanks for the reviewer's suggestions for the improvement of the article. We are very happy that the reviewer is satisfied with our article and response, and supports our publication in Nature Communications.

Reviewer #3 (Remarks to the Author):

The authors have adequately addressed my major concerns highlighted in the original review.

There are 2 follow-up questions:

1) For the drug synergism experiments in Figure RL21 and Figure RL24, the authors should clarify and state explicitly the ratio of drug used in the combination. It would only make sense to report a single IC50 value for the drug combination treatment if a 1:1 ratio was used (i.e. same IC50 for each drug).

Response: Thanks for the reviewer's comments. We apologize for not clarifying the ratio of drugs used in the combination treatments. The drug synergism experiments shown in Figure RL21 and Figure RL24 are about the combination treatments of trastuzumab and XELOX, as well as the PI3K-AKT inhibitor, separately. For these combination treatments, trastuzumab and XELOX or the PI3K-AKT inhibitor were added to the gastric cancer cells at a 1:1 ratio (Cancer Res, 2021, PMID: 34711611)¹. The IC50 values were calculated as the concentrations of single drug. We apologize for only reporting a single IC50 value of the combination therapies in the previous version again.

According to reviewer's suggestion mentioned in Q2, we performed further *in vitro* validation experiment with buparlisib (BKM120), which is a commonly used potent, pan-class I PI3K inhibitor approved for clinical trials (J Clin Oncol, 2012, PMID: 22162589; Cancer Cell, 2022, PMID: 35120601; Nat Commun, 2022, PMID: 35523793)²⁻⁴, to support the synergistic effects of the combination of anti-HER2 and the PI3K-AKT inhibitor in the revision. We treated NCI-N87 cells with trastuzumab (TRA), buparlisib (BUP), and combination of trastuzumab and buparlisib (BUP+TRA) with a ratio of 1:1, respectively. The results have been updated and summarized as follows.

The IC50 curves related to different treatments are shown in **Figure RL1a**: BUP (green), TRA (orange), BUP+TRA (blue, representing BUP+TRA combination treatment). Due to the ratio of BUP and TRA in the combination treatment is 1:1, the blue curve represented the same result of BUP or TRA in the combination treatment. To be more clearly, we presented the IC50 values of single drug in each treatment in **Figure RL1b**. We observed the IC50 values of BUP and TRA were 8.19 μ M and 46.57 μ M in NCI-N87 cells, respectively. As indicated in **Figure RL1b**, the IC50 value of BUP in the combination treatment (BUP+TRA) was significantly decreased to 3.44 μ M, compared with single BUP treatment (two-sided Student's t test, $P = 3.4E-3$). Similarly, the IC50 value of TRA in the combination treatment (BUP+TRA) was also significantly decreased to 3.44 μ M, compared with single TRA treatment (two-sided Student's t test, $P < 1.0E-4$). In the BUP+TRA combination therapy, the IC50 values of BUP were same as ones of TRA because of the 1:1 ratio of the two drugs. These results suggested BUP could synergize

with TRA, resulting an enhanced anti-tumor effect, and provided a promising therapeutic strategy for HER2-positive GC patients. Overall, the PI3K-AKT inhibition combined with anti-HER2 therapy could represent an improved therapy strategy for HER2-positive GC patients.

Additionally, we also accordingly updated presentation in the combination treatment of XELOX and TRA (**Figure RL1c, d**). As indicated in **Figure RL1d**, the IC50 value of the XELOX in the combination treatment (XELOX+TRA) was significantly decreased from 36.6 μM to 19.6 μM (two-sided Student's t test, $P = 1.4\text{E-}2$). Similarly, the IC50 value of TRA in the combination treatment (XELOX+TRA) was also significantly decreased from 43.7 μM to 19.6 μM (two-sided Student's t test, $P = 8.2\text{E-}3$).

Finally, in the revision, we have clarified the ratio of drugs in these combination treatments, and updated relevant descriptions in the **"Method"**, **"Result"**, and **"Figure Legends"** section. The **Figure RL 1a and b** have been updated in **Fig. 5i**, **Figure RL 1c and d** have been updated in **Fig. 4j**.

Figure RL 1. The synergistic effects *in vitro* of the combination of anti-HER2 with PI3K- AKT inhibition (**Figure RL 24 in the previous version**), and XELOX (**Figure RL 21 in the previous version**), respectively. **a** Dose-response curves of NCI-N87 cells after 72-h treatments with

buparlisib (BUP), trastuzumab (TRA), and combination of TRA and BUP with a ratio of 1:1. IC50, half-maximal inhibitory concentration. **b** The comparison of IC50 values of different therapies (two-sided Student's t test). Bars represent the mean of n = 3 independent experiments with error bars indicating SD. **c** Dose-response curves of NCI-N87 cells after 72-h treatments with XELOX, trastuzumab (TRA), and combination of TRA and XELOX with a ratio of 1:1. IC50, half-maximal inhibitory concentration. **d** The comparison of IC50 values of different therapies (two-sided Student's t test). Bars represent the mean of n = 3 independent experiments with error bars indicating SD.

2) In Figure RL24, the choice of isorhamnetin as a pharmacological means of inhibiting PI3K/AKT signaling cascade is less convincing given that many specific PI3K/AKT pathway inhibitors are available (both approved for clinical use or in clinical trials). Can the authors support their findings with more mainstream modes of PI3K/AKT inhibition?

Response: Thanks for the reviewer's comments. According to reviewer's suggestion, we performed further *in vitro* validation experiment with buparlisib (BKM120), which is a commonly used potent, pan-class I PI3K inhibitor approved for clinical trials (J Clin Oncol, 2012, PMID: 22162589; Cancer Cell, 2022, PMID: 35120601; Nat Commun, 2022, PMID: 35523793)²⁻⁴, to support the synergistic effects of the combination of anti-HER2 and the PI3K-AKT inhibitor in the revision. We treated NCI-N87 cells with trastuzumab (TRA), buparlisib (BUP), and combination of trastuzumab and buparlisib (BUP+TRA) with a ratio of 1:1, respectively. The drug sensitivity was estimated by their half-maximal inhibitory concentration (IC50) values. We observed the IC50 values of BUP and TRA were 8.19 μ M and 46.57 μ M in NCI-N87 cells, respectively (**Figure RL2a, Figure RL24a in the previous version**). As indicated in **Figure RL2b (Figure RL24b in the previous version)**, the IC50 value of BUP in the combination treatment (BUP+TRA) was significantly decreased to 3.44 μ M, compared with single BUP treatment (two-sided Student's t test, $P = 3.4E-3$). Similarly, the IC50 value of TRA in the combination treatment (BUP+TRA) was also significantly decreased to 3.44 μ M, compared with single TRA treatment (two-sided Student's t test, $P < 1.0E-4$). In the BUP+TRA combination therapy, the IC50 values of BUP were same as ones of TRA because of the 1:1 ratio of the two

drugs. These results suggested BUP could synergize with TRA, resulting an enhanced anti-tumor effect, and provided a promising therapeutic strategy for HER2-positive GC patients. Overall, the PI3K-AKT inhibition combined with anti-HER2 therapy could represent an improved therapy strategy for HER2-positive GC patients.

In the revision, we have updated our findings with buparlisib for PI3K/AKT inhibition in **Fig. 5i**, and supplemented relevant description in the “**Result**”, “**Method**”, and “**Figure Legends**” section.

Figure RL2. The synergistic effects *in vitro* of the combination of anti-HER2 and PI3K- AKT inhibition (**Figure RL24 in the previous version**). a Dose-response curves of NCI-N87 cells after 72-h treatments with buparlisib (BUP), trastuzumab (TRA), and combination of trastuzumab and buparlisib with a ratio of 1:1. IC₅₀, half-maximal inhibitory concentration. b The comparison of IC₅₀ values of different therapies (two-sided Student’s t test). Bars represent the mean of n = 3 independent experiments with error bars indicating SD.

References

- 1 Fan, F. S. *et al.* A Dual PI3K/HDAC Inhibitor Induces Immunogenic Ferroptosis to Potentiate Cancer Immune Checkpoint Therapy. *Cancer Res* **81**, 6233-6245, doi:10.1158/0008-5472.Can-21-1547 (2021).
- 2 Bendell, J. C. *et al.* Phase I, Dose-Escalation Study of BKM120, an Oral Pan-Class I PI3K Inhibitor, in Patients With Advanced Solid Tumors. *J Clin Oncol* **30**, 282-290, doi:10.1200/Jco.2011.36.1360 (2012).
- 3 Alam, A. *et al.* Fungal mycobiome drives IL-33 secretion and type 2 immunity in

pancreatic cancer. *Cancer Cell* **40**, 153-+, doi:10.1016/j.ccell.2022.01.003 (2022).

- 4 La, H. M. *et al.* Distinctive molecular features of regenerative stem cells in the damaged male germline. *Nat Commun* **13**, doi:ARTN 250010.1038/s41467-022-30130-z (2022).